# A Regularized Newton Method for Nonconvex Optimization with Global and Local Complexity Guarantees

**Yuhao Zhou[1], Jintao Xu[2], Bingrui Li[1], Chenglong Bao[3,4], Chao Ding[5], Jun Zhu[1]** *

[1]Department of Computer Science and Technology, Tsinghua AI Institute, BNRist Lab,
Tsinghua-Bosch Joint Center for ML, Tsinghua University
[2]Department of Applied Mathematics, The Hong Kong Polytechnic University
[3]Yau Mathematical Sciences Center, Tsinghua University
[4]Beijing Institute of Mathematical Sciences and Applications
[5]Academy of Mathematics and Systems Science, Chinese Academy of Sciences
`yuhaoz.cs@gmail.com` `xujtmath@163.com` `lbr22@mails.tsinghua.edu.cn`
`clbao@mail.tsinghua.edu.cn` `dingchao@amss.ac.cn` `dcszj@tsinghua.edu.cn`

## Abstract

Finding an $\epsilon$-stationary point of a nonconvex function with a Lipschitz continuous Hessian is a central problem in optimization. Regularized Newton methods are a classical tool and have been studied extensively, yet they still face a trade-off between global and local convergence. Whether a parameter-free algorithm of this type can simultaneously achieve optimal global complexity and quadratic local convergence remains an open question. To bridge this long-standing gap, we propose a new class of regularizers constructed from the current and previous gradients, and leverage the conjugate gradient approach with a negative curvature monitor to solve the regularized Newton equation. The proposed algorithm is adaptive, requiring no prior knowledge of the Hessian Lipschitz constant, and achieves a global complexity of $O(\epsilon^{-\frac{3}{2}})$ in terms of the second-order oracle calls, and $\tilde{O}(\epsilon^{-\frac{7}{4}})$ for Hessian-vector products, respectively. When the iterates converge to a point where the Hessian is positive definite, the method exhibits quadratic local convergence. Preliminary numerical results, including training the physics-informed neural networks, illustrate the competitiveness of our algorithm.

## 1 Introduction

Nonconvex optimization lies at the heart of numerous scientific and engineering applications, including machine learning [34] and computational physics [46]. In such settings, the objective is to minimize a smooth nonconvex function $\varphi : \mathbb{R}^n \to \mathbb{R}$ with a globally Lipschitz continuous Hessian. Given the intractability of finding a global minimum in general nonconvex problems, a more practical goal is to find an $\epsilon$-stationary point $x^*$ satisfying $\|\nabla \varphi(x^*)\| \leq \epsilon$ for a prescribed accuracy $\epsilon > 0$.

The Newton-type method is one of the most powerful tools for solving such problems, known for its quadratic local convergence near a solution with positive definite Hessian. The classical Newton method uses the second-order information at the current iterate $x_k$ to construct the following local model $m_k(d)$ and generate the next iterate $x_{k+1} = x_k + d_k$ by minimizing this model:

$$\min_{d \in \mathbb{R}^n} \left\{ m_k(d) := d^\top \nabla \varphi(x_k) + \frac{1}{2} d^\top \nabla^2 \varphi(x_k) d \right\}, \text{ where } k \geq 0. \tag{1.1}$$

---

*The corresponding author.

Although this method enjoys a quadratic local rate, it is well-known that it may fail to converge globally (i.e., converge from any initial point) even for a strongly convex function. Various globalization techniques have been developed to ensure global convergence by introducing regularization or constraints in (1.1) to adjust the direction $d_k$, including Levenberg-Marquardt regularization [35, 39], trust-region methods [8], and damped Newton methods with a linesearch procedure [44].

However, the original versions of these approaches exhibit a slow $O(\epsilon^{-2})$ worst-case performance [8, 5], leading to extensive efforts to improve the global complexity of second-order methods. Among these, the cubic regularization method [42] overcomes this issue and achieves an iteration complexity of $O(\epsilon^{-\frac{3}{2}})$, which has been shown to be optimal [4], while retaining the quadratic local rate. Meanwhile, Levenberg-Marquardt regularization, also known as quadratic regularization, with gradient norms as the regularization coefficients $\rho_k$, has also received several attentions due to its simplicity and computational efficiency [36, 45]. This method approximately solves the regularized subproblem $\min_d \left\{ m_k(d) + \frac{\rho_k}{2}\|d\|^2 \right\}$ to generate $d_k$ and the next iterate $x_{k+1} = x_k + \alpha_k d_k$, where $\alpha_k$ is either fixed or one selected through a linesearch. When the regularized subproblem is strongly convex, it is equivalent to solving the linear equation $(\nabla^2\varphi(x_k) + \rho_k \mathrm{I}_n)d_k = -\nabla\varphi(x_k)$, which is simpler than the cubic-regularized subproblem and can be efficiently implemented using iterative methods such as the *conjugate gradient* (CG). Furthermore, each CG iteration only requires a Hessian-vector product (HVP), facilitating large-scale problem-solving [53, 38, 37, 51, 55].

While such gradient regularization can preserve the superlinear local rate, the fast global rate has remained unclear for some time. Recent studies have achieved such iteration complexity for convex problems [41, 14]. Nevertheless, the regularized subproblem may become ill-defined for nonconvex functions. Consequently, modifications to these methods are necessary to address cases involving indefinite Hessians. A possible solution is to apply CG as if the Hessian is positive definite, and choose a first-order direction if evidence of indefiniteness is found [44], although this may result in a deterioration of the global rate. In contrast, Gratton et al. [24] introduced a method with a near-optimal global rate of $O(\epsilon^{-\frac{3}{2}} \log \frac{1}{\epsilon})$ and a superlinear local rate. Instead of relying on a first-order direction, their method switches to a direction constructed from the *minimal eigenvalue* and the corresponding eigenvector when indefiniteness is encountered.

On the other hand, Royer et al. [49] proposed the *capped CG* by modifying the standard CG method to monitor whether a negative curvature direction is encountered during the iterations, and switching to such a direction if it exists. It is worth noting that this modification introduces only one additional HVP throughout the entire CG iteration process, avoiding the need for the minimal eigenvalue computation used in Gratton et al. [24]. Furthermore, when the regularizer is *fixed*, an $O(\epsilon^{-\frac{3}{2}})$ global rate can be proved [49]. Building on this method, He et al. [29, 30] improved the dependency of the Lipschitz constant by adjusting the linesearch rule, and generalized it to achieve an optimal global rate for Hölder continuous Hessian, without requiring prior knowledge of problem parameters. Despite the appealing global performance, it is unclear whether the superlinear local rate can be preserved using these regularizers. Along similar lines, Zhu and Xiao [56] combined the gradient regularizer with capped CG and established a superlinear local convergence rate, assuming either the error bound condition or global strong convexity. However, it remains unclear whether this holds for nonconvex problems that exhibit local strong convexity.

Motivated by the discussions above, our goal is to figure out whether the optimal global order can be achieved by a quadratic regularized Newton method (RNM) *without incurring the logarithmic factor*, while simultaneously achieving *quadratic local convergence*. Since the Hessian Lipschitz constant $L_H$ is typically unknown and large for many problems, we design our algorithm to avoid both the computation of minimal eigenvalues and the reliance on prior knowledge of $L_H$, yet still attain optimal dependence on $L_H$ in the global complexity bound. In this work, we develop a new class of regularizers and a parameter-free RNM that answers this question affirmatively and close this long-standing gap in RNMs. Our approach demonstrates competitive performance against other second-order methods on standard nonlinear optimization benchmarks, as well as in training medium-scale physics-informed neural networks for solving partial differential equations [46].

The remaining parts of this article are organized as follows: We list the notations used throughout the paper below. Background and our main results are provided in Section 2. Techniques of our method are outlined in Section 3, with detailed proofs deferred to the appendix. Finally, we present some preliminary numerical results to illustrate the performance of our algorithm in Section 4, and discuss potential directions in Section 5. We also provide further discussions of related work in Section A.

## 2 Background and our results

**Notations** We use $\mathbb{N}$, $[i]$, and $I_{i,j}$ to denote the set of non-negative integers, $\{1, \ldots, i\}$, and $\{i, .., j-1\}$, respectively. For a set $S$, $|S|$ denotes its cardinality, and $\mathbf{1}_{\{j \in S\}} = 1$ if $j \in S$, and $0$ otherwise. For a symmetric matrix $X$, $X \succ (\succeq) 0$, $\lambda_{\min}(X)$ and $\|X\|$ denote the positive (semi-)definiteness, minimal eigenvalue and spectral norm, respectively. $\mathrm{I}_n \in \mathbb{R}^{n \times n}$ is the identity matrix. The Big-O notation $f(x) = O(g(x))$ means that there exists $C > 0$ such that $|f(x)| \leq C|g(x)|$ for sufficiently large $x$, and $f(x) = \tilde{O}(g(x))$ has the same meaning, except that it suppresses polylogarithmic factors in $x$. Similarly, $f(x) = \Omega(g(x))$ denotes there exists $c > 0$ such that $|f(x)| \geq c|g(x)|$ for suffciently large $x$. $\|x\|$ is the Euclidean norm of $x \in \mathbb{R}^n$. For a sequence $\{x_k\}_{k \geq 0}$ generated by the algorithm, we define $g_k = \|\nabla \varphi(x_k)\|$, $\epsilon_k = \min_{j \leq k} g_j$, and $\Delta_\varphi = \varphi(x_0) - \inf \varphi$, $U_\varphi = \sup_{\varphi(x) \leq \varphi(x_0)} \|\nabla \varphi(x)\|$.

**Capped CG** The capped CG proposed by Royer et al. [49] solves the equation $\bar{H}\tilde{d} = -g$ using the standard CG, where $\bar{H} = H + 2\rho \mathrm{I}_n$. It also monitors whether the iterates generated by the algorithm are negative curvature directions, or the algorithm converges slower than expected. If such an evidence is found, the algorithm will output a negative curvature direction. Specifically, the algorithm outputs a pair $(\mathrm{d\_type}, \tilde{d})$ with $\mathrm{d\_type} \in \{\mathrm{SOL}, \mathrm{NC}\}$. When $\mathrm{d\_type} = \mathrm{NC}$, $\tilde{d}$ is a negative curvature direction such that $\tilde{d}^\top H \tilde{d} \leq -\rho\|\tilde{d}\|^2$; and when $\mathrm{d\_type} = \mathrm{SOL}$, $\tilde{d}$ approximately satisfies the equation. In both cases, the solution can be found within $\min(n, \tilde{O}(\rho^{-\frac{1}{2}}))$ HVPs. We provide the algorithm and its properties in Section B.

**Complexity of RNMs** Continuing from Section 1, we further discuss RNMs. The key to proving a global rate is the following descent inequality, or its variants [2, 49, 41, 14, 30, 29, 56, 24]:

$$\varphi(x_{k+1}) - \varphi(x_k) \leq -C \min\left(g_{k+1}^2 \rho_k^{-1}, \rho_k^3\right), \text{ where } k \geq 0. \tag{2.1}$$

The dependence on the future gradient $g_{k+1}$ arises from the inability to establish a lower bound on $\|d_k\|$ using only the information available at the current iterate, since once the iterations enter a superlinear convergence region, the descent becomes small. If we were able to choose $\rho_k$ such that the descent were at least $\epsilon^{\frac{3}{2}}$, then by telescoping the sum we would obtain $\varphi(x_k) - \varphi(x_0) \leq -Ck\epsilon^{\frac{3}{2}}$. The optimal global rate $O(\epsilon^{-\frac{3}{2}})$ would follow from $-Ck\epsilon^{\frac{3}{2}} \geq \varphi(x_k) - \varphi(x_0) \geq -\Delta_\varphi$. Therefore, the regularizer $\rho_k$ plays a central rule in the global rate. In the thread of work starting from Royer et al. [49], $\rho_k \propto \sqrt{\epsilon}$, and the required descent is guaranteed as long as $g_{k+1} \geq \epsilon$; otherwise, $x_{k+1}$ is a solution. Another line of works related to Mishchenko [41] and Gratton et al. [24] use $\rho_k \propto \sqrt{g_k}$. With this choice, a $g_k^{\frac{3}{2}}$ descent is achieved when $g_{k+1} \geq g_k$. However, when $g_{k+1} < g_k$, the descent becomes $g_{k+1}^2 g_k^{-\frac{1}{2}}$, but the control over $g_{k+1}$ is lost. To resolve this issue, the iterations are divided into two sets: a successful set $\mathcal{I}_s = \{k : g_{k+1} \geq g_k/2\}$ and a failure set $\mathcal{I}_f = \mathbb{N} \setminus \mathcal{I}_s$. It is shown that when $|\mathcal{I}_f|$ is large the gradient will decrease below $\epsilon$ rapidly; and otherwise, sufficient descent is still achieved. The logarithmic factor in the complexity of Gratton et al. [24] can be understood as follows: a sufficient descent occurs at least once in every $O(\log \frac{1}{\epsilon})$ iterations. Yet, as shown in Theorem 3.2, it actually occurs in every $O(\log \log \frac{1}{\epsilon})$ iterations.

**Local convergence** We say $\{g_k\}_{k \geq 0}$ has a superlinear local rate of order $1 + \bar{\nu}$ if $g_{k+1} = O(g_k^{1+\bar{\nu}})$ for sufficiently large $k$, and a quadratic local rate corresponds to the case $\bar{\nu} = 1$. Assuming $\nabla^2 \varphi(x^*) \succ 0$, then the classical Newton method achieves the quadratic local rate in a neighborhood of $x^*$, which we refer to as the *local region* in this paper. In the nonconvex setting, identifying whether an iterate lies within this region is challenging, as it requires knowledge of the solution $x^*$. To assess whether a given regularizer is possible to attain quadratic local convergence, we can consider the quadratic function $\varphi(x) = \|x\|^2$: the fixed regularizer $\rho_k \propto \sqrt{\epsilon}$ of Royer et al. [49] yields linear convergence, while a gradient-based regularizer $\rho_k \propto g_k^{\bar{\nu}}$ with $\bar{\nu} \in (0, 1]$ achieves a superlinear rate of order $1 + \bar{\nu}$ [12, 36, 19, 1, 40]. Hence, choosing $\bar{\nu} = \frac{1}{2}$ as in Gratton et al. [24] leads to a local rate of only $\frac{3}{2}$.

## 2.1 Intuitions and results

We adopt the standard assumption from Royer et al. [49], which also guarantees $\Delta_\varphi < \infty$ and $U_\varphi < \infty$. While the Lipschitz continuity assumption can be relaxed to hold only on the level set $L_\varphi(x_0)$ using techniques in He et al. [30], we retain this assumption for simplicity, as it is required for the descent lemma (Theorem C.1) and is orthogonal to our analysis.

**Assumption 2.1** (Smoothness). *The level set $L_\varphi(x_0) := \{x \in \mathbb{R}^n : \varphi(x) \le \varphi(x_0)\}$ is compact, and $\nabla^2\varphi$ is $L_H$-Lipschitz continuous on an open neighborhood of $L_\varphi(x_0)$ containing the trial points generated in Algorithm 1, where $x_0$ is the initial point.*

**The choice of regularizers**  The preceding discussion reveals a tension between global and local convergence in RNMs: near-optimal global rate requires $\rho_k \propto \sqrt{g_k}$, whereas quadratic local convergence demands a *much smaller* regularizer $\rho_k \lesssim g_k$. A principled approach to reconcile this trade-off is to dynamically adapt $\rho_k$ to meet these requirements. Ideally, we may set $\rho_k = \sqrt{g_k}\delta_k$, where $\delta_k = 1$ outside the local region to guarantee global complexity, and $\delta_k \lesssim \sqrt{g_k}$ within the local region to achieve quadratic convergence. However, this choice for $\delta_k$ is not practically implementable, as it presumes knowledge of whether the current iterate lies in the local region, which is typically unknown in the nonconvex setting. Instead, our adjustment scheme is motivated by the observation that, in the local region where superlinear convergence of order $1 + \bar\nu$ occurs, the ratio $\delta_k = g_k/g_{k-1} \le g_{k-1}^{\bar\nu}$ rapidly decays to zero. Hence, this ratio serves as a reasonable heuristic for reducing the regularizer and improving the convergence in the local region, though the extent of this improvement remains unclear. The technical analysis in Section 3 reveals that achieving a quadratic local rate requires a refined choice, namely $\delta_k^\theta = \min(1, g_k^\theta/g_{k-1}^\theta)$ with $\theta > 1$, which is smaller than the original ratio $g_k/g_{k-1}$. Additionally, for $\theta \in (0, 1]$, the local rate can still be improved, albeit sub-quadratically, as illustrated in Figure 1 and formalized in Theorem 3.7.

Outside the local region, although the convergence is typically linear or sublinear such that $\delta_k \approx c \in (0, 1]$ and $\rho_k = \sqrt{g_k}\delta_k^\theta \propto \sqrt{g_k}$ for most itertaions, there may still be occasional sharp drops in $g_k$ that cause $\delta_k$ to become extremely small, unintendedly reducing the regularizer and thereby degrading the global complexity. To address this issue, we observe that a necessary condition for entering the local region is that the sequence $\{g_k\}$ becomes monotonically decreasing. Based on it, we switch to the regularizer $\rho_k = \sqrt{g_k}$ whenever this condition is not satisfied. Lines 2-5 of Algorithm 1 describe this procedure, where $\omega_k^t$ corresponds to the choice $\sqrt{g_k}\delta_k^\theta$ for accelerating local convergence, and $\omega_k^f = \sqrt{g_k}$ serves as the fallback choice to maintain the global rate, and `NewtonStep` generates the next iterate based on these regularizers. In practice, $\delta_k$ rarely exhibits sharp drops, allowing the fallback step to be relaxed or even omitted (see Section G.2). Theoretically, as established in Theorem 3.2, at least one suitable $\rho_k$ can be identified within $O(\log\log \frac{1}{\epsilon})$ iterations, yielding an $O(\epsilon^{-\frac{3}{2}} \log\log \frac{1}{\epsilon})$ iteration complexity. Furthermore, our analysis reveals that the logarithmic factor comes from abrupt increases of $\sqrt{g_k}$ (Theorem 3.4). It also suggests that replacing $g_k$ with $\epsilon_k = \min_{j \le k} g_j$ in the regularizer eliminates this factor, thereby achieving the optimal global rate. This alternative can be interpreted as a mechanism that retains historical information through $\epsilon_k$, effectively preventing the growth of $\sqrt{\epsilon_k}$.

Thus far, the structure of our regularizers has taken the form $\rho_k = \omega_k^t = \omega_k^f \delta_k^\theta$, and our discussion has focused on complexity with respect to the tolerance parameter $\epsilon$, without addressing the dependence on the Hessian Lipschitz constant $L_H$. To attain the optimal global complexity with respect to $L_H$, we require $\rho_k = \sqrt{L_H}\omega_k^t$. However, since $L_H$ is typically unknown and may vary locally, we dynamically estimate it via the sequence $M_k$ using the subroutine `LipEstimation`, and set $\rho_k = \sqrt{M_k}\omega_k^t$ in L8. The update scheme for $M_k$ is derived from a thorough analysis of the algorithm (see Theorem C.1). Roughly speaking, if the actual descent $\Delta_k = \varphi(x_k) - \varphi(x_{k+1})$ is smaller than the predicted value from the analysis, this suggests that $M_k$ underestimates $L_H$, and we increase it; conversely, when the prediction is fulfilled, we attempt to decrease $M_k$. Our analysis shows that after $\tilde{O}(1)$ iterations, it produces a desirable estimation of $L_H$.

**The design of NewtonStep**  The subroutine `NewtonStep` follows the version of Royer et al. [49] and He et al. [30], utilizing the `CappedCG` subroutine defined in Section B to find a descent direction. The key modification in this subroutine is the linesearch rule in L11-15 for selecting the stepsize when the negative curvature direction is not detected, and the subroutine `LipEstimation`. The criterion (2.4) aligns with the classical globalization approach of Newton methods [17], and can be

shown to generate a unit stepsize (i.e., $\alpha = 1$) when the iteration is sufficiently close to a solution with a positive definite Hessian, leading to superlinear convergence (see Theorem E.3). However, as previously discussed, our regularizers may become small when a sharp drop of $g_k$ occurs, which also degrades the oracle complexity in terms of the function evaluations and HVPs. To address these issues, we introduce an additional criterion (2.5) to ensure that it remains uniformly bounded as the iteration progresses. In this criterion, the choice of $\hat{\alpha}$ in L12 is motivated by the observation that selecting the stepsize according to the r.h.s. of (D.9) guarantees acceptance in the linesearch. The role of $\hat{\alpha}$ is thus to approximate this stepsize, while leaving the unknown term on the r.h.s. to be determined adaptively by the linesearch procedure. Another modification is the introduction of the fifth parameter $\bar{\rho}$ and the additional TERM state of d_type in `CappedCG`. This state is triggered when the iteration number exceeds $\tilde{\Omega}(\bar{\rho}^{-\frac{1}{2}})$, and is designed to ensure non-degenerate oracle complexity in terms of HVPs.

**Complexity**    Combining all these components, we are able to obtain the complexity results summarized in Theorems 2.2 and 2.3. Table 1 also compares them with other RNMs for nonconvex optimization. All parameters aside from the regularizers can be chosen arbitrarily, provided they satisfy the requirements in Algorithm 1. For the regularizers in Theorem 2.2, Theorem 2.3 shows that the complexity in terms of HVPs is $\tilde{O}(\epsilon^{-\frac{7}{4}})$, matching the results in Carmon et al. [3], Royer et al. [49]. Moreover, the complexity in terms of the second-order oracle outputting $\{\varphi(x), \nabla\varphi(x), \nabla^2\varphi(x)\}$ is $O(\epsilon^{-\frac{3}{2}}) + \tilde{O}(1)$, attaining the lower bound of Carmon et al. [4] up to an additive $\tilde{O}(1)$ term coming from the lack of prior knowledge about $L_H$. Notably, the $\sqrt{L_H}$ scaling in the iteration complexity is also optimal [4].

**Theorem 2.2** (Iteration complexity, proof and the non-asymptotic version in Sections C.2 and E.1). *Let $\{x_k\}_{k\geq 0}$ be generated by Algorithm 1. Under Assumption 2.1 and define $\epsilon_k = \min_{0 \leq i \leq k} g_i$ with $g_{-1} = \epsilon_{-1} = g_0$, the following two iteration bounds hold for achieving the $\epsilon$-stationary point for $\theta \geq 0$:*

*1. If $\omega_k^{\mathrm{f}} = \sqrt{g_k}$, $\omega_k^{\mathrm{t}} = \omega_k^{\mathrm{f}}\delta_k^\theta$, and $\delta_k = \min(1, g_k g_{k-1}^{-1})$, then*

$$k \lesssim \Delta_\varphi L_H^{\frac{1}{2}} \epsilon^{-\frac{3}{2}} \log\log \frac{U_\varphi}{\epsilon} + |\log L_H| \log \frac{U_\varphi}{\epsilon}; \qquad (2.2)$$

*2. If $\omega_k^{\mathrm{f}} = \sqrt{\epsilon_k}$, $\omega_k^{\mathrm{t}} = \omega_k^{\mathrm{f}}\delta_k^\theta$, and $\delta_k = \epsilon_k \epsilon_{k-1}^{-1}$, then*

$$k \lesssim \Delta_\varphi L_H^{\frac{1}{2}} \epsilon^{-\frac{3}{2}} + |\log L_H| + \log \frac{U_\varphi}{\epsilon}. \qquad (2.3)$$

*Furthermore, there exists a subsequence $\{x_{k_j}\}_{j\geq 0}$ such that $\lim_{j\to\infty} x_{k_j} = x^*$ with $\nabla\varphi(x^*) = 0$. If $\theta > 1$ and $\nabla^2\varphi(x^*) \succ 0$, then the whole sequence $\{x_k\}$ converges to a local minimum $x^*$, and for sufficiently large $k$, quadratic local rate exists for both of these choices, i.e., $g_{k+1} \leq O(g_k^2)$.*

**Theorem 2.3** (Oracle complexity, proof in Section C.3). *Each iteration in the main loop of Algorithm 1 requires at most $2(m_{\max} + 1)$ function evaluations; and at most $2$ gradient evaluations; and either $1$ Hessian evaluation or at most $\min\left(n, \tilde{O}((\omega_k^{\mathrm{f}})^{-\frac{1}{2}})\right)$ HVPs.*

Finally, we note that the overall computational complexity can be viewed as the product of two factors: (i) the number of HVP evaluations required by the algorithm, and (ii) the cost of performing a single HVP evaluation. Since the cost of an individual HVP evaluation is typically fixed and does not vary across iterations, the complexity analysis reduces to counting the number of HVP evaluations, as given by the above theorem.

## 3    Overview of the techniques

We outline the key steps in this section and defer the complete proofs to Sections C and D.

**The global iteration complexity**    Let $\Delta_k = \varphi(x_k) - \varphi(x_{k+1})$ denote the objective function decrease at iteration $k$, and define the index set $\mathcal{N}_k = \{j \leq k : \Delta_j \gtrsim L_H^{-1/2}\epsilon^{3/2}\}$ to contain the iterations that achieve *sufficient descent*. As previously mentioned, a key step in the complexity

---

**Algorithm 1:** Adaptive regularized Newton-CG (**ARNCG**)

---

**Input** : Initial point $x_0 \in \mathbb{R}^n$, parameters $\mu \in (0, 1/2)$, $\beta \in (0, 1)$, $\tau_- \in (0, 1)$, $\tau_+ \in (0, 1]$, $\tau \in (0, 1]$, $\gamma \in (1, \infty)$, $m_{\max} \in [1, \infty)$, $M_0 \in (0, \infty)$, and $\eta \subseteq [0, 1]$, and regularizers $\{\omega_k^t, \omega_k^f\}_{k \geq 0} \subseteq (0, \infty)$ for trial and fallback steps.

1   **for** $k = 0, 1, \dots$ **do**                                        `// the main loop`

2      $(x_{k+\frac{1}{2}}, M_{k+1}) \leftarrow \texttt{NewtonStep}(x_k, \omega_k^t, M_k, \omega_k^f)$           `// trial step`

3      **if** *(the above step returns FAIL) or* $\left(g_{k+\frac{1}{2}} > g_k \text{ and } g_k \leq g_{k-1}\right)$ **then**

4         $(x_{k+1}, M_{k+1}) \leftarrow \texttt{NewtonStep}(x_k, \omega_k^f, M_k, \omega_k^f)$         `// fallback step`

5      **else** $x_{k+1} \leftarrow x_{k+\frac{1}{2}}$                      `// accept the trial step`

6   **Subroutine** $\texttt{NewtonStep}(x, \omega, M, \bar{\omega})$

7      $\tilde{\eta} \leftarrow \min\left(\eta, \sqrt{M}\omega\right)$

8      $(\texttt{d\_type}, \tilde{d}) \leftarrow \texttt{CappedCG}(\nabla^2 \varphi(x), \nabla \varphi(x), \sqrt{M}\omega, \tilde{\eta}, \tau\sqrt{M}\bar{\omega})$     `// see Section B`

9      **if** $\texttt{d\_type} = TERM$ **then return** FAIL          `// never reached if` $\omega \geq \bar{\omega}$

10     **else if** $\texttt{d\_type} = SOL$ **then**                       `// a normal solution`

11        Set $d \leftarrow \tilde{d}$ and $\alpha \leftarrow \beta^m$, where $0 \leq m \leq m_{\max}$ is the minimum integer such that

$$\varphi(x + \beta^m d) \leq \varphi(x) + \mu\beta^m d^\top \nabla\varphi(x). \tag{2.4}$$

12        **if** *the above m does not exist* **then**         `// switch to a smaller stepsize`

13           Set $\hat{\alpha} \leftarrow \min(1, \omega^{\frac{1}{2}} M^{-\frac{1}{4}} \|d\|^{-\frac{1}{2}})$

14           Set $\alpha \leftarrow \hat{\alpha}\beta^{\hat{m}}$, where $0 \leq \hat{m} \leq m_{\max}$ is the minimum integer such that

$$\varphi(x + \hat{\alpha}\beta^{\hat{m}} d) \leq \varphi(x) + \mu\hat{\alpha}\beta^{\hat{m}} d^\top \nabla\varphi(x). \tag{2.5}$$

15           **if** *the above $\hat{m}$ does not exist* **then return** $(x, \gamma M)$

16      **else**                  `// a negative curvature direction (d_type = NC)`

17        Set $\bar{d} \leftarrow \|\tilde{d}\|^{-1}\tilde{d}$ and adjust it to a descent direction with length $L(\bar{d})$:

$$d \leftarrow -L(\bar{d})\text{sign}\left(\bar{d}^\top \nabla\varphi(x)\right)\bar{d}, \quad \text{where } L(\bar{d}) := M^{-1}|\bar{d}^\top \nabla^2\varphi(x)\bar{d}|. \tag{2.6}$$

18        Set $\alpha \leftarrow \beta^m$, where $0 \leq m \leq m_{\max}$ is the minimum integer such that

$$\varphi(x + \beta^m d) \leq \varphi(x) - M\mu\beta^{2m}\|d\|^3. \tag{2.7}$$

19        **if** *the above m does not exist* **then return** $(x, \gamma M)$

20      $x^+ \leftarrow x + \alpha d$

21      $M^+ \leftarrow \texttt{LipEstimation}(x, x^+, \tau_-, \tau_+, \omega, M, \gamma, \beta, \mu, \texttt{d\_type})$

22      **return** $(x^+, M^+)$

23   **Subroutine** $\texttt{LipEstimation}(x, x^+, \tau_-, \tau_+, \omega, M, \gamma, \beta, \mu, \texttt{d\_type})$

24      $M^+ \leftarrow M$

25      $\Delta \leftarrow \varphi(x) - \varphi(x^+)$

26      **if** $\texttt{d\_type} = SOL$ *and* $m = 0$ *satisfies* (2.4) **then**

27        **if** $\Delta \leq \frac{4}{33}\mu\tau_+ M^{-\frac{1}{2}} \min\left(\|\nabla\varphi(x^+)\|^2 \omega^{-1}, \omega^3\right)$ **then** $M^+ \leftarrow \gamma M$

28        **else if** $\Delta \geq \frac{4}{33}\mu\tau_- M^{-\frac{1}{2}}\bar{\omega}^3$ **then** $M^+ \leftarrow \gamma^{-1} M$

29      **else if** $\texttt{d\_type} = SOL$ *and* $\Delta \leq \tau_+ \beta\mu M^{-\frac{1}{2}}\omega^3$ **then** $M^+ \leftarrow \gamma M$

30      **else if** $\texttt{d\_type} = NC$ *and* $\Delta \leq \tau_+(1 - 2\mu)^2\beta^2\mu M^{-\frac{1}{2}}\omega^3$ **then** $M^+ \leftarrow \gamma M$

31      **else if** $\Delta \geq \mu\tau_- M^{-\frac{1}{2}}\bar{\omega}^3$ **then** $M^+ \leftarrow \gamma^{-1} M$

32      **return** $M^+$

---

Table 1: Comparison of regularized Newton methods for nonconvex optimization. The parameter $M_k$ estimates $L_H$ and is independent of $\omega_k^f$ and $\omega_k^t$ in Theorem 2.2. For details, see arguments of CappedCG in Algorithm 1. We define $g_k = \|\nabla\varphi(x_k)\|$ and $\epsilon_k = \min_{i \le k} g_k$. The *additive* $\tilde{O}(1)$ terms in some algorithms come from $L_H$ estimation. "EPS" in the last column indicates that $\epsilon$ is used in the regularization coefficient, and "ME" means the method needs to compute the minimal eigenvalue to determine its parameters.

| Algorithm | Iteration Complexity | Local Order | Regularization Coefficient | Requirements |
|---|---|---|---|---|
| Royer et al. [49, Theorem 3] | $O(L_H^3 \epsilon^{-\frac{3}{2}})$ | $1^\ddagger$ | $\sqrt{\epsilon}$ | EPS |
| Zhu and Xiao [56, Theorem 5] | $O(L_H^2 \epsilon^{-\frac{3}{2}})$ | $1^\dagger$ | $2\tau_k g_k^\theta$ for $\tau_k \in [g_k^{-\theta}\sqrt{\epsilon}, \hat{\tau}g_k^{-\theta}\sqrt{\epsilon}]$ | EPS |
| He et al. [29, Theorem 1] | $O(L_H^{\frac{1}{2}} \epsilon^{-\frac{3}{2}})$ | $1^\ddagger$ | $\sqrt{M_k\epsilon}$ | EPS |
| Gratton et al. [24, Theorem 3.5] | $O(\max(L_H^2, L_H^{\frac{1}{2}})\epsilon^{-\frac{3}{2}}\log\frac{1}{\epsilon}) + \tilde{O}(1)$ | $1.5^\ddagger$ | $\sqrt{M_k g_k} + [-\lambda_{\min}(\nabla^2\varphi(x_k))]_+$ | ME |
| **Theorem 2.2** | $O(L_H^{\frac{1}{2}} \epsilon^{-\frac{3}{2}}\log\log\frac{1}{\epsilon}) + \tilde{O}(1)$ | 2 if $\theta > 1$ | $\sqrt{M_k g_k}\min(1, g_k^\theta g_{k-1}^{-\theta})$ for $\theta \ge 0$ | - |
| **Theorem 2.2** | $O(L_H^{\frac{1}{2}} \epsilon^{-\frac{3}{2}}) + \tilde{O}(1)$ | 2 if $\theta > 1$ | $\sqrt{M_k}\epsilon_k^{\frac{1}{2}+\theta}\epsilon_{k-1}^{-\theta}$ for $\theta \ge 0$ | - |

$\dagger$ Zhu and Xiao [56, Lemma 11] with $\beta = 1$ gives a linear rate.
$\ddagger$ The local rate is not mentioned in the original papers, see the discussions in Section 2.

analysis is to establish a lower bound on $|\mathcal{N}_k|$. For example, since $\Delta_\varphi \ge \varphi(x_0) - \varphi(x_k) \ge \sum_{j\le k}\Delta_j \gtrsim |\mathcal{N}_k|L_H^{-1/2}\epsilon^{3/2}$, then it follows that $k \propto |\mathcal{N}_k| \lesssim L_H^{1/2}\epsilon^{-3/2}$ as long as $|\mathcal{N}_k| \gtrsim k$.

To obtain such a lower bound, we first identify the conditions under which the dependence of $\Delta_k$ on $L_H$ is valid. Let the index sets $\mathcal{J}^i = \{k : M_{k+1} = \gamma^i M_k\}$ for $i = -1, 0, 1$ represent iterations where $M_k$ is decreased, unchanged, or increased, respectively. Our analysis in Theorem C.1 in appendix shows that for $k \in \mathcal{J}^0 \cup \mathcal{J}^{-1}$, the descent satisfies $\Delta_k \gtrsim M_k^{-1/2}D_k$, where $D_k$ captures the descent amount independent of $M_k$ and will be discussed subsequently. Therefore, establishing a lower bound on $|\mathcal{N}_k|$ reduces to counting the number of iterations where $M_k \lesssim L_H$ and $D_k \gtrsim \epsilon^{3/2}$ hold.

Theorem C.1 further establishes that if $k \in \mathcal{J}^1$, then $M_k \lesssim L_H$ holds. Since $M_k$ is only increased when $k \in \mathcal{J}^1$, we can conclude that $M_k \lesssim \max(M_0, L_H)$. As the bound also depends on the initial value $M_0$, the inequality $M_k \lesssim L_H$ does not hold when $M_k \gtrsim L_H$ is overestimated. However, in this case, we find that $M_k$ will be decreased (i.e., $k \in \mathcal{J}^{-1}$) as long as $g_k$ does not exhibit a sharp drop. Building on this, Theorem 3.5 establishes that a satisfactory estimate of $M_k$ can be obtained within $\tilde{O}(1)$ iterations. It remains to analyze how frequently the event $D_k \gtrsim \epsilon^{3/2}$ occurs throughout the iterations. Since under the choices of regularizers, we have either $\omega_k^f = \sqrt{g_k} \ge \sqrt{\epsilon}$ or $\omega_k^f = \sqrt{\epsilon_k} \ge \sqrt{\epsilon}$, then ensuring sufficient descent can be reduced to counting the occurrences of the event $D_k \ge (\omega_k^f)^3$.

Throughout this section, we partition $\mathbb{N}$ into a disjoint union of intervals $\mathbb{N} = \bigcup_{j\ge 1} I_{\ell_j, \ell_{j+1}}$ such that $0 = \ell_1$ and $\ell_j < \ell_{j+1}$ for $j \ge 1$, where $I_{i,j} = \{i, .., j-1\}$ is defined in the notation section. These intervals are constructed such that the following conditions hold for every $j \ge 1$:

$$g_{\ell_j} \ge g_{\ell_j+1} \ge \cdots \ge g_{\ell_{j+1}-1} \text{ and } g_{\ell_{j+1}-1} < g_{\ell_{j+1}}. \tag{3.1}$$

In other words, the sequence $\{x_k\}_{k\ge 0}$ is divided into subsequences where the gradient norms are non-increasing. The following lemma shows that sufficient descent occurs during the transition between adjacent subsequences, provided that $\ell_j - 1 \notin \mathcal{J}^1$. The fallback step is primarily designed to ensure this lemma holds. Without the fallback step, a sudden gradient decrease (i.e., a small $\delta_k$) could result in a small regularizer, causing the sufficient descent guaranteed by this lemma to vanish.

**Lemma 3.1** (Transition between adjacent subsequences, see Theorem C.2). *Under the regularizers in Theorem 2.2 with $\theta \ge 0$, we have $\omega_{\ell_j-1} = \omega_{\ell_j-1}^f$ for each $j > 1$, and*

$$\varphi(x_{\ell_j}) - \varphi(x_{\ell_j-1}) \lesssim -M_{\ell_j-1}^{-\frac{1}{2}}\mathbf{1}_{\{\ell_j-1\notin\mathcal{J}^1\}}(\omega_{\ell_j-1}^f)^3. \tag{3.2}$$

*Moreover, if $M_{\ell_j-1} \gtrsim L_H$, then $\ell_j - 1 \in \mathcal{J}^{-1}$.*

The following lemma characterizes the overall decrease of the function within a subsequence. It roughly states that there are at most $O\left(\log\log\frac{g_{\ell_j}}{g_k}\right)$ iterations with insufficient descent in the subsequence $I_{\ell_j, \ell_{j+1}}$, since otherwise the gradient decreases superlinearly below $g_k$.

**Lemma 3.2** (Iteration within a subsequence, see Theorem C.3). *Under the regularizers in Theorem 2.2 with $\theta \geq 0$, then for $j \geq 1$ and $\ell_j < k < \ell_{j+1}$, we have*

$$\varphi(x_k) - \varphi(x_{\ell_j}) \lesssim -C_{\ell_j,k}\left(|I_{\ell_j,k} \cap \mathcal{J}^{-1}| + \max\left(0, |I_{\ell_j,k} \cap \mathcal{J}^0| - T_{\ell_j,k} - 5\right)\right)(\omega_k^{\mathrm{f}})^3, \quad (3.3)$$

*where $C_{i,j} = \min_{i \leq l < j} M_l^{-\frac{1}{2}}$ and $T_{i,j} = 2\log\log\left(3(\omega_i^{\mathrm{f}})^2(\omega_j^{\mathrm{f}})^{-2}\right)$.*

Combining Theorems 3.1 and 3.2, we have the following proposition about the accumulated function descent, and find that there are $\Sigma_k$ iterations with sufficient descent.

**Proposition 3.3** (Accumulated descent, see Theorem C.4). *Under the choices of Theorem 2.2 with $\theta \geq 0$, for each $k \geq 0$, we have*

$$\varphi(x_k) - \varphi(x_0) \lesssim -C_{0,k}\big(\underbrace{|I_{0,k} \cap \mathcal{J}^{-1}| + \max\left(|S_k \cap \mathcal{J}^0|, |I_{0,k} \cap \mathcal{J}^0| - V_k - 5J_k\right)}_{\Sigma_k}\big)\epsilon_k^{\frac{3}{2}}, \quad (3.4)$$

*where $V_k = \sum_{j=1}^{J_k-1} T_{\ell_j, \ell_{j+1}} + T_{\ell_{J_k},k}$, and $S_k = \bigcup_{j=1}^{J_k-1}\{\ell_{j+1} - 1\}$, and $J_k = \max\{j : \ell_j \leq k\}$.*

The difference of the logarithmic factor in the iteration complexity of Theorem 2.2 arises from the following lemma, which provides an upper bound for $V_k$. This lemma shows that the choice $\omega_k^{\mathrm{f}} = \sqrt{\epsilon_k}$ leads to a better control over $V_k$ due to the monotonicity of $\epsilon_k$, resulting in improved lower bound for $\Sigma_k$, as indicated by Theorem D.7.

**Lemma 3.4** (See Section D.3). *Let $V_k, J_k$ be defined in Theorem 3.3, then we have (1). If $\omega_k^{\mathrm{f}} = \sqrt{g_k}$, then $V_k \leq J_k \log\log \frac{U_\varphi}{\epsilon_k}$; (2). If $\omega_k^{\mathrm{f}} = \sqrt{\epsilon_k}$, then $V_k \leq \log\frac{\epsilon_0}{\epsilon_k} + J_k$.*

Finally, we need to determine the aforementioned hitting time $k_{\mathrm{init}}$ such that $M_{k_{\mathrm{init}}} \leq O(L_H)$, and apply Theorem 3.3 for $\{x_k\}_{k \geq k_{\mathrm{init}}}$ to achieve the $L_H^{-\frac{1}{2}}$ dependence in the iteration complexity. The idea behind the following proposition is that when $M_k > \Omega(L_H)$ but $k \in \mathcal{J}^0$, we will find that the gradient decreases linearly, implying that this event can occur at most $O\left(\log\frac{U_\varphi}{\epsilon_{k_{\mathrm{init}}}}\right)$ times.

**Proposition 3.5** (Initial phase, see Theorem C.5). *Let $k_{\mathrm{init}} = \min\{j : M_j \leq O(L_H)\}$ and assume $M_0 > \Omega(L_H)$, then for the first choice in Theorem 2.2, we have $k_{\mathrm{init}} \leq O\left(\log\frac{M_0}{L_H}\log\frac{U_\varphi}{\epsilon_{k_{\mathrm{init}}}}\right)$; and for the second choice, we have $k_{\mathrm{init}} \leq O\left(\log\frac{M_0}{L_H} + \log\frac{U_\varphi}{\epsilon_{k_{\mathrm{init}}}}\right)$.*

**The local convergence order** From the compactness of $L_\varphi(x_0)$ in Assumption 2.1, we know there exists a subsequence $\{x_{k_j}\}_{j \geq 0}$ converging to some $x^*$ with $\nabla\varphi(x^*) = 0$ (see Theorem C.6). In the analysis of the local convergence rate, we need to assume the positive definiteness of $\nabla^2\varphi(x^*)$, under which the whole sequence $\{x_k\}_{k \geq 0}$ also converges to $x^*$ (see Theorem E.4). Analyzing the local convergence of RNMs requires establishing that the Newton direction $(\nabla^2\varphi(x_k) + \omega_k I)^{-1}\nabla\varphi(x_k)$ leads to superlinear convergence, and that it is eventually selected by the algorithm. Since the latter is algorithm-specific, we present its proof in Section F.1, and state the main results below.

**Lemma 3.6.** *Assuming $\nabla^2\varphi(x^*) \succeq \alpha I_n$, if $d\_type_k = \mathtt{SOL}$ and $m_k = 0$, and $x_k$ is close enough to $x^*$, we have $g_{k+1} \leq O(g_k^2 + \omega_k g_k)$. Furthermore, under the choices of regularizers in Theorem 2.2, if $x_k$ is close enough to $x^*$, we know the trial step is accepted, and $d\_type_k = \mathtt{SOL}$ and $m_k = 0$.*

*Remark on the local convergence neighborhood.* We observe that when taking $\omega_k^{\mathrm{t}} = \omega_k^{\mathrm{f}} = O(g_k^{\bar{\nu}})$ with $\bar{\nu} \in (0, 1]$, the gradient norm converges superlinearly with order $1 + \bar{\nu}$. For the choices in Theorem 2.2, we find $\max(\omega_k^{\mathrm{t}}, \omega_k^{\mathrm{f}}) \leq \sqrt{g_k}$ so a local rate of order $\frac{3}{2}$ can be achieved in the neighborhood $U_0$ independent of $\theta$, after which the proof of Theorem 3.6 guarantees that the trial step is accepted at $K_1 := O(\log\log \mathrm{poly}(L_H^\theta, U_\varphi^\theta))$ iterations. Then, the following technical lemma shows that the local order can be improved to arbitrarily close to $1 + \nu_\infty \in \left(\frac{3}{2}, 2\right]$ for $\theta > 0$ with $\nu_\infty$ defined in Theorem 3.7 (see Figure 1 for an illustration), and achieves quadratic convergence for $\theta > 1$ after $K_2 := 2\log\frac{2\theta-1}{2\theta-2} + 1$ steps. Hence, achieving quadratic convergence requires at most $K_1 + K_2$ extra iterations once the algorithm has entered the $\theta$-independent neighborhood $U_0$.

**Lemma 3.7** (Local rate boosting, proof in Section F.2). *Let $\theta > 0$ and $\{g_k\}_{k \geq 0} \subseteq (0, \infty)$. Suppose $g_1 \leq O\big(g_0^{\frac{3}{2}}\big)$ and $g_{k+1} \leq O\big(g_k^2 + g_k^{\frac{3}{2}} g_k^\theta g_{k-1}^{-\theta}\big)$ holds for each $k \geq 1$, and $g_0$ is sufficiently small.*

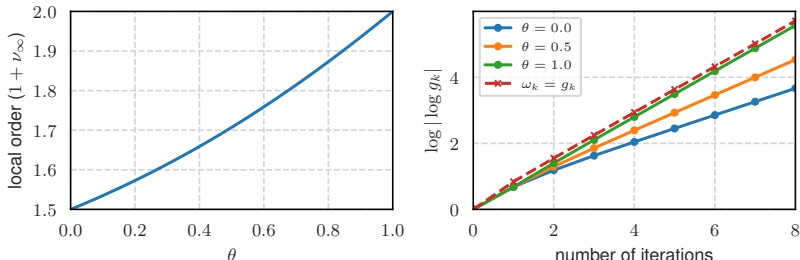

Figure 1: The left plot illustrates the local order achievable by the regularizers in Theorem 2.2 for $\theta \in (0, 1]$. It can be made arbitrarily close to $1 + \nu_\infty$. The right plot illustrates the local order for different $\theta$ using $\varphi(x) = x^2$, and its slope reflects the local order and aligns with our predictions.

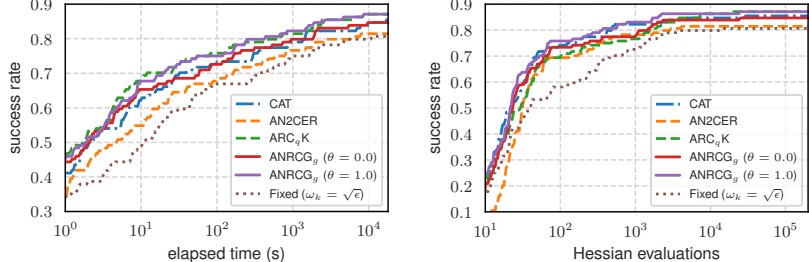

Figure 2: Comparison of success rates as functions of elapsed time and Hessian evaluations for CUTEst benchmark problems. $\mathbf{ARNCG}_g$, $\mathbf{ARNCG}_\epsilon$, and "Fixed" correspond to Algorithm 1 with the first and second regularizers from Theorem 2.2, and a fixed $\omega_k \equiv \sqrt{\epsilon}$, respectively. For Hessian evaluations, since our algorithm accesses this information only via Hessian-vector products, we count multiple products involving $\nabla^2\varphi(x)$ at the same point $x$ as a single evaluation.

*Then, (1). If $\theta \in (0, 1]$, let $\nu_\infty$ be the positive root of the equation $\frac{1}{2} + \frac{\theta\nu_\infty}{1+\nu_\infty} = \nu_\infty$, then we have $g_{k+1} \leq O\big(g_k^{1+\nu_\infty-(4\theta/9)^k}\big)$, i.e., $g_k$ has local order $1 + \nu_\infty - \delta$ for any $\delta > 0$; (2). If $\theta > 1$ and $k \geq 2\log\frac{2\theta-1}{2\theta-2} + 1$, then $g_{k+1} \leq O(g_k^2)$, i.e., $g_k$ converges quadratically.*

## 4  Preliminary numerical results

In this section, we present some preliminary numerical results to provide an overall sense of our algorithm's performance and the effects of its components, and to illustrate the potential application in training physics-informed neural networks. Detailed results are deferred to Sections G and H.

**CUTEst benchmark**    Since the recently proposed trust-region-type method **CAT** has an optimal rate and shows competitiveness with state-of-the-art solvers [26], we adopt their experimental setup and compare with it, as well as the regularized Newton-type method **AN2CER** proposed by Gratton et al. [24] and the recently proposed adaptive cubic regularization method $\mathbf{ARC}_q\mathbf{K}$ [16]. The experiments are conducted on the 124 unconstrained problems with more than 100 variables from the widely used CUTEst benchmark for nonlinear optimization [22]. The algorithm is considered successful if it terminates with $\epsilon_k \leq \epsilon = 10^{-5}$ such that $k \leq 10^5$. If the algorithm fails to terminate within 5 hours, it is also recorded as a failure.

The detailed oracle evaluations and HVP computations are reported in Tables 3 and 5 in Section G, from which we observe that the fallback step has insignificant impact on performance yet increases computational cost, suggesting it can be relaxed or removed. Furthermore, $\theta \in [0.5, 1]$ balances computational efficiency and local behavior and a small $m_{\max}$ is preferable. Finally, the second linesearch step (2.5) and the TERM state of CappedCG are rarely taken in practice. Figure 2 shows our method without the fallback step. It is comparable to $\mathrm{ARC}_q\mathrm{K}$ and slightly faster than CAT and AN2CER, as each iteration uses only a few Hessian-vector products, whereas CAT relies on multiple Cholesky factorizations and AN2CER involves minimal eigenvalue computations. Meanwhile, our method requires a similar number of Hessian evaluations as CAT, and slightly fewer than AN2CER and $\mathrm{ARC}_q\mathrm{K}$. We also note that using a fixed $\omega_k = \sqrt{\epsilon}$ in Algorithm 1 may lead to failures when $g_k \gg \epsilon$, resulting in deteriorated performance. Additionally, our method requires significantly less

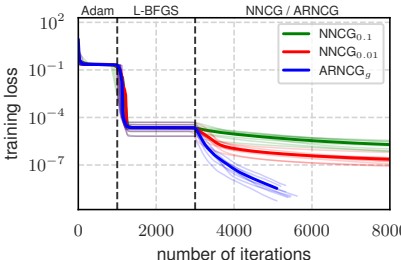

| | | Convection | Reaction | Wave |
|---|---|---|---|---|
| Training Loss | $NNCG_{0.1}$ | $1.15 \times 10^{-7}$ | $1.11 \times 10^{-7}$ | $9.23 \times 10^{-4}$ |
| | $NNCG_{0.01}$ | $1.38 \times 10^{-8}$ | $1.32 \times 10^{-8}$ | $8.31 \times 10^{-5}$ |
| | $ARNCG_g$ | $\mathbf{2.72 \times 10^{-11}}$ | $\mathbf{5.48 \times 10^{-10}}$ | $\mathbf{3.16 \times 10^{-6}}$ |
| Test L2RE | $NNCG_{0.1}$ | $5.63 \times 10^{-4}$ | $4.69 \times 10^{-3}$ | $5.14 \times 10^{-2}$ |
| | $NNCG_{0.01}$ | $2.27 \times 10^{-4}$ | $2.32 \times 10^{-3}$ | $1.31 \times 10^{-2}$ |
| | $ARNCG_g$ | $\mathbf{1.38 \times 10^{-5}}$ | $\mathbf{4.36 \times 10^{-4}}$ | $\mathbf{3.29 \times 10^{-3}}$ |
| Running Time Budget | | 7.5 hours | 2 hours | 18 hours |
| Peak GPU Memory | | 4.7GB | 3.3GB | 10.2GB |

Figure 3: Loss curves for training PINN on the reaction problem. Thin lines are 8 independent runs; the bold line shows the average. The subscript in NNCG denotes the regularization coefficient.

Table 2: Best training loss and test $\ell_2$ relative error (L2RE) on training PINNs over 8 runs. We terminate training based on a fixed time budget. The time limit is chosen such that $ARNCG_g$ performs approximately 2000 iterations. The peak memory usages of two methods are similar.

memory ($\sim$6GB) compared to CAT ($\sim$74GB) for the largest problem in the benchmark with 123200 variables, as it avoids constructing the full Hessian.

**Physics-informed neural networks**   Physics-Informed Neural Networks (PINNs) parameterize partial differential equations (PDEs) in physical problems using neural networks, and train the network by using the residuals of the equations as the loss function [46]. These PDEs often lead to a poor condition number for the PINN loss [47, 33, 50], making it difficult for first-order optimization methods like Adam to achieve high-precision solutions. To address this issue, a strategy is to use Adam first and then switch to quasi-Newton methods such as L-BFGS [47, 32]. However, Rathore et al. [47] observed that L-BFGS is still insufficient for effectively training PINNs. To address this, they proposed the **NNCG** method and further demonstrated that switching to NNCG after the L-BFGS phase can lead to additional loss reduction and improved solution quality. However, this approach relies on fixed a regularizer and does not fully resolve the challenges arising from the non-convexity of the objective function and still requires hyperparameter tuning for regularizers.

Our goal here is to demonstrate that is applicable to medium-size PINNs and offers improved stability and ease of use, owing to its globalization and adaptivity. The PINN used in our experiments consists of 81201 parameters in double precision. Since our method only relies on HVP, Table 2 shows the peak GPU memory usage is at most 10.2GB, whereas storing the full Hessian would require 49.1GB of memory. As shown in Figure 3, $ARNCG_g$ outperforms NNCG in both iteration complexity and runtime. Further details are provided in Section H.

## 5   Discussions

In this paper, we present the adaptive regularized Newton-CG method and show that two classes of regularizers achieve optimal global convergence order and quadratic local convergence. Our techniques in Section 3 can be extended to Riemannian optimization, as only Theorem C.1 needs to be modified. For the setting with Hölder continuous Hessians, a variant of this lemma can be derived following He et al. [29], and the subsequent proof may also be generalized (see Section F.2 for local rates). However, this case presents additional challenges since the Hölder exponent is also unknown and requires estimation. It would also be interesting to investigate whether these regularizers are suitable for the convex settings studied in Doikov and Nesterov [13], Doikov et al. [15] and whether they can be extended to inexact methods such as Yao et al. [54] and stochastic optimization.

## Acknowledgments and Disclosure of Funding

Y. Zhou and J. Zhu are supported by the National Natural Science Foundation of China (Nos. 62550004, 92270001), Tsinghua Institute for Guo Qiang, and the High Performance Computing Center, Tsinghua University; J. Zhu was also supported by the XPlorer Prize. C. Bao is supported by the National Key R&D Program of China (No. 2021YFA1001300) and the National Natural Science Foundation of China (No. 12271291). J. Xu is supported in part by PolyU postdoc matching fund scheme of the Hong Kong Polytechnic University (No. 1-W35A), and Huawei's Collaborative Grants

"Large scale linear programming solver" and "Solving large scale linear programming models for production planning". C. Ding is supported in part by the National Key R&D Program of China (No. 2021YFA1000300, No. 2021YFA1000301) and CAS Project for Young Scientists in Basic Research (No. YSBR-034).

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

# Contents

# A   Additional discussions

## A.1   Related work

**Relationship to previous RNMs**   Our primary contributions lie in the introduction of a new type of regularizers, as established in Theorem 2.2, which enable a smooth transition from the global to the local convergence phase, and in the design of a parameter-free algorithm built upon these regularizers. As illustrated in Table 1 and discussed in the introduction, the works most closely related to ours are Royer et al. [49] and Gratton et al. [24], and we elaborate on the differences below.

The basic framework of `NewtonStep` consists of three components: using capped conjugate gradient (CG) to compute the search direction, a linesearch procedure to determine the stepsize, and an estimation scheme for the Lipschitz constant. The first two components are similar to the capped CG framework proposed by Royer et al. [49], and a similar structure is adopted in He et al. [30] and Zhu and Xiao [56]. However, these methods are not adaptive and exhibit worse dependence on the Hessian Lipschitz constant $L_H$ in their iteration complexity. Specifically, Royer et al. [49] employs a cubic linesearch rule for both `SOL` and `NC` states, resulting in an $L_H^3$ dependence, while He et al. [30] improves this to $L_H^2$ by adopting a quadratic linesearch strategy. Subsequently, He et al. [29] proposed an adaptive algorithm that achieves the optimal $\sqrt{L_H}$ dependence in iteration complexity. However, it remains unclear whether the number of function evaluations arised from the linesearch procedure may grow unbounded as iterations proceed or how it depends on the tolerance parameter $\epsilon$, leaving the overall complexity in terms of second-order oracle calls unclear. In our analysis, we find that to ensure the number of function evaluations remains bounded, it is necessary to incorporate a secondary linesearch condition (2.5), enforce an upper bound $m_{\max}$ on the number of linesearch steps, and increase $M_k$ whenever this limit is reached. The combination of these algorithmic components guarantees that the total number of function evaluations remains bounded, and removing them results in $O(\log \frac{1}{\epsilon})$ function evaluations per iteration. Furthermore, the regularization strategies employed in this line of work are proportional to $\epsilon$, which requires a prescribed tolerance parameter $\epsilon$ and limits the local convergence rate to be merely linear.

Gradient-based regularizers in RMNs have also been studied by Gratton et al. [24]. Rather than employing the capped CG method, they explicitly test for negative curvature using the condition $\lambda_{\min}(\nabla^2 \varphi(x_k)) \lesssim -\sqrt{M_k g_k}$.[2] When this condition holds, the magnitude of the minimal eigenvalue is sufficiently large, the corresponding eigenvector is used directly to construct a sufficient descent direction. Otherwise, they switch to a regularizer of the form $\rho_k = \sqrt{M_k g_k} + [-\lambda_{\min}(\nabla^2 \varphi(x_k))]_+ \propto \sqrt{M_k g_k}$ and solve the regularized Newton equation to compute the direction. These two cases correspond conceptually to the `NC` and `SOL` states in the capped CG framework, respectively. Their method adopts a unit stepsize and uses an acceptance ratio to decide whether to accept the new iterate and how to update the Lipschitz estimate $M_k$, based on the accuracy of the local model. However, although their algorithm incorporates a mechanism for updating $M_k$, the dependence on $L_H$ in the complexity remains $\max(L_H^2, \sqrt{L_H})$, and it is unclear whether the optimal order $\sqrt{L_H}$ can be achieved. Furthermore, their regularizers are proportional to $\sqrt{g_k}$, which, as discussed in Section 2, limits the local convergence rate to $3/2$. On the other hand, incorporating the acceleration factor $\delta_k^\theta$ could potentially improve the local convergence rate of their methods, but this may lead to increased HVP complexity due to possible sharp drops in $g_k$. This issue is addressed in our method through the introduction of the `TERM` state in `CappedCG`.

Finally, we note that the proof in Gratton et al. [24] applies only to $g_k$-based regularizers instead of the $\epsilon_k$-based ones, while the proof in Royer et al. [49] is not applicable to $g_k$-based ones. Our partition (3.1) is new in RNMs and unifies the two regularizers into the same analysis.

**Other second-order methods with fast global rates**   The trust-region method is another important approach to globalizing the Newton method. By introducing a ball constraint $\|d\| \leq r_k$ to (1.1), it

---

[2]Gratton et al. [24] also proposed a variant that replace $\nabla^2 \varphi(x_k)$ with an approximation defined over the Krylov subspace to reduce computational cost. For notational simplicity, we focus on the full-space version in our discussion.

provides finer control over the descent direction. Several variants of this method have achieved optimal or near-optimal rates [10, 11, 9, 31]. For example, Curtis et al. [11], Jiang et al. [31] incorporated a Levenberg-Marquardt regularizer into the trust-region subproblem. Except for the regularization coefficient $\rho_k$, these regularized trust-region methods introduce an additional free parameter: the trust-region radius $r_k$, which provides extra flexibility to attain quadratic local convergence. For instance, regarding the local convergence rate, Jiang et al. [31, Tab. 2] first examine whether the smallest eigenvalue of the Hessian is sufficiently large. If this condition holds, they set $\rho_k = 0$ and employ the trust-region constraint purely as a globalization mechanism. Under a local convexity assumption, the method then effectively reduces to the classical Newton scheme in a neighborhood of the solution, thereby achieving quadratic convergence. However, this strategy requires an additional eigenvalue check and cannot be directly extended to RNMs, which rely only on the regularization parameter $\rho_k$. As a result, achieving a quadratic rate is more challenging for RNMs.

Hamad and Hinder [25, 26] introduced an elegant and powerful trust-region algorithm that does not modify the subproblem, achieving both an optimal global order and a quadratic local rate. In contrast, our results show that the RNM can also achieve both, while using less memory than Hamad and Hinder [26], as shown in Section 4. Interestingly, the disjunction of fast gradient decay and sufficient loss decay, as discussed above in the context of RNMs, is also reflected in several of these works. They also partition the iterations into failure and success sets, which leads to an additional logarithmic factor [31]. Our partition based on non-increasing subintervals of the gradient norm, as defined in (3.1), may also be used to improve this factor. It is worth noting that, previous to Royer et al. [49], a linesearch method with negative detection was proposed by Royer and Wright [48]. For convex problems, damped Newton methods achieving fast rates have also been developed [27, 28], and the method of Jiang et al. [31] can also be applied.

For adaptive cubic-regularization methods such as Dussault et al. [16], the core step consists of minimizing a cubically regularized subproblem, which can equivalently be interpreted as solving the regularized Newton equation $(\nabla^2 \varphi(x) + \rho I)d = -\nabla \varphi(x)$, where $\rho \propto \sigma \|d\|$ depends on the solution $d$. A central component of Dussault et al. [16] is the search for an appropriate value of $\rho$. In contrast, our method employs a prescribed regularization coefficient that is independent of $d$, thereby eliminating this search phase.

**Adaptive and universal algorithms** Since the introduction of cubic regularization, *adaptive* cubic regularization attaining the optimal rate without using the knowledge of problem parameters (i.e., the Lipschitz constant) were developed by Cartis et al. [6, 7], and *universal* algorithms based on this regularization that are applicable to different problem classes (e.g., functions with Hölder continuous Hessians with unknown Hölder exponents) are studied by Grapiglia and Nesterov [23], Doikov and Nesterov [13]. Additionally, some adaptive trust-region methods have also been introduced [31, 26]. Recently, several universal algorithms for RNMs have also been proposed, including those by He et al. [29], Doikov et al. [15], Gratton et al. [24]. As discussed earlier, Doikov et al. [15] focus on convex problems, the $L_H$ dependence in the complexity of Gratton et al. [24] is suboptimal, and the local convergence rate of He et al. [29] is not quadratic. Therefore, none of these methods achieve our goal of a parameter-free approach with both optimal global complexity and quadratic local convergence.

## A.2 Limitations

Despite achieving optimal global complexity and quadratic local convergence, as well as demonstrating competitiveness with other second-order methods in numerical experiments, our methods for training neural networks still require multiple CG iterations to find a descent direction. It would be desirable to develop better preconditioners or alternative strategies to further reduce the cost of each iteration. Furthermore, the Lipschitz continuity of the Hessian is assumed, which may not hold for nonsmooth activation functions such as ReLU.

## A.3 Broader impact

Our work primarily focuses on the theoretical properties of RNMs and proposes a new algorithm. We do not anticipate any potential negative societal impacts.

**Algorithm 2:** Capped conjugate gradient [49, Algorithm 1]

**Input** : A symmetric matrix $H \in \mathbb{R}^{d \times d}$, a vector $g \in \mathbb{R}^d$, a regularizer $\rho \in (0, \infty)$, a parameter $\bar{\rho} \in (0, \infty)$ used to decide whether to terminate the algorithm earlier, and a tolerance parameter $\xi \in (0, 1)$.

**Output** : $(\text{d\_type}, \tilde{d})$ such that $\text{d\_type} \in \{\texttt{NC}, \texttt{SOL}, \texttt{TERM}\}$ and Theorem B.2 holds.

1 **Subroutine** CappedCG($H, g, \rho, \xi, \bar{\rho}$)

2    $(y_0, r_0, p_0, j) \leftarrow (0, g, -g, 0)$

3    $\bar{H} \leftarrow H + 2\rho I_n$

4    $M \leftarrow \frac{\|Hp_0\|}{\|p_0\|}$

5    **if** $p_0^\top \bar{H} p_0 < \rho \|p_0\|^2$ **then return** $(\texttt{NC}, p_0)$

6    **while** *True* **do**

      // Beginning of standard CG

7       $\alpha_k \leftarrow \frac{\|r_k\|^2}{p_k^\top \bar{H} p_k}$

8       $y_{k+1} \leftarrow y_k + \alpha_k p_k$

9       $r_{k+1} \leftarrow r_k + \alpha_k \bar{H} p_k$

10      $\beta_{k+1} \leftarrow \frac{\|r_{k+1}\|^2}{\|r_k\|^2}$

11      $p_{k+1} \leftarrow -r_{k+1} + \beta_{k+1} p_k$

      // End of standard CG

12      $k \leftarrow k + 1$

13      $M \leftarrow \max\left(M, \frac{\|Hp_k\|}{\|p_k\|}, \frac{\|Hr_k\|}{\|r_k\|}, \frac{\|Hy_k\|}{\|y_k\|}\right)$          // Estimate the norm of $H$

14      $(\kappa, \hat{\xi}, \tau, T) \leftarrow \left(\frac{M+2\rho}{\rho}, \frac{\xi}{3\kappa}, \frac{\sqrt{\kappa}}{\sqrt{\kappa}+1}, \frac{4\kappa^4}{(1-\sqrt{\tau})^2}\right)$

15      **if** $y_k^\top \bar{H} y_k < \rho \|y_k\|^2$ **then return** $(\texttt{NC}, y_k)$

16      **else if** $\|r_k\| \leq \hat{\xi} \|r_0\|$ **then return** $(\texttt{SOL}, y_k)$

17      **else if** $p_k^\top \bar{H} p_k < \rho \|p_k\|^2$ **then return** $(\texttt{NC}, p_k)$

18      **else if** $\|r_k\| > \sqrt{T} \tau^{\frac{k}{2}} \|r_0\|$ **then**

19        $\alpha_k \leftarrow \frac{\|r_k\|^2}{p_k^\top \bar{H} p_k}$

20        $y_{k+1} \leftarrow y_k + \alpha_k p_k$

21        Find $i \in \{0, \ldots, k-1\}$ such that

$$\frac{(y_{k+1} - y_i)^\top \bar{H}(y_{k+1} - y_i)}{\|y_{k+1} - y_i\|^2} < \rho. \tag{B.1}$$

22        **return** $(\texttt{NC}, y_{k+1} - y_i)$

23      **else if** $k \geq J(M, \bar{\rho}, \xi) + 1$ **then**

24        **return** $(\texttt{TERM}, y_k)$          // $J(M, \bar{\rho}, \xi)$ is defined in (B.2)

# B   Details and properties of capped CG

The capped CG in Royer et al. [49] is presented in Algorithm 2, with an additional termination condition $k \geq J(M, \bar{\rho}, \xi) + 1$ and type TERM. Note that in Algorithm 1, we will take $\rho = \sqrt{M}\omega$. The following lemma states the number of iterations for the original version of capped CG.

**Lemma B.1** (Lemma 1 of Royer et al. [49]). *When the termination condition for TERM is removed, Algorithm 2 terminates in $\min(n, J(M, \rho, \xi)) + 1 \leq \min(n, \tilde{O}(\rho^{-\frac{1}{2}}))$ iterations, where*

$$J(M, \rho, \xi) = 1 + \left(\sqrt{\kappa} + \frac{1}{2}\right) \log\left(\frac{144 \left(\sqrt{\kappa} + 1\right)^2 \kappa^6}{\xi^2}\right), \quad \kappa = \frac{M + \rho}{\rho}. \tag{B.2}$$

The additional termination condition indicates that the regularizer $\rho$ may be too small to find a solution within the given computational budget.

For the oracle complexity, each iteration of Algorithm 2 requires only one Hessian-vector product, since the quantities $Hy_k$, $Hp_k$ and $Hr_k$ used in the negative curvature monitor can be recursively constructed from $\bar{H}p_k$ generated in the standard CG iteration. When the residual decays slower than expected, one more CG iteration is performed, and if the historical iterations are stored, only one additional Hessian-vector product is needed.

The properties of our version with the TERM state are summarized below.

**Lemma B.2.** *Invoking the subroutine $\mathtt{CappedCG}(H, g, \rho, \xi, \bar{\rho})$ obtains $(d\_type, \tilde{d})$, then we have the following properties.*

1. *When d_type = SOL, $\tilde{d}$ is an approximated solution of $(H + 2\rho I_n)\tilde{d} = -g$ such that*

   $$\tilde{d}^\top (H + 2\rho I_n)\tilde{d} \geq \rho\|\tilde{d}\|^2, \tag{B.3}$$

   $$\tilde{d}^\top H\tilde{d} \geq -\rho\|\tilde{d}\|^2, \tag{B.4}$$

   $$\|\tilde{d}\| \leq 2\rho^{-1}\|g\|, \tag{B.5}$$

   $$\|(H + 2\rho I_n)\tilde{d} + g\| \leq \frac{1}{2}\rho\xi\|\tilde{d}\| \leq \xi\|g\|, \tag{B.6}$$

   $$\tilde{d}^\top g = -\tilde{d}^\top (H + 2\rho I_n)\tilde{d} \leq -\rho\|\tilde{d}\|^2. \tag{B.7}$$

2. *When d_type = NC, $\tilde{d}$ is a negative curvature direction such that*

   $$\tilde{d}^\top H\tilde{d} \leq -\rho\|\tilde{d}\|^2. \tag{B.8}$$

3. *When d_type = TERM, then $\rho < \bar{\rho}$. In other words, if $\bar{\rho} \leq \rho$ the algorithm terminates with d_type $\in \{SOL, NC\}$.*

4. *Suppose there exist $\alpha, a, b > 0$ such that $H \succeq \alpha I_n$, $\bar{\rho} \leq b\rho^a$ and $\rho \leq 1$, then the algorithm terminates with d_type = SOL when $\xi = \rho \leq C(\alpha, a, b, \|H\|)$, where*

   $$C(\alpha, a, b, U) := \min\left(1, \left(\frac{\alpha^2}{bU}\right)^{\frac{1}{a}}, \left(\frac{24\alpha^7}{b^7\sqrt{U}(U+2)}\right)^{\frac{1}{7a}}, \left(\frac{12\alpha^7}{b^7}\right)^{\frac{1}{7a+2}}\right).$$

*Proof.* The first two cases directly follow from Royer et al. [49, Lemma 3].[3] The third case follows from Theorem B.1 and the monotonic non-increasing property of the map $\rho \mapsto J(M, \rho, \xi)$.

The fourth case follows from the standard property of CG for positive definite equation, since $H \succeq \alpha I_n$ the capped CG reduces to the standard CG. Specifically, let $\{y_k, r_k\}_{k \geq 0}$ be the sequence generated by Algorithm 2, then Nocedal and Wright [44, Equation (5.36)] gives that

$$\|e_k\|_{\bar{H}} \leq 2\left(\frac{\sqrt{\kappa(\bar{H})} - 1}{\sqrt{\kappa(\bar{H})} + 1}\right)^k \|e_0\|_{\bar{H}} \leq 2\exp\left(\frac{-2k}{\sqrt{\kappa(\bar{H})}}\right) \|e_0\|_{\bar{H}},$$

---

[3]This lemma assumes that $H = \nabla^2\varphi(x)$, $g = \nabla\varphi(x)$, and $\varphi$ has Lipschitz Hessian. However, the statement of this lemma and the capped CG involve only the Hessian of $\varphi$ at a single point $x$, and hence the assumption can be removed.

where $\|e_k\|_{\bar{H}}^2 := e_k^\top \bar{H} e_k$ and $\kappa(\bar{H}) = (\alpha + 2\rho)^{-1}(\|H\| + 2\rho)$ is the condition number, and $e_k = y_k + \bar{H}^{-1}g = \bar{H}^{-1}r_k$ and $\bar{H} = H + 2\rho I_n$. Then, the above display becomes

$$\frac{1}{\|H\| + 2\rho}\|r_k\|^2 \leq r_k^\top \bar{H}^{-1} r_k \leq 4 \exp\left(\frac{-4k}{\sqrt{\kappa(\bar{H})}}\right) r_0^\top \bar{H}^{-1} r_0$$

$$\leq 4 \exp\left(\frac{-4k}{\sqrt{\kappa(\bar{H})}}\right) \frac{1}{\alpha + 2\rho}\|r_0\|^2.$$

Let $M, \kappa, \hat{\xi}$ be the quantities updated in the algorithm. Then, we have $M \geq \alpha$ and $\kappa \leq \rho^{-1}\|H\| + 2$ and $\hat{\xi} = \frac{\xi}{3\kappa} \geq \frac{\xi}{3\rho^{-1}\|H\|+6}$. Hence, when the TERM state is removed, and suppose Algorithm 2 terminates at $k_*$-th step with SOL. Then, we have

$$k_* \leq \left\lceil \frac{1}{2}\sqrt{\kappa(\bar{H})} \log \frac{6\sqrt{\kappa(\bar{H})}(\rho^{-1}\|H\| + 2)}{\xi} \right\rceil. \tag{B.9}$$

Since $\kappa(\bar{H}) \leq \frac{\|H\|}{\alpha}$ and $\rho \leq 1$, we know

$$k_* \leq \frac{1}{2}\sqrt{\frac{\|H\|}{\alpha}} \log \frac{6\sqrt{\|H\|}(\|H\| + 2)}{\sqrt{\alpha}\rho\xi} + 1 =: K(\rho, \xi).$$

When incorporating the TERM state, and suppose it is triggered at the $\hat{k}$-th step, then

$$K(\rho, \xi) \geq k_* > \hat{k} \geq J(M, \bar{\rho}, \xi) + 1 \geq J(M, \bar{\rho}, \xi). \tag{B.10}$$

However, when $b\rho^a \geq \bar{\rho}$, we have

$$J(M, \bar{\rho}, \xi) \geq J(\alpha, \bar{\rho}, \xi) \geq J(\alpha, b\rho^a, \xi) \geq \sqrt{\frac{\alpha}{b\rho^a}} \log \frac{144\alpha^7}{\xi^2 b^7 \rho^{7a}}.$$

Hence, when $\xi = \rho \leq C(\alpha, a, b, \|H\|)$, we have $\frac{\alpha}{b\rho^a} \geq \frac{\|H\|}{\alpha} \geq 1$ and $\frac{144\alpha^7}{b^7\rho^{7a+2}} \geq \frac{6\sqrt{\|H\|}(\|H\|+2)}{\sqrt{\alpha}\rho^2}$ and $\frac{144\alpha^7}{b^7\rho^{7a+2}} \geq 12$. Then,

$$0 \overset{(B.10)}{\geq} J(M, \bar{\rho}, \rho) - K(\rho, \rho)$$

$$\geq \sqrt{\frac{\alpha}{b\rho^a}} \log \frac{144\alpha^7}{b^7\rho^{7a+2}} - \frac{1}{2}\sqrt{\frac{\|H\|}{\alpha}} \log \frac{6\sqrt{\|H\|}(\|H\| + 2)}{\sqrt{\alpha}\rho^2} - 1$$

$$\geq \frac{1}{2}\sqrt{\frac{\|H\|}{\alpha}} \log \frac{144\alpha^7}{b^7\rho^{7a+2}} - 1 \geq \frac{\log 12}{2} - 1 > 0,$$

which leads to a contradiction. Therefore, the algorithm will terminate with SOL. □

# C   Main results for global rates

Throughout this section, we follow the partition (3.1) defined in Section 3 and provide detailed proofs for the global rates in Theorem 2.2 and corresponding lemmas described in Section 3. For the sake of readability, we restate the lemmas mentioned in Section 3.

## C.1   Details in Section 3

As discussed at the beginning of Section 3, the following lemma summarizes key properties of Algorithm 1 and plays a central role in estimating the number of iterations that yield sufficient descent. Its proof is technically involved and is deferred to Section D.1.

**Lemma C.1** (Summarized descent lemma, proof in Section D.2). *Let* $\{x_k, M_k, d\_type_k, m_k\}_{k\geq 0}$ *be the sequence generated by Algorithm 1, and denote* $\omega_k := \omega_k^t$ *if the trial step is accepted and* $\omega_k := \omega_k^f$ *otherwise. Define the index sets* $\mathcal{J}^i = \{k : M_{k+1} = \gamma^i M_k\}$ *for* $i = -1, 0, 1$, *and the constants* $\tilde{C}_4 = \max\left(1, \tau_-^{-1}(9\beta)^{-\frac{1}{2}}, \tau_-^{-1}(3\beta(1 - 2\mu))^{-1}\right)$ *and* $\tilde{C}_5 = \min(2, 3 - 6\mu)^{-1}$, *then*

1. *If $k \in \mathcal{J}^1$, then $M_k \leq \tilde{C}_5 L_H$;*

2. *For the regularizers in Theorem 2.2, if $M_k > \tilde{C}_4 L_H$ and $\tau_- \leq \min\left(\delta_k^\alpha, \delta_{k+1}^\alpha\right)$, then $k \in \mathcal{J}^{-1}$, where $\alpha = \max(2, 3\theta)$.*

*Moreover, we have $\bigcup_{i=-1,0,1}(\mathcal{J}^i \cap I_{0,k}) = I_{0,k}$, and*

$$|\mathcal{J}^1 \cap I_{0,k}| \leq |\mathcal{J}^{-1} \cap I_{0,k}| + [\log_\gamma(\gamma \tilde{C}_5 M_0^{-1} L_H)]_+, \tag{C.1}$$

$$k = |I_{0,k}| \leq 2|\mathcal{J}^{-1} \cap I_{0,k}| + |\mathcal{J}^0 \cap I_{0,k}| + [\log_\gamma(\gamma \tilde{C}_5 M_0^{-1} L_H)]_+, \tag{C.2}$$

*and the following descent inequality holds:*

$$\varphi(x_{k+1}) - \varphi(x_k) \leq \begin{cases} 0, & \text{if } k \in \mathcal{J}^1, \\ -\tilde{C}_1 M_k^{-\frac{1}{2}} D_k, & \text{if } k \in \mathcal{J}^0 \cup \mathcal{J}^{-1}, \end{cases} \tag{C.3}$$

*where $\tilde{C}_1 = \min\left(9\beta^2(1-2\mu)^2\mu, 36\beta\mu(1-\mu)^2, 4\mu/33\right)$, and*

$$D_k = \begin{cases} (\omega_k^{\mathrm{f}})^3, & \text{if } k \in \mathcal{J}^{-1}, \\ \min\left((\omega_k^{\mathrm{f}})^3, \omega_k^3, g_{k+1}^2 \omega_k^{-1}\right), & \text{if } d\_type_k = \mathtt{SOL} \text{ and } m_k = 0 \text{ and } k \notin \mathcal{J}^{-1}, \\ \min\left((\omega_k^{\mathrm{f}})^3, \omega_k^3\right), & \text{otherwise.} \end{cases} \tag{C.4}$$

**Lemma C.2** (Restatement of Theorem 3.1). *Under the regularizer choices of Theorem 2.2, we have $\omega_{\ell_j-1} = \omega_{\ell_j-1}^{\mathrm{f}}$ for each $j \geq 2$, and*

$$\varphi(x_{\ell_j}) - \varphi(x_{\ell_j-1}) \leq -\tilde{C}_1 M_{\ell_j-1}^{-\frac{1}{2}} \mathbf{1}_{\{\ell_j-1 \notin \mathcal{J}^1\}}(\omega_{\ell_j-1}^{\mathrm{f}})^3, \tag{C.5}$$

*where $\tilde{C}_1, \tilde{C}_4$ are defined in Theorem C.1. Moreover, if $M_{\ell_j-1} > \tilde{C}_4 L_H$, then $\ell_j - 1 \in \mathcal{J}^{-1}$.*

*Proof.* Let $k = \ell_j - 1$. If the fallback step is taken, then $\omega_k = \omega_k^{\mathrm{f}}$ holds. We consider the case where the trial step at $k$-th iteration is accepted, then we know $g_{k+\frac{1}{2}} = g_{k+1} > g_k$ by the partition rule (3.1). However, the acceptance rule of the trial step in Algorithm 1 gives that $g_k > g_{k-1}$, and hence $\min(1, g_k^\theta g_{k-1}^{-\theta}) = 1$. Moreover, we have $g_{k-1} \geq \epsilon_{k-1}$ and then

$$\epsilon_k = \min(\epsilon_{k-1}, g_k) \geq \min(\epsilon_{k-1}, g_{k-1}) = \epsilon_{k-1} \geq \epsilon_k.$$

Therefore, $\epsilon_k^\theta \epsilon_{k-1}^{-\theta} = 1$. Combining these discussions, we know $\omega_k = \omega_k^{\mathrm{f}}$ for the two choices of regularizers.

It remains to show that $D_k \geq (\omega_k^{\mathrm{f}})^3$ for $D_k$ defined in Theorem C.1, which holds since we know $g_{k+1} > g_k$ by the partition rule (3.1), and $g_k \geq (\omega_k^{\mathrm{f}})^2$ by the choice of regularizers, and therefore,

$$D_k \overset{(C.4)}{\geq} \min((\omega_k^{\mathrm{f}})^3, g_{k+1}^2(\omega_k^{\mathrm{f}})^{-1}) \geq \min((\omega_k^{\mathrm{f}})^3, g_k^2(\omega_k^{\mathrm{f}})^{-1}) \geq (\omega_k^{\mathrm{f}})^3. \tag{C.6}$$

Finally, when $M_k > \tilde{C}_4 L_H$, we use Theorem D.5 to show that $k \in \mathcal{J}^{-1}$. For the first case in that corollary, since $\tau_- < 1$, then $\omega_k = \omega_k^{\mathrm{f}} > \tau_- \omega_k^{\mathrm{f}}$, then the corollary gives $k \in \mathcal{J}^{-1}$. For the second case, the results follows from (C.6) and $\min(\omega_k^3, g_{k+1}^2 \omega_k^{-1}) \geq (\omega_k^{\mathrm{f}})^3 > \tau_-(\omega_k^{\mathrm{f}})^3$. $\qquad\square$

**Lemma C.3** (Restatement of Theorem 3.2). *Under the regularizer choices of Theorem 2.2, we have $(\omega_k^{\mathrm{f}})^{1+2\theta}(\omega_{k-1}^{\mathrm{f}})^{-2\theta} \leq \omega_k \leq \omega_k^{\mathrm{f}}$ for each $k \geq 1$. Moreover, for $j \geq 1$ and $\ell_j < k < \ell_{j+1}$,*

$$\varphi(x_k) - \varphi(x_{\ell_j}) \leq -C_{\ell_j,k}\left(|I_{\ell_j,k} \cap \mathcal{J}^{-1}| + \max\left(0, |I_{\ell_j,k} \cap \mathcal{J}^0| - T_{\ell_j,k} - 5\right)\right)(\omega_k^{\mathrm{f}})^3, \tag{C.7}$$

*where $C_{i,j} = \tilde{C}_1 \min_{i \leq l < j} M_l^{-\frac{1}{2}}$, $T_{i,j} = 2\log\log\left(3(\omega_i^{\mathrm{f}})^2(\omega_j^{\mathrm{f}})^{-2}\right)$, and $\tilde{C}_1$ is defined in Theorem C.1.*

*Proof.* Under the regularizers choices, we know for each $k \in \mathbb{N}$, $D_k$ defined in (C.4) satisfies that

$$D_k \geq \min\left((\omega_k^{\mathrm{f}})^3, g_{k+1}^2 \omega_k^{-1}, \omega_k^3\right) = \min\left(g_{k+1}^2 \omega_k^{-1}, \omega_k^3\right)$$
$$\geq \min\left(g_{k+1}^2(\omega_k^{\mathrm{f}})^{-1}, (\omega_k^{\mathrm{f}})^{3+6\theta}(\omega_{k-1}^{\mathrm{f}})^{-6\theta}\right). \tag{C.8}$$

**Case 1** For the first choice of regularizers, we have $\omega_i^{\mathrm{f}} = \sqrt{g_i}$ and $T_{i,j} = 2\log\log\frac{3g_i}{g_j}$, and

$$\varphi(x_{i+1}) - \varphi(x_i) \overset{(C.3)}{\leq} \begin{cases} -C_i \min\left(g_{i+1}^2 g_i^{-\frac{1}{2}}, g_i^{\frac{3}{2}+3\theta} g_{i-1}^{-3\theta}\right), & \text{if } i \notin \mathcal{J}^{-1}, \\ -C_i g_i^{\frac{3}{2}}, & \text{if } i \in \mathcal{J}^{-1}, \end{cases}$$

where $C_i := \tilde{C}_1 M_i^{-\frac{1}{2}}$.

When $\theta > 0$, for any $\ell_j < k \leq \ell_{j+1} - 1$, using Theorem D.9 with

$$(p_1, q_1, p_2, q_2, a, A, K, S) = \left(2, \frac{1}{2}, \frac{3}{2}+3\theta, 3\theta, g_k, g_{\ell_j}, k - \ell_j - 1, I_{\ell_j,k} \cap \mathcal{J}^0\right), \tag{C.9}$$

we see that

$$\varphi(x_k) - \varphi(x_{\ell_j}) \overset{(C.3)}{\leq} -\tilde{C}_1 \sum_{\substack{\ell_j \leq i < k \\ i \in \mathcal{J}^{-1}}} M_i^{-\frac{1}{2}} g_i^{\frac{3}{2}} - \tilde{C}_1 \sum_{\substack{\ell_j \leq i < k \\ i \in \mathcal{J}^0}} M_i^{-\frac{1}{2}} \min\left(g_{i+1}^2 g_i^{-\frac{1}{2}}, g_i^{\frac{3}{2}+3\theta} g_{i-1}^{-3\theta}\right)$$

$$\leq -C_{\ell_j,k} \sum_{\substack{\ell_j \leq i < k \\ i \in \mathcal{J}^{-1}}} g_i^{\frac{3}{2}} - C_{\ell_j,k} \sum_{\substack{\ell_j \leq i < k \\ i \in \mathcal{J}^0}} \min\left(g_{i+1}^2 g_i^{-\frac{1}{2}}, g_i^{\frac{3}{2}+3\theta} g_{i-1}^{-3\theta}\right)$$

$$\overset{(D.29)}{\leq} -C_{\ell_j,k}\left(|I_{\ell_j,k} \cap \mathcal{J}^{-1}| + \max\left(0, |I_{\ell_j,k} \cap \mathcal{J}^0| - T_{\ell_j,k} - 5\right)\right) g_k^{\frac{3}{2}}. \tag{C.10}$$

On the other hand, when $\theta = 0$, we know $\varphi(x_{i+1}) - \varphi(x_i) \leq -C_i g_{i+1}^2 g_i^{-\frac{1}{2}}$ for $i \notin \mathcal{J}^{-1}$, and (C.10) also holds by applying Theorem D.8 with

$$(p, q, a, A, K, S) = \left(2, \frac{1}{2}, g_k, g_{\ell_j}, k - \ell_j - 1, I_{\ell_j,k} \cap \mathcal{J}^0\right).$$

**Case 2** For the second choice of the regularizers, we have $\omega_i^{\mathrm{f}} = \sqrt{\epsilon_i}$ and $T_{i,j} = 2\log\log\frac{3\epsilon_i}{\epsilon_j}$.

Since $\epsilon_k$ is non-increasing and $\omega_k \leq \sqrt{\epsilon_k}$ for each $k \in \mathbb{N}$, then for a fixed $i$ such that $\ell_j \leq i < \ell_{j+1} - 1$, we know $g_i \geq g_{i+1}$ and have the following two cases.

1. If $g_{i+1} \geq \epsilon_{i-1}$, we know $\epsilon_i = \min(\epsilon_{i-1}, g_i) \geq \min(\epsilon_{i-1}, g_{i+1}) = \epsilon_{i-1} \geq \epsilon_i$. Then,

$$D_i \overset{(C.8)}{\geq} \min\left(g_{i+1}^2 \epsilon_i^{-\frac{1}{2}}, \epsilon_i^{\frac{3}{2}+3\theta} \epsilon_{i-1}^{-3\theta}\right) \overset{(g_{i+1} \geq \epsilon_{i-1})}{\geq} \min\left(\epsilon_{i-1}^2 \epsilon_i^{-\frac{1}{2}}, \epsilon_i^{\frac{3}{2}+3\theta} \epsilon_{i-1}^{-3\theta}\right) \overset{(\epsilon_i = \epsilon_{i-1})}{=} \epsilon_i^{\frac{3}{2}}.$$

2. If $g_{i+1} < \epsilon_{i-1}$, then using $g_{i+1} \geq \min(g_{i+1}, \epsilon_i) = \epsilon_{i+1}$, we have

$$D_i \overset{(C.8)}{\geq} \min\left(g_{i+1}^2 \epsilon_i^{-\frac{1}{2}}, \epsilon_i^{\frac{3}{2}+3\theta} \epsilon_{i-1}^{-3\theta}\right) \overset{(g_{i+1} \geq \epsilon_{i+1})}{\geq} \min\left(\epsilon_{i+1}^2 \epsilon_i^{-\frac{1}{2}}, \epsilon_i^{\frac{3}{2}+3\theta} \epsilon_{i-1}^{-3\theta}\right).$$

Thus, from Theorem C.1, we know for $\ell_j \leq i < \ell_{j+1} - 1$, it holds that

$$\varphi(x_{i+1}) - \varphi(x_i) \overset{(C.3)}{\leq} \begin{cases} -C_i \min\left(\epsilon_{i+1}^2 \epsilon_i^{-\frac{1}{2}}, \epsilon_i^{\frac{3}{2}+3\theta} \epsilon_{i-1}^{-3\theta}\right), & \text{if } i \notin \mathcal{J}^{-1} \text{ and } g_{i+1} < \epsilon_{i-1}, \\ -C_i \epsilon_i^{\frac{3}{2}}, & \text{if } i \in \mathcal{J}^{-1} \text{ or } g_{i+1} \geq \epsilon_{i-1}. \end{cases}$$

Define $\mathcal{J}_+^0 = \mathcal{J}^0 \cap \{i : g_{i+1} \geq \epsilon_{i-1}\}$ and $\mathcal{J}_-^0 = \mathcal{J}^0 \setminus \mathcal{J}_+^0$. For any $\ell_j < k \leq \ell_{j+1} - 1$ and $\theta > 0$, we can apply Theorem D.9, with the parameters $a, A$, and $\{g_i\}_{0 \leq i \leq K+1}$ therein chosen as $\epsilon_k, \epsilon_{\ell_j}$, and $\{\epsilon_i\}_{\ell_j \leq i \leq k}$, respectively, and other parameter choices remain the same as (C.9). Then, we know

$$\varphi(x_k) - \varphi(x_{\ell_j}) \overset{(C.3)}{\leq} -C_{\ell_j,k} \sum_{\substack{\ell_j \leq i < k \\ i \in \mathcal{J}^{-1} \cup \mathcal{J}_+^0}} \epsilon_i^{\frac{3}{2}} - C_{\ell_j,k} \sum_{\substack{\ell_j \leq i < k \\ i \in \mathcal{J}_-^0}} \min\left(\epsilon_{i+1}^2 \epsilon_i^{-\frac{1}{2}}, \epsilon_i^{\frac{3}{2}+3\theta} \epsilon_{i-1}^{-3\theta}\right)$$

$$\overset{(D.27)}{\leq} -C_{\ell_j,k}\left(|I_{\ell_j,k} \cap (\mathcal{J}^{-1} \cup \mathcal{J}_+^0)| + \max\left(0, |I_{\ell_j,k} \cap \mathcal{J}_-^0| - T_{\ell_j,k} - 5\right)\right) \epsilon_k^{\frac{3}{2}}$$

$$= -C_{\ell_j,k}\left(|I_{\ell_j,k} \cap \mathcal{J}^{-1}| + \max\left(|I_{\ell_j,k} \cap \mathcal{J}_+^0|, |I_{\ell_j,k} \cap \mathcal{J}^0| - T_{\ell_j,k} - 5\right)\right) \epsilon_k^{\frac{3}{2}}$$

$$\leq -C_{\ell_j,k}\left(|I_{\ell_j,k} \cap \mathcal{J}^{-1}| + \max\left(0, |I_{\ell_j,k} \cap \mathcal{J}^0| - T_{\ell_j,k} - 5\right)\right) \epsilon_k^{\frac{3}{2}}. \tag{C.11}$$

Similarly, when $\theta = 0$ we can invoke Theorem D.8 to obtain the same result. $\qquad\square$

**Proposition C.4** (Restatement of Theorem 3.3). *Under the regularizer choices of Theorem 2.2, for each $k \geq 0$, we have*

$$\varphi(x_k) - \varphi(x_0) \leq -C_{0,k} \bigg( \underbrace{|I_{0,k} \cap \mathcal{J}^{-1}| + \max\big(|S_k \cap \mathcal{J}^0|, |I_{0,k} \cap \mathcal{J}^0| - V_k - 5J_k\big)}_{\Sigma_k} \bigg) \epsilon_k^{\frac{3}{2}},$$

(C.12)

*where $C_{0,k}$ is defined in Theorem C.3, and $V_k = \sum_{j=1}^{J_k-1} T_{\ell_j,\ell_{j+1}} + T_{\ell_{J_k},k}$, and $S_k = \bigcup_{j=1}^{J_k-1} \{\ell_{j+1} - 1\}$, and $J_k = \max\{j : \ell_j \leq k\}$.*

*Proof.* For each $j \geq 0$ such that $\ell_{j+1} - \ell_j \geq 2$, using (C.7) with $k = \ell_{j+1} - 1$ and (C.5), and $\mathbf{1}_{\{k \notin \mathcal{J}^1\}} = \mathbf{1}_{\{k \in \mathcal{J}^{-1}\}} + \mathbf{1}_{\{k \in \mathcal{J}^0\}}$, we find

$$\varphi(x_{\ell_{j+1}}) - \varphi(x_{\ell_j}) = \big(\varphi(x_{\ell_{j+1}}) - \varphi(x_{\ell_{j+1}-1})\big) + \big(\varphi(x_{\ell_{j+1}-1}) - \varphi(x_{\ell_{j+1}})\big)$$

$$\leq -C_{\ell_j,\ell_{j+1}} \big(|I_{\ell_j,\ell_{j+1}} \cap \mathcal{J}^{-1}| + \max\big(\mathbf{1}_{\{\ell_{j+1}-1 \in \mathcal{J}^0\}}, |I_{\ell_j,\ell_{j+1}} \cap \mathcal{J}^0| - T_j - 5\big)\big) (\omega_{\ell_{j+1}-1}^{\mathrm{f}})^3,$$

where $T_j := T_{\ell_j,\ell_{j+1}}$ and $I_{i,j}, T_{i,j}, C_{i,j}$ are defined in Theorem C.3. On the other hand, when $\ell_{j+1} - \ell_j = 1$, then the above inequality also holds since it reduces to (C.5).

Define $J_k = \max\{j : \ell_j \leq k\}$, then $\ell_{J_k} \leq k < \ell_{J_k+1}$, and the following inequality holds by noticing that for each $i \in \mathbb{N}$, either $\omega_i^{\mathrm{f}} = \sqrt{\epsilon_i}$ or $\omega_i^{\mathrm{f}} = \sqrt{g_i} \geq \sqrt{\epsilon_i}$.

$$\varphi(x_k) - \varphi(x_0) = \varphi(x_k) - \varphi(x_{\ell_{J_k}}) + \sum_{j=1}^{J_k-1} \big(\varphi(x_{\ell_{j+1}}) - \varphi(x_{\ell_j})\big)$$

$$\leq -C_{\ell_{J_k},k} \left( |I_{\ell_{J_k},k} \cap \mathcal{J}^{-1}| + \max\left(0, |I_{\ell_{J_k},k} \cap \mathcal{J}^0| - T_{\ell_{J_k},k} - 5\right) \right) \epsilon_k^{\frac{3}{2}}$$

$$- \sum_{j=1}^{J_k-1} C_{\ell_j,\ell_{j+1}} \left( |I_{\ell_j,\ell_{j+1}} \cap \mathcal{J}^{-1}| + \max\left(\mathbf{1}_{\{\ell_{j+1}-1 \in \mathcal{J}^0\}}, |I_{\ell_j,\ell_{j+1}} \cap \mathcal{J}^0| - T_j - 5\right) \right) \epsilon_{\ell_{j+1}-1}^{\frac{3}{2}}$$

$$\leq -C_{0,k} \epsilon_k^{\frac{3}{2}} \left( |I_{0,k} \cap \mathcal{J}^{-1}| + \max\left(|S_k \cap \mathcal{J}^0|, |I_{0,k} \cap \mathcal{J}^0| - V_k - 5J_k\right) \right), \tag{C.13}$$

where $V_k = \sum_{j=1}^{J_k-1} T_j + T_{\ell_{J_k},k}$, $S_k = \bigcup_{j=1}^{J_k-1} \{\ell_{j+1} - 1\}$ and the last inequality follows from $\max(a,b) + \max(c,d) \geq \max(a+c, b+d)$. □

**Proposition C.5** (Restatement of Theorem 3.5). *Let $k_{\mathrm{init}} = \min\{j : M_j \leq \tilde{C}_4 L_H\}$ if $M_0 > \tilde{C}_4 L_H$ and $k_{\mathrm{init}} = 0$ otherwise, then for the first choice of regularizers in Theorem 2.2, we have*

$$k_{\mathrm{init}} \leq \left\lceil \log_\gamma \frac{\gamma M_0}{\tilde{C}_4 L_H} \right\rceil_+ \left( \tilde{C}_3 \log \frac{U_\varphi}{\epsilon_{k_{\mathrm{init}}}} + 3 \right) + 2, \tag{C.14}$$

*where $\tilde{C}_3^{-1} = \frac{1}{2\max(2,3\theta)} \log \frac{1}{\tau_-} > 0$ and $\tilde{C}_4$ is defined in Theorem C.1, and $[x]_+$ denotes $\max(0, x)$. For the second choice of regularizers, we have*

$$k_{\mathrm{init}} \leq \left\lceil \log_\gamma \frac{M_0}{\tilde{C}_4 L_H} \right\rceil_+ + \tilde{C}_3 \log \frac{U_\varphi}{\epsilon_{k_{\mathrm{init}}}} + 2. \tag{C.15}$$

*Proof.* Using Theorem C.1 and observing that the constants therein satisfy $\tilde{C}_4 \geq \tilde{C}_5$, then we know $M_k$ is non-increasing for $k < k_{\mathrm{init}}$. Hence, $\tilde{C}_4 L_H < M_k = M_0 \gamma^{-|I_{0,k} \cap \mathcal{J}^{-1}|}$, and equivalently,

$$\log_\gamma(\tilde{C}_4 L_H) < \log_\gamma M_k = \log_\gamma M_0 - |I_{0,k} \cap \mathcal{J}^{-1}|. \tag{C.16}$$

By definition of $\delta_k$ in Theorem 2.2, we know $\delta_k^\theta = \omega_k^{\mathrm{t}}(\omega_k^{\mathrm{f}})^{-1} \leq 1$. Let $\mathcal{I}_{i,j} = \{l \in I_{i,j} : \delta_l^\alpha < \tau_-\}$, and $\mathcal{I}_{i,j}^+ = \{l \in I_{i,j} : \delta_{l+1}^\alpha < \tau_-\}$. From Theorem C.1, when $M_k > \tilde{C}_4 L_H$ and $\tau_- \leq \min\left(\delta_k^\alpha, \delta_{k+1}^\alpha\right)$, we have $k \in \mathcal{J}^{-1}$. Equivalently, we have $(I_{i,j} \backslash \mathcal{I}_{i,j}) \cap (I_{i,j} \backslash \mathcal{I}_{i,j}^+) \subseteq I_{i,j} \cap \mathcal{J}^{-1}$ for $i < j < k_{\mathrm{init}}$. Then,

$$|I_{i,j} \cap \mathcal{J}^{-1}| \geq |(I_{i,j} \backslash \mathcal{I}_{i,j}) \cap (I_{i,j} \backslash \mathcal{I}_{i,j}^+)| = |I_{i,j} \backslash (\mathcal{I}_{i,j} \cup \mathcal{I}_{i,j}^+)|$$

$$\geq |I_{i,j}| - (|\mathcal{I}_{i,j}| + |\mathcal{I}_{i,j}^+|) \geq |I_{i,j}| - 2|\mathcal{I}_{i-1,j}^+|, \tag{C.17}$$

where the last inequality follows from $\mathcal{I}_{i,j} = \mathcal{I}_{i-1,j-1}^+ \subseteq \mathcal{I}_{i-1,j}^+$. Reformulating (C.17) obtains

$$|\mathcal{I}_{i,j+1}^+| \geq \frac{1}{2} \left(|I_{i+1,j+1}| - |I_{i+1,j+1} \cap \mathcal{J}^{-1}|\right), \forall 0 \leq i < j < k_{\mathrm{init}} - 1. \tag{C.18}$$

**Case 1** We consider the first choice of regularizers, i.e., $\delta_k = \min(1, g_k g_{k-1}^{-1})$. Following the partition (3.1), for any $\ell_j \leq l < \ell_{j+1} - 1$ and $l < k_{\mathrm{init}} - 1$, we know $g_{l+1} \leq g_l$ and $\delta_{l+1} = g_{l+1} g_l^{-1}$. Therefore, since $\log \delta_{l+1} \leq 0$ and $\log \tau_- < 0$, it holds that

$$\log \frac{g_{l+1}}{g_{\ell_j}} = \sum_{\ell_j \leq i \leq l} \log \delta_{i+1} \leq \sum_{i \in \mathcal{I}^+_{\ell_j, l+1}} \log \delta_{i+1}$$

$$< \frac{\log \tau_-}{\alpha} |\mathcal{I}^+_{\ell_j, l+1}| \overset{(\mathrm{C}.18)}{\leq} -A(|I_{\ell_j+1, l+1}| - |I_{\ell_j+1, l+1} \cap \mathcal{J}^{-1}|), \qquad (\mathrm{C}.19)$$

where $A = \frac{1}{2\alpha} \log \frac{1}{\tau_-} > 0$. Let $k < k_{\mathrm{init}} - 1$ and $\hat{J}_k = \max\{j : \ell_j \leq k + 1\}$, then

$$\hat{J}_k \log \frac{\epsilon_{k+1}}{U_\varphi} \leq \sum_{j=1}^{\hat{J}_k - 1} \log \frac{g_{\ell_{j+1}-1}}{g_{\ell_j}} + \log \frac{g_{k+1}}{g_{\ell_{\hat{J}_k}}}$$

$$\overset{(\mathrm{C}.19)}{\leq} -A \sum_{j=1}^{\hat{J}_k - 1} (|I_{\ell_j+1, \ell_{j+1}-1}| - |I_{\ell_j+1, \ell_{j+1}-1} \cap \mathcal{J}^{-1}|)$$

$$\quad - A(|I_{\ell_{\hat{J}_k}+1, k+1}| - |I_{\ell_{\hat{J}_k}+1, k+1} \cap \mathcal{J}^{-1}|)$$

$$\leq -A(|I_{1, k+1}| - 2\hat{J}_k - |I_{1, k+1} \cap \mathcal{J}^{-1}|), \qquad (\mathrm{C}.20)$$

where the last inequality follows from $|I_{\ell_j+1, \ell_{j+1}-1}| = |I_{\ell_j+1, \ell_{j+1}+1}| - 2$ and $I_{\ell_j+1, \ell_{j+1}-1} \cap \mathcal{J}^{-1} \subseteq I_{\ell_j+1, \ell_{j+1}+1} \cap \mathcal{J}^{-1}$.

For $1 \leq j \leq \hat{J}_k$, we have $\ell_j - 1 \leq k < k_{\mathrm{init}} - 1$, then Theorem C.2 gives $\ell_j - 1 \in \mathcal{J}^{-1}$, Therefore, $|I_{0, k+1} \cap \mathcal{J}^{-1}| \geq \hat{J}_k$ and (C.16) yields $\log_\gamma(\tilde{C}_4 L_H) < \log_\gamma M_0 - \hat{J}_k$. That is, $\hat{J}_k \leq \log_\gamma \frac{\gamma M_0}{\tilde{C}_4 L_H}$. From (C.16), we know

$$k = |I_{1, k+1}| \overset{(\mathrm{C}.20)}{\leq} J_k \left( A^{-1} \log \frac{U_\varphi}{\epsilon_{k+1}} + 2 \right) + |I_{1, k+1} \cap \mathcal{J}^{-1}|$$

$$\overset{(\mathrm{C}.16)}{\leq} J_k \left( A^{-1} \log \frac{U_\varphi}{\epsilon_{k+1}} + 2 \right) + \log_\gamma \frac{M_0}{\tilde{C}_4 L_H} \leq \log_\gamma \frac{\gamma M_0}{\tilde{C}_4 L_H} \left( A^{-1} \log \frac{U_\varphi}{\epsilon_{k+1}} + 3 \right).$$

**Case 2** When $\delta_k = \epsilon_k \epsilon_{k-1}^{-1}$ for each $k \in \mathbb{N}$. For any $k < k_{\mathrm{init}} - 1$, we know a similar version of (C.19) holds since $\log \delta_{i+1} \leq 0$:

$$\log \frac{\epsilon_{k+1}}{\epsilon_0} = \sum_{i \in I_{0, k+1}} \log \delta_{i+1} \leq \sum_{i \in \mathcal{I}^+_{0, k+1}} \log \delta_{i+1}$$

$$< -2A |\mathcal{I}^+_{0, k+1}| \overset{(\mathrm{C}.18)}{\leq} -A(|I_{1, k+1}| - |I_{1, k+1} \cap \mathcal{J}^{-1}|).$$

Therefore, we have

$$k = |I_{1, k+1}| \leq A^{-1} \log \frac{\epsilon_0}{\epsilon_{k+1}} + |I_{1, k+1} \cap \mathcal{J}^{-1}| \overset{(\mathrm{C}.16)}{\leq} A^{-1} \log \frac{\epsilon_0}{\epsilon_{k+1}} + \log_\gamma \frac{\gamma M_0}{\tilde{C}_4 L_H}.$$

Finally, the proof is completed by setting $k = k_{\mathrm{init}} - 2$, and noticing that the conclusion automatically holds when $M_0 \leq \tilde{C}_4 L_H$. $\qquad \square$

## C.2 Proof of the global rates in Theorem 2.2

The following theorem provides a precise version of the global rates in Theorem 2.2. It can be translated into Theorem 2.2 by using the identity $[\log L_H]_+ + [\log L_H^{-1}]_+ = |\log L_H|$.

Since the right-hand sides of the following bounds are non-decreasing as $\epsilon_k$ decreases, whenever an $\epsilon$-stationary point is encountered such that $\epsilon_k \leq g_k \leq \epsilon$, the two inequalities below hold with $\epsilon_k$ replaced by $\epsilon$. Hence, the iteration bounds in Theorem 2.2 are valid.

**Theorem C.6** (Precise statement of the global rates in Theorem 2.2). *Let $\{x_k\}_{k\geq 1}$ be generated by Algorithm 1 with $\theta \geq 0$. Under Assumption 2.1 and let $C = \max(\tilde{C}_4, \gamma \tilde{C}_5)^{\frac{1}{2}} \tilde{C}_1^{-1}$ with the constants $\tilde{C}_1, \tilde{C}_4, \tilde{C}_5$ defined in Theorem C.1, and let $\tilde{C}_3, k_{\mathrm{init}}$ be defined in Theorem C.5, we have*

1. *If $\omega_k^{\mathrm{f}} = \sqrt{g_k}$, and $\omega_k^{\mathrm{t}} = \omega_k^{\mathrm{f}} \min(1, g_k^\theta g_{k-1}^{-\theta})$, then*

$$k \leq \left[\log_\gamma \frac{\gamma M_0}{\tilde{C}_4 L_H}\right]_+ \left(\tilde{C}_3 \log \frac{U_\varphi}{\epsilon_k} + 3\right)$$

$$+ 5\left(C\Delta_\varphi L_H^{\frac{1}{2}} \epsilon_k^{-\frac{3}{2}} + \left[\log_\gamma \frac{\tilde{C}_5 L_H}{M_0}\right]_+ + 2\right)\left(\log\log \frac{U_\varphi}{\epsilon_k} + 7\right) + 2.$$

2. *If $\omega_k^{\mathrm{f}} = \sqrt{\epsilon_k}$, and $\omega_k^{\mathrm{t}} = \omega_k^{\mathrm{f}} \epsilon_k^\theta \epsilon_{k-1}^{-\theta}$, then*

$$k \leq 40\left(C\Delta_\varphi L_H^{\frac{1}{2}} \epsilon_k^{-\frac{3}{2}} + \left[\log_\gamma \frac{\tilde{C}_5 L_H}{M_0}\right]_+ + 2\right)$$

$$+ \left[\log_\gamma \frac{M_0}{\tilde{C}_4 L_H}\right]_+ + (24 + \tilde{C}_3)\log \frac{U_\varphi}{\epsilon_k} + 2.$$

*Moreover, there exists a subsequence $\{x_{k_j}\}_{j\geq 0}$ such that $\lim_{j\to\infty} x_{k_j} = x^*$ with $\nabla\varphi(x^*) = 0$.*

*Proof.* Let $k_{\mathrm{init}}$ be defined in Theorem C.5, without loss of generality, we can drop the iterations $\{x_j\}_{j\leq k_{\mathrm{init}}}$ and assume $M_0 \leq \tilde{C}_4 L_H$, where $\tilde{C}_4$ is defined in Theorem C.1. By Theorem C.1, we know $k \in \mathcal{J}^1$ implies $M_k \leq \tilde{C}_5 L_H$, and hence $\sup_{j\geq 0} M_j \leq \max(\tilde{C}_4, \gamma\tilde{C}_5)L_H$.

By applying Theorem C.4, we have

$$-\Delta_\varphi \leq \varphi(x_k) - \varphi(x_0) \overset{(\mathrm{C.12})}{\leq} -C_{0,k}\Sigma_k \epsilon_k^{\frac{3}{2}} \leq -\tilde{C}_1(\max(\tilde{C}_4, \gamma\tilde{C}_5)L_H)^{-\frac{1}{2}}\Sigma_k \epsilon_k^{\frac{3}{2}},$$

which implies that $\Sigma_k \leq CL_H^{\frac{1}{2}}\Delta_\varphi \epsilon_k^{-\frac{3}{2}}$ with $C = \max(\tilde{C}_4, \gamma\tilde{C}_5)^{\frac{1}{2}}\tilde{C}_1^{-1}$, and the theorem can be proved by find a lower bound over $\Sigma_k$.

**Case 1** For the first choice of regularizers, Theorem 3.4 shows that $V_k \leq J_k \log\log \frac{U_\varphi}{\epsilon_k}$, and hence,

$$\Sigma_k \geq |I_{0,k} \cap \mathcal{J}^{-1}| + \max\left(|S_k \cap \mathcal{J}^{-1}|, |I_{0,k} \cap \mathcal{J}^0| - J_k\left(\log\log \frac{U_\varphi}{\epsilon_k} + 5\right)\right)$$

$$\overset{(\mathrm{D.25})}{\geq} \frac{k}{5\left(\log\log \frac{U_\varphi}{\epsilon_k} + 7\right)} - \left[\log_\gamma \frac{\tilde{C}_5 L_H}{M_0}\right]_+ - 2,$$

where Theorem D.7 is invoked with $W_k = 0$ and $U_k = \log\log \frac{U_\varphi}{\epsilon_k} + 5$. Reorganizing the above inequality and incorporating the initial phase in Theorem 3.5 yields

$$k \leq k_{\mathrm{init}} + 5\left(C\Delta_\varphi L_H^{\frac{1}{2}} \epsilon_k^{-\frac{3}{2}} + \left[\log_\gamma \frac{\tilde{C}_5 L_H}{M_0}\right]_+ + 2\right)\left(\log\log \frac{U_\varphi}{\epsilon_k} + 7\right).$$

**Case 2** For the second choice of regularizers, Theorem 3.4 shows that $V_k \leq \log \frac{U_\varphi}{\epsilon_k} + J_k$, and

$$\Sigma_k \geq |I_{0,k} \cap \mathcal{J}^{-1}| + \max\left(|S_k \cap \mathcal{J}^{-1}|, |I_{0,k} \cap \mathcal{J}^0| - \log \frac{U_\varphi}{\epsilon_k} - 6J_k\right).$$

Using Theorem D.7 with $U_k = 6$ and $W_k = \log \frac{U_\varphi}{\epsilon_k}$, we know either $\log \frac{U_\varphi}{\epsilon_k} \geq k/24$, or

$$\Sigma_k \geq \frac{k}{40} - \left[\log_\gamma \frac{\tilde{C}_5 L_H}{M_0}\right]_+ - 2.$$

By incorporating the case $k \leq 24\log \frac{U_\varphi}{\epsilon_k}$ and the initial phase in Theorem 3.5, the proof is completed.

**The subsequence convergence** From the global complexity we know $\lim_{k \to \infty} \epsilon_k = 0$. Since $\epsilon_k = \min(\epsilon_{k-1}, g_k)$, we can construct a subsequence $\{x_{k_j}\}_{j \geq 0}$ such that $g_{k_j} = \epsilon_{k_j}$. Note $\varphi(x_{k_j}) \leq \varphi(x_0)$ and the compactness of the sublevel set $L_\varphi(x_0)$ in Assumption 2.1, we know there is a further subsequence of $\{x_{k_j}\}$ converging to some point $x^*$. Since $\nabla \varphi$ is a continuous map, we know $\nabla \varphi(x^*) = 0$. □

### C.3 Proof of Theorem 2.3

*Proof.* The two gradient evaluations come from $\nabla\varphi(x_k)$ and $\nabla\varphi(x_k + d_k)$. The number of function value evaluations in a linesearch criterion is upper bounded by $m_{\max} + 1$, In the SOL case, at most two criteria are tested, in the NC case one criterion is tested. Thus, the total number of function evaluations is bounded by $2m_{\max} + 2$. The number of Hessian-vector product evaluations can be bounded using Theorem B.2. □

## D Technical lemmas for global rates

### D.1 Descent lemmas and their proofs

In this section we provide the descent lemmas for the NC case (Theorem D.2) and the SOL case (Theorem D.3). The lemma for the NC case is the same as He et al. [30, Lemma 6.3], and we include the proof for completeness. However, the linesearch rules for the SOL case are different, so we need a complete proof.

The following lemma transfers Theorem 2.1 to two useful inequalities.

**Lemma D.1** (Nesterov et al. [43]). *Under Theorem 2.1, we have the following inequalities:*

$$\|\nabla\varphi(x+d) - \nabla\varphi(x) - \nabla^2\varphi(x)d\| \leq \frac{L_H}{2}\|d\|^2, \tag{D.1}$$

$$\varphi(x+d) \leq \varphi(x) + \nabla\varphi(x)^\top d + \frac{1}{2}d^\top\nabla^2\varphi(x)d + \frac{L_H}{6}\|d\|^3. \tag{D.2}$$

**Lemma D.2** (Descent lemma for the NC state). *Suppose d_type, $d, \tilde{d}, m$ be the those in the subroutine NewtonStep of Algorithm 1, and $x, \omega, M$ be its inputs. Suppose d_type = NC and let $m_*$ be the smallest integer such that (2.7) holds. If $0 < m_* \leq m_{\max}$, we have*

$$\beta^{m_*-1} > \frac{3M(1-2\mu)}{L_H}, \tag{D.3}$$

$$\varphi(x+\beta^{m_*}d) - \varphi(x) < -\frac{9\beta^2(1-2\mu)^2\mu}{L_H^2}M^{\frac{3}{2}}\omega^3. \tag{D.4}$$

*When $m_* = 0$, the linesearch rule gives*

$$\varphi(x+d) - \varphi(x) \leq -\mu M^{-\frac{1}{2}}\omega^3. \tag{D.5}$$

*Finally, when $m_* > m_{\max}$, we have $M \leq (3-6\mu)^{-1}L_H$.*

*Proof.* Let $H = \nabla^2\varphi(x)$, from (2.6) we can verify that $\|d\| = L(\bar{d}) = M^{-1}\|d\|^{-2}|d^\top Hd|$, where $\bar{d} = \|\tilde{d}\|^{-1}\tilde{d}$ and $\tilde{d}$ is the direction satisfying Theorem B.2. Then, $d^\top Hd = -M\|d\|^3$ and $d^\top\nabla\varphi(x) \leq 0$. When $m_* \geq 1$, let $0 \leq j \leq m_* - 1$, then (2.7) fails to hold with $m = j$, and

$$-\mu\beta^{2j}M\|d\|^3 < \varphi(x+\beta^j d) - \varphi(x) \overset{(D.2)}{\leq} \beta^j\nabla\varphi(x)^\top d + \frac{\beta^{2j}}{2}d^\top Hd + \frac{L_H}{6}\beta^{3j}\|d\|^3$$

$$\leq \frac{\beta^{2j}}{2}d^\top Hd + \frac{L_H}{6}\beta^{3j}\|d\|^3 \tag{D.6}$$

$$= -\frac{\beta^{2j}}{2}M\|d\|^3 + \frac{L_H}{6}\beta^{3j}\|d\|^3. \tag{D.7}$$

Dividing both sides by $\beta^{2j}\|d\|^3$ we have

$$-M\mu < -\frac{M}{2} + \frac{L_H}{6}\beta^j. \tag{D.8}$$

Therefore, rearranging the above inequality gives (D.3).

From (B.8) and (2.6), we know $\tilde{d}^\top H \tilde{d} \le -\sqrt{M}\omega\|\tilde{d}\|^2$ and hence $\|d\| = M^{-1}\frac{|\tilde{d}^\top H \tilde{d}|}{\|\tilde{d}\|^2} \ge M^{-\frac{1}{2}}\omega$. By the linesearch rule (2.7), we have

$$\varphi(x + \beta^{m_*}d) - \varphi(x) \le -\mu\beta^{2m_*}M\|d\|^3 \le -\mu\beta^{2m_*}M^{-\frac{1}{2}}\omega^3 \overset{(D.3)}{<} -\frac{9\beta^2(1-2\mu)^2\mu}{L_H^2}M^{\frac{3}{2}}\omega^3.$$

When $m_* = 0$, (D.5) can be also proven using the above argument.

Finally, when $m_* > m_{\max} \ge 0$, we know (2.7) fails to holds with $m = 0$, and then (D.8) holds with $j = 0$. Therefore, we have $M < (3 - 6\mu)^{-1}L_H$.

$\square$

The following lemma summarizes the properties of `NewtonStep` for `SOL` case. Its first item is the necessary condition that the linesearch (2.4) or (2.5) fails, which will be used by subsequent items.

**Lemma D.3** (Descent lemma for the `SOL` state). *Suppose $d\_type, d, m, \hat{m}, \alpha$ be the those in the subroutine `NewtonStep` of Algorithm 1, and $x, \omega, M$ be its inputs. Suppose $d\_type = SOL$, and let $m_* \ge 0$ be the smallest integer such that (2.4) holds, and $\hat{m}_* \ge 0$ be the smallest integer such that (2.5) holds, then we have*

1. *Suppose $\mu\tau\beta^j d^\top \nabla\varphi(x) < \varphi(x + \tau\beta^j d) - \varphi(x)$ for some $\tau \in (0, 1]$ and $j \ge 0$, then*

$$\beta^j > \sqrt{\frac{6(1-\mu)M^{\frac{1}{2}}\omega}{L_H\tau^2\|d\|}} = \frac{\sqrt{2}C_M\omega^{\frac{1}{2}}}{\tau M^{\frac{1}{4}}\|d\|^{\frac{1}{2}}}, \tag{D.9}$$

*where $C_M := \sqrt{\frac{3(1-\mu)M}{L_H}} \ge \sqrt{\frac{M}{L_H}}$.*

2. *If $m_{\max} \ge m_* > 0$, then $\alpha = \beta^{m_*}$ and*

$$\beta^{m_*-1} > \max\left(\beta^{m_{\max}-1}, C_M\|\nabla\varphi(x)\|^{-\frac{1}{2}}\omega\right), \tag{D.10}$$

$$\varphi(x + \alpha d) - \varphi(x) < -\frac{36\beta\mu(1-\mu)^2}{L_H^2}M^{\frac{3}{2}}\omega^3. \tag{D.11}$$

3. *If $m_* > m_{\max}$ but $m_{\max} \ge \hat{m}_* > 0$, then $\beta^{\hat{m}_*-1} > \sqrt{2}C_M$.*

4. *If $m_* > m_{\max}$ but $m_{\max} \ge \hat{m}_* \ge 0$, then $\alpha = \hat{\alpha}\beta^{\hat{m}_*}$ with $\hat{\alpha} = \min(1, \omega^{\frac{1}{2}}M^{-\frac{1}{4}}\|d\|^{-\frac{1}{2}})$, and*

$$\varphi(x + \alpha d) - \varphi(x) < -\mu\beta^{\hat{m}_*}C_M^3 \min\left(C_M, 1\right)M^{-\frac{1}{2}}\omega^3. \tag{D.12}$$

5. *If both $m_* > m_{\max}$ and $\hat{m}_* > m_{\max}$, then $M \le \frac{L_H}{2}$.*

6. *If $m_* = 0$ (i.e., the stepsize $\alpha = 1$), then*

$$\varphi(x + d) - \varphi(x) \le -\frac{4\mu M^{-\frac{1}{2}}}{25 + 8L_H M^{-1}} \min\left(\|\nabla\varphi(x+d)\|^2\omega^{-1}, \omega^3\right). \tag{D.13}$$

*Proof.* Let $H = \nabla^2\varphi(x)$. We note that in the `SOL` setting, the direction $d$ is the same as $\tilde{d}$ returned by `CappedCG`, so Theorem B.2 holds for $d$.

(1). By the assumption we have

$$\mu\tau\beta^j d^\top \nabla\varphi(x) < \varphi(x + \tau\beta^j d) - \varphi(x) \overset{(D.2)}{\le} \tau\beta^j d^\top \nabla\varphi(x) + \frac{\tau^2\beta^{2j}}{2}d^\top Hd + \frac{L_H}{6}\tau^3\beta^{3j}\|d\|^3,$$

Rearranging the above inequality and dividing both sides by $\tau\beta^j$, we have

$$-(1-\mu)d^\top \nabla\varphi(x) < \frac{\tau\beta^j}{2}d^\top Hd + \frac{L_H}{6}\tau^2\beta^{2j}\|d\|^3. \tag{D.14}$$

From Theorem B.2, we know that $d^\top \nabla\varphi(x) = -d^\top H d - 2\sqrt{M}\omega\|d\|^2$, then since $\mu \in (0, 1/2)$, $j \geq 0$ and $\beta \in (0, 1)$, $\tau \in (0, 1]$, we have $1 - \mu > 1/2 \geq \beta^j/2 \geq \tau\beta^j/2$ and

$$\frac{L_H}{6}\tau^2\beta^{2j}\|d\|^3 \overset{(D.14)}{>} \left(1 - \mu - \frac{\tau\beta^j}{2}\right) d^\top H d + 2\sqrt{M}\omega(1-\mu)\|d\|^2$$

$$\overset{(B.4)}{>} -\sqrt{M}\omega\left(1 - \mu - \frac{\tau\beta^j}{2}\right)\|d\|^2 + 2\sqrt{M}\omega(1-\mu)\|d\|^2$$

$$= \sqrt{M}\omega\left(1 - \mu + \frac{\tau\beta^j}{2}\right)\|d\|^2.$$

Therefore, we have

$$\beta^{2j} > \frac{6\sqrt{M}\omega(1 - \mu + \tau\beta^j/2)}{L_H\tau^2\|d\|} \geq \frac{6\sqrt{M}\omega(1-\mu)}{L_H\tau^2\|d\|}, \tag{D.15}$$

which proves (D.9).

(2). In particular, when $m_* > 0$, we know (2.4) is violated for $m = 0$, then (D.9) with $\tau = 1$ and $j = 0$ gives a lower bound of $d$:

$$\|d\| > \frac{6\sqrt{M}\omega(1-\mu)}{L_H} \geq C_M^2 M^{-\frac{1}{2}}\omega. \tag{D.16}$$

Note that (2.4) is also violated for $m_* - 1$, then (D.9) holds with $(j, \tau) = (m_* - 1, 1)$, and we have

$$\beta^{m_*-1} \overset{(D.9)}{\geq} \sqrt{\frac{6\sqrt{M}\omega(1-\mu)}{L_H\|d\|}} \overset{(B.5)}{\geq} \sqrt{\frac{3(1-\mu)}{L_H}\frac{M\omega^2}{\|\nabla\varphi(x)\|}} = C_M\|\nabla\varphi(x)\|^{-\frac{1}{2}}\omega, \tag{D.17}$$

which yields (D.10). Moreover, the descent of the function value can be bounded as follows:

$$\varphi(x + \beta^{m_*}d) - \varphi(x) \overset{(2.4)}{\leq} \mu\beta^{m_*}d^\top\nabla\varphi(x)$$

$$\overset{(B.7)}{=} -\mu\beta^{m_*}d^\top(H + 2\sqrt{M}\omega I_n)d \overset{(B.3)}{\leq} -\mu\sqrt{M}\omega\beta^{m_*}\|d\|^2$$

$$\overset{(D.17)}{<} -\mu\beta\sqrt{M}\omega\|d\|^2\sqrt{\frac{6\sqrt{M}\omega(1-\mu)}{L_H\|d\|}} = -\mu\beta(\sqrt{M}\omega\|d\|)^{\frac{3}{2}}\sqrt{\frac{6(1-\mu)}{L_H}}$$

$$\overset{(D.16)}{<} -\frac{36\beta\mu(1-\mu)^2}{L_H^2}M^{\frac{3}{2}}\omega^3. \tag{D.18}$$

(3). The linesearch rule (2.5) can be regarded as using the rule in (2.4) with a new direction $\hat\alpha d$, where $\hat\alpha = \min(1, \omega^{\frac{1}{2}}M^{-\frac{1}{4}}\|d\|^{-\frac{1}{2}})$. Since $\hat{m}_* > 0$, then (2.5) is violated for $0 \leq j < \hat{m}_*$, and (D.9) with $\tau = \hat\alpha$ gives

$$\beta^{2j} > \frac{6\sqrt{M}\omega(1-\mu)}{L_H\hat\alpha^2\|d\|} \geq \frac{6M(1-\mu)}{L_H} = 2C_M^2. \tag{D.19}$$

Thus, the result follows from setting $j = \hat{m}_* - 1$.

(4). Since $m_* > m_{\max} \geq 0$, then the linesearch rule (2.4) is violated for $m = 0$ such that (D.16) holds. Hence, following the first two lines of the proof of (D.18), we have

$$\varphi(x + \hat\alpha\beta^{\hat{m}_*}d) - \varphi(x) \leq -\mu\beta^{\hat{m}_*}M^{\frac{1}{2}}\omega\hat\alpha\|d\|^2$$

$$= -\mu\beta^{\hat{m}_*}M^{\frac{1}{2}}\omega\min\left(\|d\|^2, \omega^{\frac{1}{2}}M^{-\frac{1}{4}}\|d\|^{\frac{3}{2}}\right)$$

$$\overset{(D.16)}{\leq} -\mu\beta^{\hat{m}_*}M^{\frac{1}{2}}\omega\min\left(C_M^4 M^{-1}\omega^2, C_M^3 M^{-1}\omega^2\right)$$

$$= -\mu\beta^{\hat{m}_*}C_M^3\min\left(C_M, 1\right)M^{-\frac{1}{2}}\omega^3.$$

(5). Since $\hat{m}_* > m_{\max} \geq 0$, then (D.19) holds with $j = 0$, which implies that $1 > 2C_M^2$, i.e., $2M \leq L_H$.

(6). When $m_* = 0$, by the linesearch rule and Theorem B.2 we have

$$\varphi(x + d) - \varphi(x) \leq \mu d^\top \nabla \varphi(x) \leq -\mu \sqrt{M} \omega \|d\|^2. \tag{D.20}$$

It remains to give a lower bound of $\|d\|$ as in (D.16), which is similar to the proof of He et al. [30, Lemma 6.2] with their $\epsilon_H$ and $\zeta$ replaced with our $\sqrt{M}\omega$ and $\tilde{\eta}$. Since special care must be taken with respect to $M$, we present the proof below. Note that

$$
\begin{aligned}
\|\nabla \varphi(x + d)\| &\leq \|\nabla \varphi(x + d) - \nabla \varphi(x) - \nabla^2 \varphi(x) d\| \\
&\quad + \|\nabla \varphi(x) + (\nabla^2 \varphi(x) + 2\sqrt{M}\omega I_n) d\| + 2\sqrt{M}\omega \|d\| \\
&\overset{(B.6)}{\leq} \frac{L_H}{2} \|d\|^2 + \sqrt{M} \left( \frac{1}{2} \omega \tilde{\eta} + 2\omega \right) \|d\|.
\end{aligned}
$$

Then, by the property of quadratic functions, we know

$$
\begin{aligned}
\|d\| &\geq \frac{-(\tilde{\eta} + 4) + \sqrt{(\tilde{\eta} + 4)^2 + 8 L_H (\sqrt{M}\omega)^{-2} \|\nabla \varphi(x + d)\|}}{2 L_H} \sqrt{M} \omega \\
&\geq c_0 \sqrt{M} \omega \min \left( \omega^{-2} \|\nabla \varphi(x + d)\|, 1 \right),
\end{aligned}
$$

where $c_0 := \frac{4M^{-1}}{4 + \tilde{\eta} + \sqrt{(4+\tilde{\eta})^2 + 8M^{-1}L_H}} \geq \frac{2M^{-1}}{\sqrt{(4+\tilde{\eta})^2 + 8M^{-1}L_H}} \geq \frac{2M^{-1}}{\sqrt{25 + 8M^{-1}L_H}}$, and we have used the inequality $-a + \sqrt{a^2 + bs} \geq (-a + \sqrt{a^2 + b}) \min(s, 1)$ from Royer and Wright [48, Lemma 17], with $a = \tilde{\eta} + 4 \leq 5$, $b = 8 L_H M^{-1}$ and $s = \omega^{-2} \|\nabla \varphi(x + d)\|$. Combining with (D.20), we get (D.13). $\qquad\square$

## D.2 Proof of Lemma C.1

In this section, we provide the proof of Theorem C.1. It is highly technical but mostly based on the descent lemmas (Theorems D.2 and D.3) and the choices of regularizers in Theorem 2.2.

First, we give an auxiliary lemma for the claim about $k \in \mathcal{J}^{-1}$ in Theorem C.1.

**Lemma D.4.** *Suppose the following two properties are true:*

1. *Suppose $d\_type_k \neq \mathtt{SOL}$ or $m_k > 0$. If $M_k > \tilde{C}_4 L_H$ and $\omega_k \geq \tau_- \omega_k^{\mathrm{f}}$, then $k \in \mathcal{J}^{-1}$;*

2. *Suppose $d\_type_k = \mathtt{SOL}$ and $m_k = 0$. If $M_k > L_H$ and $\min \left( \omega_k^3, g_{k+1}^2 \omega_k^{-1} \right) \geq \tau_- (\omega_k^{\mathrm{f}})^3$, then $k \in \mathcal{J}^{-1}$,*

*where $\delta_k^\theta = \omega_k^{\mathrm{t}} (\omega_k^{\mathrm{f}})^{-1}$ is defined in Theorem 2.2. Then, if $M_k > \tilde{C}_4 L_H$ and $\tau_- \leq \min \left( \delta_k^\alpha, \delta_{k+1}^\alpha \right)$, we know $k \in \mathcal{J}^{-1}$.*

*Proof.* Let $\alpha = \max(2, 3\theta)$. We consider the following two cases:

1. Note that $\tau_- < 1$. If $\omega_k < \tau_- \omega_k^{\mathrm{f}}$, then we know the trial step is accepted since $\omega_k \neq \omega_k^{\mathrm{f}}$, and hence, $\omega_k = \omega_k^{\mathrm{t}}$ and $\tau_- > \delta_k^\theta \geq \delta_k^\alpha$ since $\delta_k \in (0, 1]$ and $\theta \leq \alpha$.

2. If $\min \left( g_{k+1}^2 \omega_k^{-1}, \omega_k^3 \right) < \tau_- (\omega_k^{\mathrm{f}})^3$, we use the choice $\omega_k^{\mathrm{f}} = \sqrt{g_k}$ as an example, the case for $\omega_k^{\mathrm{f}} = \sqrt{\epsilon_k}$ is similar and follows from $g_{k+1} \geq \epsilon_{k+1}$. In this case, we have $\delta_k = \min(1, g_k g_{k-1}^{-1})$. When the fallback step is taken, we have $\omega_k = \omega_k^{\mathrm{f}}$, and

$$\tau_- > g_k^{-\frac{3}{2}} \min \left( g_{k+1}^2 g_k^{-\frac{1}{2}}, g_k^{\frac{3}{2}} \right) = \delta_k^2.$$

Since $\delta_k \in (0, 1]$ and $2 \leq \alpha$, we have $\tau_- > \delta_k^\alpha$. On the other hand, when the trial step is taken, we have $\omega_k = \omega_k^{\mathrm{t}} = \sqrt{g_k} \delta_k^\theta$ and

$$
\begin{aligned}
\tau_- &> g_k^{-\frac{3}{2}} \min \left( g_{k+1}^2 g_k^{-\frac{1}{2}} \delta_k^{-\theta}, g_k^{\frac{3}{2}} \delta_k^{3\theta} \right) \overset{(\delta_k \leq 1)}{\geq} g_k^{-\frac{3}{2}} \min \left( g_{k+1}^2 g_k^{-\frac{1}{2}}, g_k^{\frac{3}{2}} \delta_k^{3\theta} \right) \\
&= \min \left( g_{k+1}^2 g_k^{-2}, \delta_k^{3\theta} \right) \geq \min \left( \delta_{k+1}^2, \delta_k^{3\theta} \right) \geq \min \left( \delta_{k+1}^\alpha, \delta_k^\alpha \right).
\end{aligned}
$$

Conversely, we find when $\tau_- \leq \min\left(\delta_k^\alpha, \delta_{k+1}^\alpha\right)$, the assumptions of this lemma give that $k \in \mathcal{J}^{-1}$. $\qquad\square$

We will also show that the two properties listed in Theorem D.4 hold in the proof of Theorem C.1 below, and leave this fact as a corollary for our subsequent usage.

**Corollary D.5.** *Under the regularizers in Theorem 2.2, the two properties in Theorem D.4 hold.*

*Proof of Theorem C.1.* Define $\Delta_k = \varphi(x_k) - \varphi(x_{k+1})$. We denote $\omega_k = \omega_k^{\mathrm{t}}$ if the trial step is taken, and $\omega_k = \omega_k^{\mathrm{f}}$ otherwise.

**Case 1** When $\mathrm{d\_type}_k = \mathrm{SOL}$ and $m_k = 0$, i.e., $x_{k+1} = x_k + d_k$, we define $E_k := \min\left(g_{k+1}^2 \omega_k^{-1}, \omega_k^3\right)$.

1. When $k \in \mathcal{J}^1$, i.e., $M_{k+1} = \gamma M_k$, we have

$$\frac{4\mu}{33}\tau_+ M_k^{-\frac{1}{2}} E_k \geq \Delta_k \overset{(D.13)}{\geq} \frac{4\mu M_k^{-\frac{1}{2}}}{25 + 8L_H M_k^{-1}} E_k,$$

where the first inequality follows from the condition for increasing $M_k$ in Algorithm 1. The above display implies $25 + 8L_H M_k^{-1} \geq 33\tau_+^{-1} \geq 33$ as $\tau_+ \leq 1$, and hence, $M_k \leq L_H$.

2. When $E_k \geq \tau_-(\omega_k^{\mathrm{f}})^3$ and $M_k > L_H$, we have $k \in \mathcal{J}^{-1}$ since

$$\Delta_k \overset{(D.13)}{\geq} \frac{4\mu M_k^{-\frac{1}{2}} E_k}{25 + 8L_H M_k^{-1}} > \frac{4\mu M_k^{-\frac{1}{2}}\tau_-(\omega_k^{\mathrm{f}})^3}{25 + 8} = \frac{4}{33}\mu\tau_- M_k^{-\frac{1}{2}}(\omega_k^{\mathrm{f}})^3,$$

which satisfies the condition in Algorithm 1 for decreasing $M_k$ since $\bar\omega$ therein is $\omega_k^{\mathrm{f}}$. Thus, the second property of Theorem D.4 is true.

**Case 2** When $\mathrm{d\_type}_k = \mathrm{SOL}$, and let $m_*$ and $\hat{m}_*$ be the smallest integer such that (2.4) and (2.5) hold, respectively, as defined in Theorem D.3. We also recall that $C_{M_k}^2 = \frac{3(1-\mu)M_k}{L_H} \geq \frac{M_k}{L_H}$.

Since the previous case addresses $m_* = 0$, we assume $m_* > 0$ here. Then, the condition for increasing $M_k$ in Algorithm 1 is

$$\Delta_k \leq \tau_+ \beta\mu M_k^{-\frac{1}{2}}\omega_k^3. \tag{D.21}$$

The condition for decreasing $M_k$ is

$$\Delta_k \geq \mu\tau_- M_k^{-\frac{1}{2}}(\omega_k^{\mathrm{f}})^3. \tag{D.22}$$

1. When $k \in \mathcal{J}^1$ and $m_{\max} \geq m_* > 0$, i.e., $m_k = m_*$ and $x_{k+1} = x_k = \beta^{m_k} d_k$, we have

$$\tau_+ \beta\mu M_k^{-\frac{1}{2}}\omega_k^3 \overset{(D.21)}{\geq} \Delta_k \overset{(D.11)}{\geq} \frac{36\beta\mu(1-\mu)^2}{L_H^2}M_k^{\frac{3}{2}}\omega_k^3 \geq \frac{9\beta\mu}{L_H^2}M_k^{\frac{3}{2}}\omega_k^3,$$

Since $\tau_+ \leq 1$, then we know $M_k \leq \tau_+^{\frac{1}{2}} L_H/3 \leq L_H/3$.

2. When $m_{\max} \geq m_* > 0$ and $M_k \geq \tau_-^{-1}(9\beta)^{-\frac{1}{2}}L_H$ and $\omega_k \geq \tau_-\omega_k^{\mathrm{f}}$, then

$$\Delta_k \overset{(D.11)}{\geq} \frac{9\beta\mu}{L_H^2}M_k^{\frac{3}{2}}\omega_k^3 = \left(\frac{9\beta\mu}{L_H^2}M_k^2\right)M_k^{-\frac{1}{2}}\omega_k^3 \geq \mu\tau_-^{-2}M_k^{-\frac{1}{2}}(\tau_-^3(\omega_k^{\mathrm{f}})^3) = \mu\tau_- M_k^{-\frac{1}{2}}(\omega_k^{\mathrm{f}})^3,$$

which satisfies (D.22), and hence $k \in \mathcal{J}^{-1}$.

3. When $k \in \mathcal{J}^1$ and $m_* > m_{\max}$ and $m_{\max} \geq \hat{m}_* \geq 0$, then we know

$$\tau_+ \beta\mu M_k^{-\frac{1}{2}}\omega_k^3 \overset{(D.21)}{\geq} \Delta_k \overset{(D.12)}{\geq} \mu\beta^{\hat{m}_*} C_{M_k}^3 \min\left(C_{M_k}, 1\right) M_k^{-\frac{1}{2}}\omega_k^3,$$

which implies $\beta \geq \beta\tau_+ \geq \beta^{\hat{m}_*}C_{M_k}^3 \min\left(C_{M_k}, 1\right)$. If $C_{M_k} \leq 1$, then its definition implies that $M_k \leq 2L_H/3$. Otherwise, we have $\beta \geq \beta^{\hat{m}_*}C_{M_k}^3$. When $\hat{m}_* = 0$, we know $C_{M_k}^3 \leq \beta \leq 1$ and hence $M_k \leq 2L_H/3$; when $\hat{m}_* > 0$, Theorem D.3 shows $\beta^{\hat{m}_*-1} > \sqrt{2}C_{M_k} > C_{M_k}$, and hence $C_{M_k}^4 \leq 1$, leading to $M_k \leq 2L_H/3$.

4. When $m_* > m_{\max}$ and $m_{\max} \geq \hat{m}_* \geq 0$, and $M_k \geq L_H$, we have $C_{M_k} \geq 1$ and by Theorem D.3, $\hat{m}_* = 0$, since otherwise we have $1 \geq \beta^{\hat{m}_* - 1} > \sqrt{2} C_{M_k} > 1$, leading to a contradiction. Then, (D.12) gives $\Delta_k \geq \mu M_k^{-\frac{1}{2}} \omega_k^3$, and therefore $k \in \mathcal{J}^{-1}$ as long as $\omega_k \geq \tau_- \omega_k^{\mathrm{f}}$.

5. When $m_* > m_{\max}$ and $\hat{m}_* > m_{\max}$, then Theorem D.3 shows that $M_k \leq L_H/2$, and the algorithm directly increases $M_k$ so that $k \in \mathcal{J}^1$.

The above arguments show that when $k \in \mathcal{J}^1$, we have $M_k \leq L_H \leq \tilde{C}_5 L_H$, and when $\omega_k \geq \tau_- \omega_k^{\mathrm{f}}$ and $M_k > \tilde{C}_4 L_H \geq \max(1, \tau_-^{-1}(9\beta)^{-\frac{1}{2}}) L_H$, we have $k \in \mathcal{J}^{-1}$, i.e., the first property of Theorem D.4 is true for SOL case.

**Case 3** When $\mathrm{d\_type}_k = \mathtt{NC}$, let $m_*$ be the smallest integer such that (2.7) holds, as defined in Theorem D.2. In this case, the condition for decreasing $M_k$ is also (D.22), and the condition for increasing it is

$$\Delta_k \leq \tau_+ (1 - 2\mu)^2 \beta^2 \mu M_k^{-\frac{1}{2}} \omega_k^3. \tag{D.23}$$

1. When $k \in \mathcal{J}^1$ and $m_* > 0$, we can similarly use (D.4) in Theorem D.2 and (D.23) to show that $M_k \leq L_H/3$.

2. When $m_* > 0$ and $M_k \geq \tau_-^{-1}(3\beta(1 - 2\mu))^{-1} L_H$ and $\tau_- \omega_k^{\mathrm{f}} \leq \omega_k$, then Theorem D.2 shows that (D.22) holds. Therefore, $k \in \mathcal{J}^{-1}$.

3. When $m_* = 0$, we show that $M_{k+1}$ will not increase, since otherwise (D.5) and (D.23) imply that $1 > (1 - 2\mu)^2 \beta^2 \tau_+ \geq 1$, leading to a contradiction.

4. When $m_* = 0$ and $\tau_- \omega_k^{\mathrm{f}} \leq \omega_k$, we know (D.22) holds from (D.5) and $\tau_- < 1$, and hence $k \in \mathcal{J}^{-1}$.

5. When $m_* > m_{\max}$ and $\hat{m}_* > m_{\max}$, then Theorem D.2 shows that $M_k \leq L_H/(3 - 6\mu)$, and the algorithm directly increases $M_k$ so that $k \in \mathcal{J}^1$.

The above arguments show that when $k \in \mathcal{J}^1$, we have $M_k \leq L_H/\min(1, 3 - 6\mu) \leq \tilde{C}_5 L_H$, and when $\omega_k \geq \tau_- \omega_k^{\mathrm{f}}$ and $M_k > \tilde{C}_4 L_H \geq \tau_-^{-1}(3\beta(1 - 2\mu))^{-1} L_H$, we have $k \in \mathcal{J}^{-1}$, i.e., the first property of Theorem D.4 is true for NC case.

**The cardinality of $\mathcal{J}^i$** By the definition of $\mathcal{J}^i$, we have

$$\log_\gamma M_k = \log_\gamma M_0 + |I_{0,k} \cap \mathcal{J}^1| - |I_{0,k} \cap \mathcal{J}^{-1}|.$$

For each $k$ we know $M_{k+1} > M_k$ only if $M_k \leq \tilde{C}_5 L_H$, then $\sup_k M_k \leq \max(M_0, \gamma \tilde{C}_5 L_H)$, and hence (C.1) holds. Adding $|I_{0,k} \setminus \mathcal{J}^1|$ to both sides of (C.1), we find (C.2) holds.

**The descent inequality** The $D_k$ dependence in (C.3) directly follow from Theorems D.2 and D.3. For the preleading coefficients, we consider the following three cases. (1). When $k \in \mathcal{J}^1$, the result also follows from the two lemmas and the fact that $M_k \geq 1$. We also note that the $L_H^{-\frac{5}{2}}$ dependence only comes from the case where $\mathrm{d\_type} = \mathtt{SOL}$ and $m$ does not exist, and for other cases the coefficient is of order $L_H^{-2}$; (2). When $k \in \mathcal{J}^{-1}$, the result follows from the algorithmic rule of decreasing $M_k$; (3). When $k \in \mathcal{J}^0$, we know the rules in the algorithm for increasing $M_k$ fail to hold, yielding an $M_k^{-\frac{1}{2}}$ dependence of the coefficient. $\qquad\square$

### D.3 Proof of Lemma 3.4

*Proof of Theorem 3.4.* When $\omega_k^{\mathrm{f}} = \sqrt{g_k}$, the upper bound over $V_k$ follows from the monotonicity of $\log\log \frac{3A}{a}$. On the other hand, when $\omega_k^{\mathrm{f}} = \sqrt{\epsilon_k}$, we know $3\epsilon_{\ell_j - 1} \geq 2\epsilon_{\ell_j - 1} \geq 2\epsilon_{\ell_{j+1} - 1}$ since

$\{\epsilon_k\}_{k\geq 0}$ is non-increasing. Then, we can apply Theorem D.6 below with $a = 3$ to obtain

$$V_k \leq \sum_{j=1}^{J_k-1} \log\log \frac{3\epsilon_{\ell_j-1}}{\epsilon_{\ell_{j+1}-1}} + \log\log \frac{3\epsilon_{\ell_{J_k}-1}}{\epsilon_k}$$

$$\overset{\text{(D.24)}}{\leq} \frac{1}{\log 3} \log \frac{\epsilon_{\ell_1-1}}{\epsilon_k} + J_k \log\log 3 \leq \log \frac{\epsilon_0}{\epsilon_k} + J_k,$$

where we have used the fact that $\log 3 \geq 1$ and $\log\log 3 \leq 1$. $\qquad\square$

**Lemma D.6.** *Let $\{b_j\}_{j\geq 1} \subseteq (0,\infty)$ be a sequence, and $a \geq 3$, $ab_j \geq 2b_{j+1}$, then we have for any $k \geq 1$,*

$$\sum_{j=1}^{k} \log\log \frac{ab_j}{b_{j+1}} \leq \frac{1}{\log a} \log \frac{b_1}{b_{k+1}} + k \log\log a. \tag{D.24}$$

*Proof.* Using the fact $\log(1+x) \leq x$ for $x > -1$, and $\log b_j - \log b_{j+1} \geq -\log a + \log 2 > -\log a$, we have

$$\sum_{j=1}^{k} \log\log \frac{ab_j}{b_{j+1}} = \sum_{j=1}^{k} \log\left(1 + \frac{\log b_j - \log b_{j+1}}{\log a}\right) + k \log\log a$$

$$\leq \sum_{j=1}^{k} \left(\frac{\log b_j - \log b_{j+1}}{\log a}\right) + k \log\log a$$

$$= \frac{\log b_1 - \log b_{k+1}}{\log a} + k \log\log a,$$

which completes the proof. $\qquad\square$

### D.4 The counting lemma

**Lemma D.7** (Counting lemma). *Let $\mathcal{J}^{-1}, \mathcal{J}^0, \mathcal{J}^1 \subset \mathbb{N}$ be the sets in Theorem C.1, then we have at least one of the following inequalities holds:*

$$\Sigma_k \geq \frac{k}{5(U_k + 2)} - [\log_\gamma(\tilde{C}_5 M_0^{-1} L_H)]_+ - 2, \tag{D.25}$$

$$W_k \geq \frac{k}{3(U_k + 2)}, \tag{D.26}$$

*where $\Sigma_k := |I_{0,k} \cap \mathcal{J}^{-1}| + \max\left(|S_k \cap \mathcal{J}^0|, |I_{0,k} \cap \mathcal{J}^0| - W_k - U_k J_k\right)$, and $S_k \subseteq I_{0,k}$, $U_k \geq 0$, $J_k - 1 = |S_k|$ and $W_k \in \mathbb{R}$, and $\tilde{C}_5$ is defined in Theorem C.1, $M_0$ is the input in Algorithm 1.*

*Proof.* Denote $B_k = (U_k + 2)^{-1} |I_{0,k} \cap \mathcal{J}^0|$ and $\Gamma_k = [\log_\gamma(\gamma \tilde{C}_5 M_0^{-1} L_H)]_+$. We consider the following five cases, where the first three cases deal with $J_k < B_k$, and the last two cases are the remaining parts. We also note that the facts $|I_{0,k}| = k$ and $1 \geq \frac{2}{U_k+2}$ are frequently used.

**Case 1** When $J_k < B_k$ and $W_k < B_k$, we have

$$\Sigma_k \geq |I_{0,k} \cap \mathcal{J}^{-1}| + |I_{0,k} \cap \mathcal{J}^0| - U_k J_k - W_k > |I_{0,k} \cap \mathcal{J}^{-1}| + \frac{|I_{0,k} \cap \mathcal{J}^0|}{U_k + 2}$$

$$\geq \frac{2|I_{0,k} \cap \mathcal{J}^{-1}| + |I_{0,k} \cap \mathcal{J}^0|}{U_k + 2} \overset{\text{(C.2)}}{\geq} \frac{k - \Gamma_k}{U_k + 2}.$$

**Case 2** When $J_k < B_k \leq W_k$, and $|I_{0,k} \cap \mathcal{J}^0| \leq \frac{k}{3}$, then by (C.2) we know $k \leq 2|I_{0,k} \cap \mathcal{J}^{-1}| + \frac{k}{3} + \Gamma_k$, and hence, $\Sigma_k \geq |I_{0,k} \cap \mathcal{J}^{-1}| \geq \frac{k}{3} - \frac{1}{2}\Gamma_k$.

**Case 3** When $J_k < B_k \leq W_k$, and $|I_{0,k} \cap \mathcal{J}^0| > \frac{k}{3}$, then $W_k \geq B_k > \frac{k}{3(U_k+2)}$.

**Case 4** When $|S_k \cap \mathcal{J}^0| > B_k/2$, we have

$$\Sigma_k \geq |I_{0,k} \cap \mathcal{J}^{-1}| + |S_k \cap \mathcal{J}^0| \geq \frac{2|I_{0,k} \cap \mathcal{J}^{-1}| + |I_{0,k} \cap \mathcal{J}^0|}{2(U_k + 2)} \overset{(C.2)}{\geq} \frac{k - \Gamma_k}{2(U_k + 2)}.$$

**Case 5** When $J_k \geq B_k$ and $|S_k \cap \mathcal{J}^0| \leq B_k/2$, we have

$$B_k - 1 \leq J_k - 1 = |S_k| = |S_k \cap \mathcal{J}^0| + |S_k \cap \mathcal{J}^1| + |S_k \cap \mathcal{J}^{-1}|$$
$$\leq \frac{B_k}{2} + |I_{0,k} \cap \mathcal{J}^1| + |I_{0,k} \cap \mathcal{J}^{-1}|$$
$$\overset{(C.1)}{\leq} \frac{B_k}{2} + 2|I_{0,k} \cap \mathcal{J}^{-1}| + \Gamma_k.$$

Therefore, we have

$$\Sigma_k \geq |I_{0,k} \cap \mathcal{J}^{-1}|$$
$$= \frac{1}{5}|I_{0,k} \cap \mathcal{J}^{-1}| + \frac{4}{5}|I_{0,k} \cap \mathcal{J}^{-1}|$$
$$\geq \frac{1}{5} \cdot \frac{8|I_{0,k} \cap \mathcal{J}^{-1}|}{4(U_k + 2)} + \frac{4}{5}\left(\frac{B_k}{4} - \frac{1}{2} - \frac{\Gamma_k}{2}\right)$$
$$= \frac{1}{5}\left(\frac{8|I_{0,k} \cap \mathcal{J}^{-1}| + 4|I_{0,k} \cap \mathcal{J}^0|}{4(U_k + 2)} - 2 - 2\Gamma_k\right)$$
$$\overset{(C.2)}{\geq} \frac{1}{5}\left(\frac{k - \Gamma_k}{U_k + 2} - 2 - 2\Gamma_k\right).$$

Summarizing the above cases, we conclude that

$$\Sigma_k \geq \frac{k}{5(U_k + 2)} - \Gamma_k - \frac{2}{5} \geq \frac{k}{5(U_k + 2)} - [\log_\gamma(\tilde{C}_5 M_0^{-1} L_H)]_+ - 2,$$

and the proof is completed. $\qquad\square$

### D.5 Technical lemmas for Lemma 3.2

This section establishes two crucial lemmas for proving Theorem 3.2 (a.k.a. Theorem C.3 in the appendix). Theorem D.8, mentioned in the "sketch of the idea" part of Theorem 3.2, is specifically applied to the case $\theta = 0$. For $\theta > 0$, we employ a modified version of this result as detailed in Theorem D.9.

**Lemma D.8.** *Given $K \in \mathbb{N}$, $p > q > 0$, and $A \geq a > 0$, and let $\{g_j\}_{0 \leq j \leq K+1}$ be such that $A = g_0 \geq g_1 \geq \cdots \geq g_K \geq g_{K+1} = a$. Then, for any subset $S \subseteq [K]$, we have*

$$\sum_{i \in S} \frac{g_{i+1}^p}{g_i^q} \geq \max(0, |S| - R_a - 2)e^{-q}a^{p-q}, \tag{D.27}$$

*where $R_a := \left\lfloor \log\log \frac{3A}{a} - \log\log \frac{p}{q} \right\rfloor \leq \log\log \frac{3A}{a}$.*

*Proof.* It suffices to consider the case where $A = 1$, since for general cases, we can invoke the result of $A = 1$ with $g_j, a$ replaced with $g_j/A, a/A$, respectively. Let $\tau = p/q$ and $\mathcal{I}_k = \{j \in [K] : \exp(\tau^k)a \leq g_j < \exp(\tau^{k+1})a\}$ with $0 \leq k \leq R_a$ and $\mathcal{I}_{-1} = \{j \in [K] : a \leq g_j < ea\}$. Let $\zeta_k = \exp(\tau^k)$ for $k \geq 0$ and $\zeta_{-1} = 1$, then we have $\zeta_k^p \zeta_{k+1}^{-q} \geq e^{-q}$. Note that $\{\mathcal{I}_k\}_{-1 \leq k \leq R_a}$ is a partition of $[K]$, then we have

$$\sum_{i \in S} \frac{g_{i+1}^p}{g_i^q} = \sum_{k=-1}^{R_a} \sum_{j \in \mathcal{I}_k \cap S} \frac{g_{j+1}^p}{g_j^q} = \sum_{k=-1}^{R_a} \left( \sum_{\substack{j \in S \\ j, j+1 \in \mathcal{I}_k}} \frac{g_{j+1}^p}{g_j^q} + \sum_{\substack{j \in S \\ j \in \mathcal{I}_k, j+1 \notin \mathcal{I}_k}} \frac{g_{j+1}^p}{g_j^q} \right)$$

$$\geq \sum_{k=-1}^{R_a} \sum_{\substack{j \in S \\ j, j+1 \in \mathcal{I}_k}} \frac{(\zeta_k a)^p}{(\zeta_{k+1} a)^q} \geq \sum_{k=-1}^{R_a} \sum_{\substack{j \in S \\ j, j+1 \in \mathcal{I}_k}} e^{-q} a^{p-q} = |\mathcal{I}_S| e^{-q} a^{p-q}, \tag{D.28}$$

where $\mathcal{I}_S := \{j \in S : j, j+1 \in \mathcal{I}_k, -1 \leq k \leq R_a\}$. By the monotonicity of $g_j$, we know for each $k$, there exists at most one $j \in \mathcal{I}_k$ such that $j + 1 \notin \mathcal{I}_k$. Hence, $|\mathcal{I}_S| \geq |S| - (R_a + 2)$. $\qquad\square$

**Lemma D.9.** *Given $K \in \mathbb{N}$, $p_1 > q_1 > 0$, $p_2 > q_2 > 0$ and $A \geq a > 0$, and let $\{g_j\}_{0 \leq j \leq K+1}$ be such that $A = g_0 \geq g_1 \geq \cdots \geq g_K \geq g_{K+1} = a$. Then, for any subset $S \subseteq [K]$, we have*

$$\sum_{i \in S} \min \left( A^{q_1 - p_1} \frac{g_{i+1}^{p_1}}{g_i^{q_1}}, A^{q_2 - p_2} \frac{g_i^{p_2}}{g_{i-1}^{q_2}} \right)$$

$$\geq \max(0, |S| - R_{a,1} - R_{a,2} - 4) \min \left( \left( A^{-1} a \right)^{p_1 - q_1}, \left( A^{-1} a \right)^{p_2 - q_2} \right), \qquad (\text{D.29})$$

*where $R_{a,i} := \left\lceil \log \log \frac{3A}{a} - \log \log \frac{p_i}{q_i} \right\rceil \leq \log \log \frac{3A}{a}$ for $i = 1, 2$.*

*Proof.* Similar to Theorem D.8, it suffices to show that (D.29) is true for $A = 1$. Let $\tau_i = p_i / q_i$ for $i = 1, 2$ and $\mathcal{I}_k = \{j \in [K] : \exp(\tau_1^k)a \leq g_j < \exp(\tau_1^{k+1})a\}$ with $0 \leq k \leq R_{a,1}$ and $\mathcal{I}_{-1} = \{j \in [K] : a \leq g_j < ea\}$. Note that $\{\mathcal{I}_k\}_{-1 \leq k \leq R_{a,1}}$ is a partition of $[K]$, then similar to (D.28) we have

$$\sum_{i \in S} \min \left( \frac{g_{i+1}^{p_1}}{g_i^{q_1}}, \frac{g_i^{p_2}}{g_{i-1}^{q_2}} \right) \geq \sum_{k=-1}^{R_{a,1}} \sum_{\substack{j \in S \\ j, j+1 \in \mathcal{I}_k}} \min \left( e^{-q_1} a^{p_1 - q_1}, \frac{g_i^{p_2}}{g_{i-1}^{q_2}} \right)$$

$$\geq \sum_{j \in \mathcal{I}_S} \min \left( e^{-q_1} a^{p_1 - q_1}, \frac{g_i^{p_2}}{g_{i-1}^{q_2}} \right),$$

where $\mathcal{I}_S := \{j \in S : j, j+1 \in \mathcal{I}_k, -1 \leq k \leq R_{a,1}\}$ and we have used the fact that $\min(\alpha_1, \beta) \geq \min(\alpha_2, \beta)$ if $\alpha_1 \geq \alpha_2$. Moreover, we can also conclude that $|\mathcal{I}_S| \geq |S| - R_{a,1} - 2$.

Next, we consider the partition of $\mathcal{I}_S$ and lower bound the summation in the above display. Let $\mathcal{J}_k = \{j \in \mathcal{I}_S : \exp(\tau_2^k)a \leq g_j < \exp(\tau_2^{k+1})a\}$ with $0 \leq k \leq R_{a,2}$, $\mathcal{J}_{-1} = \{j \in \mathcal{I}_S : a \leq g_j < ea\}$, and $\mathcal{J}_S := \{j \in S : j, j-1 \in \mathcal{J}_k, -1 \leq k \leq R_{a,2}\}$. Then, similar to (D.28) we have

$$\sum_{j \in \mathcal{I}_S} \min \left( e^{-q_1} a^{p_1 - q_1}, \frac{g_i^{p_2}}{g_{i-1}^{q_2}} \right) \geq \sum_{k=-1}^{R_{a,2}} \sum_{j, j-1 \in \mathcal{J}_k} \min \left( e^{-q_1} a^{p_1 - q_1}, e^{-q_2} a^{p_2 - q_2} \right)$$

$$= |\mathcal{J}_S| \min \left( e^{-q_1} a^{p_1 - q_1}, e^{-q_2} a^{p_2 - q_2} \right).$$

Therefore, the proof is completed by noticing that $|\mathcal{J}_S| \geq |\mathcal{I}_S| - R_{a,2} - 2$. $\qquad\square$

# E   Main results for local rates

In this section, we first provide the precise version of Theorem 3.6 in Theorems E.2 and E.3, and then prove the main result of the local convergence order. The proofs for technical lemmas are deferred to Sections F.1 and F.2.

**Assumption E.1** (Positive definiteness). *There exists $\alpha > 0$ such that $\nabla^2 \varphi(x^*) \succeq \alpha I_n$.*

Let $C(\alpha, a, b, U)$ be the constant defined in Theorem B.2, $\alpha$ be defined in Assumption E.1, and $\gamma, \mu, M_0, \eta$ be the inputs of Algorithm 1, and $\theta$ be defined in Theorem 2.2. We define the following

constants which will be subsequently used in Theorems E.2 and E.3:

$$U_M = \max(M_0, \tilde{C}_5 \gamma L_H), \delta_0 = \frac{\alpha}{2L_H}, L_g = \|\nabla^2 \varphi(x^*)\| + L_H \delta_0,$$

$$\tilde{c} = C\left(\frac{\alpha}{2}, (1+2\theta)^{-1}, \tau U_\varphi^{-\theta(1+2\theta)^{-1}} U_M^{\frac{1-\theta(1+2\theta)^{-1}}{2}}, L_g\right),$$

$$\delta_1^{\frac{1}{2}} = \min\left(\delta_0^{\frac{1}{2}}, \min(\eta, \tilde{c})(U_M L_g)^{-\frac{1}{2}}\right),$$

$$c_1 = \frac{4}{\alpha} \max\left(L_H \delta_1^{\frac{1}{2}}, 2(U_M L_g)^{\frac{1}{2}}(1+L_g)\right),$$

$$\delta_2^{\frac{1}{2}} = \min\left(\delta_1^{\frac{1}{2}}, \frac{1}{2c_1}, \frac{(1-2\mu)\alpha}{8L_g c_1\left(c_1\delta_1^{\frac{1}{2}}+1\right) + 32L_H \delta_1^{\frac{1}{2}}}\right),$$

$$c_2 = 4\alpha^{-2} \max\left(2\alpha^{-1} L_g L_H, (2+\alpha)L_g U_M^{\frac{1}{2}}\right),$$

$$\delta_3 = \min\left(\delta_2, c_2^{-2} L_g^{-1}(\delta_2^{\frac{1}{2}}+1)^{-2}, \frac{\alpha^2}{4}(L_H + 2U_M^{\frac{1}{2}} L_g^{\frac{1}{2}}(1+L_g))^{-2}\right).$$

**Lemma E.2** (Newton direction yields superlinear convergence). *Let $x, d, M$ and $\omega$ be those in the subroutine `NewtonStep` of Algorithm 1 with `d_type = SOL`. Let $x^*$ be such that $\nabla \varphi(x^*) = 0$ and $\nabla^2 \varphi(x^*) \succeq \alpha I_n$, then for $x \in B_{\delta_0}(x^*)$, we have the following inequalities*

$$\|x^* - (x+d)\| \le \frac{2}{\alpha}\left(L_H\|x - x^*\|^2 + 2M^{\frac{1}{2}}\omega(1+L_g)\|x - x^*\|\right), \tag{E.1}$$

$$\|\nabla \varphi(x+d)\| \le \frac{8L_g L_H}{\alpha^3}\|\nabla \varphi(x)\|^2 + \frac{4L_g(2+\alpha)}{\alpha^2} M^{\frac{1}{2}}\omega\|\nabla \varphi(x)\|. \tag{E.2}$$

The lemma below shows that the Newton direction will be taken when iterates are close enough to the solution.

**Lemma E.3** (Newton direction is eventually taken). *Let $x^* \in \mathbb{R}^n$ be such that $\nabla \varphi(x^*) = 0$ and Assumption E.1 holds. If $\max(\omega_k^{\text{t}}, \omega_k^{\text{f}}) \le \sqrt{g_k}$, then $d\_type_k = \text{SOL}$ and $m_k = 0$ exists for $x_k \in B_{\delta_2}(x^*)$. Moreover, the trial step using $\omega_k^{\text{t}}$ is accepted for $x_k \in B_{\delta_3}(x^*)$.*

## E.1 Proof of local rates in Theorem 2.2

The following proposition is the non-asymptotic statement of Theorem 2.2.

**Proposition E.4.** *Let $\{x_k\}_{k\ge 0}$ be the points generated by Algorithm 1 with the regularizer choices in Theorem 2.2 and $\theta \ge 0$; and $x^*, \{x_{k_j}\}_{j\ge 0}$ be those in Theorem C.6 such that $\lim_{j\to\infty} x_{k_j} = x^*$ and $\nabla \varphi(x^*) = 0$ and suppose Assumption E.1 holds, i.e., $\nabla^2 \varphi(x^*) \succeq \alpha I_n$.*

*Then, there exists $j_0$ such that $\epsilon_{j_0} = g_{j_0} < \min(1, (2c_2)^{-2})$ and $x_{j_0} \in B_{\delta_3}(x^*)$, and*

1. *$\lim_{k\to\infty} x_k = x^*$.*

2. *When $\theta \in (0, 1]$ and $j \ge 1$, we have*

$$\|\nabla \varphi(x_{j_0+j+1})\| \le (2c_2)^3 \|\nabla \varphi(x_{j_0+j})\|^{1+\nu_\infty - (4\theta/9)^k},$$

   *where $\nu_\infty \in \left[\frac{1}{2}, 1\right]$ is defined in Theorem F.3 and illustrated in Figure 1.*

3. *When $\theta > 1$ and $j \ge \log_2 \frac{2\theta-1}{2\theta-2} + 1$, we have*

$$\|\nabla \varphi(x_{j_0+j+1})\| \le (2c_2)^{2\theta+2} \|\nabla \varphi(x_{j_0+j})\|^2.$$

*Proof.* Since $\lim_{j\to\infty} x_{k_j} = x^*$ and $\nabla \varphi(x^*) = 0$, we know $j_0$ exists. We define the set

$$\mathcal{I} = \{j \in \mathbb{N} : g_j = \epsilon_j \text{ and } x_j \in B_{\delta_3}(x^*)\}. \tag{E.3}$$

By the existence of $j_0$, we know $j_0 \in \mathcal{I}$. Suppose $k \in \mathcal{I}$, then we will show that $k + 1 \in \mathcal{I}$. Since the choices of $\omega_k^{\text{f}}$ and $\omega_k^{\text{t}}$ in Theorem 2.2 fulfill the condition of Theorem E.3, we know the trial step is taken and $x_{k+1} = x_k + d_k$, where $d_k$ is the direction in `NewtonStep` with $\omega = \omega_k^{\text{t}}$.

From Theorem E.2 and Theorem F.2, we have $g_k \leq L_g \|x_k - x^*\| \leq L_g \delta_3$, $\omega_k^{\mathrm{t}} \leq \sqrt{g_k}$ and

$$g_{k+1} \overset{(E.2)}{\leq} c_2 g_k^2 + c_2 \omega_k^{\mathrm{t}} g_k \leq c_2 \left(L_g \delta_3 + (L_g \delta_3)^{\frac{1}{2}}\right) g_k \leq c_2 \left(L_g \delta_2^{\frac{1}{2}} + L_g^{\frac{1}{2}}\right) \delta_3^{\frac{1}{2}} g_k \leq g_k. \qquad (E.4)$$

Hence, $\epsilon_{k+1} = \min(\epsilon_k, g_{k+1}) = g_{k+1}$. Moreover, since $M_k \leq U_M$, then

$$\|x_{k+1} - x^*\| \overset{(E.1)}{\leq} \frac{2}{\alpha} \left( L_H \delta_3^2 + 2U_M^{\frac{1}{2}} (L_g \delta_3)^{\frac{1}{2}} (1 + L_g) \delta_3 \right)$$

$$\leq \frac{2}{\alpha} \left( L_H + 2U_M^{\frac{1}{2}} L_g^{\frac{1}{2}} (1 + L_g) \right) \delta_3^{\frac{3}{2}} \leq \delta_3.$$

Thus, we know $k + 1 \in \mathcal{I}$. By induction, $k \in \mathcal{I}$ for every $k \geq j_0$, which also gives the convergence of the whole sequence $\{x_k\}$ since Theorem E.2 provides a superlinear convergence with order $\frac{3}{2}$ of the sequence $\{\|x_k - x^*\|\}_{k \geq j_0}$.

Furthermore, the regularizer $\omega_k^{\mathrm{t}}$ reduces to $g_k^{\frac{1}{2}+\theta} g_{k-1}^{-\theta}$ for $k \geq j_0 + 1$ and the premises of Theorem F.3 and Theorem F.4 are satisfied, with the constants $c_0, c,$ and $\nu$ therein chosen as $c_2, c_2,$ and $1$, respectively. Then, the conclusion follows from Theorem F.3 and Theorem F.4. $\qquad \square$

# F   Technical lemmas for local rates

## F.1   Standard properties of the Newton step

This section provides the proofs of Theorems E.2 and E.3, which are the detailed version of Theorem 3.6.

The following lemma is used to show that $\nabla^2 \varphi(x) \succ 0$ in a neighborhood of $x^*$. It can be found in, e.g., Facchinei and Pang [18, Lemma 7.2.12].

**Lemma F.1** (Perturbation lemma). *Let $A, B \in \mathbb{R}^{n \times n}$ with $\|A^{-1}\| \leq \alpha$. If $\|A - B\| \leq \beta$ and $\alpha\beta < 1$, then*

$$\|B^{-1}\| \leq \frac{\alpha}{1 - \alpha\beta}. \qquad (F.1)$$

**Corollary F.2.** *Under Assumption E.1, we have the following properties:*

1. *When $x \in B_{\delta_0}(x^*)$, we know $\nabla^2 \varphi(x) \succeq \frac{\alpha}{2} \mathrm{I}_n$ and $\|(\nabla^2 \varphi(x))^{-1}\| \leq \frac{2}{\alpha}$.*

2. *$\frac{\alpha}{2} \|x - y\| \leq \|\nabla \varphi(x) - \nabla \varphi(y)\| \leq L_g \|x - y\|$ for $x, y \in B_{\delta_0}(x^*)$.*

*Proof.* The first part directly follows from Theorem F.1. Since $\nabla^2 \varphi$ is $L_H$-Lipschitz, then

$$\sup_{x \in B_{\delta_0}(x^*)} \|\nabla^2 \varphi(x)\| \leq \|\nabla^2 \varphi(x^*)\| + L_H \delta_0 = L_g,$$

implying that $\nabla \varphi$ is $L_g$-Lipschitz on $B_{\delta_0}(x^*)$. Then, the second part follows from Nesterov et al. [43, Section 1]. $\qquad \square$

*Proof of Theorem E.2.* From Theorem F.2, we know $H \succeq \frac{\alpha}{2} \mathrm{I}_n$ and $\|H^{-1}\| \leq \frac{2}{\alpha}$ for every $x \in B_\delta(x^*)$ and $H = \nabla^2 \varphi(x)$. Then, let $\epsilon = M^{\frac{1}{2}} \omega$ and note that by the choice in Algorithm 1, $\tilde{\eta} \leq M^{\frac{1}{2}} \omega = \epsilon$, we have

$$\|x^* - (x + d)\| \leq \|(H + 2\epsilon \mathrm{I}_n)^{-1} \nabla \varphi(x) + (x^* - x)\| + \|d + (H + 2\epsilon \mathrm{I}_n)^{-1} \nabla \varphi(x)\|$$

$$\overset{(B.6)}{\leq} \|(H + 2\epsilon \mathrm{I}_n)^{-1}\| \left(\|\nabla \varphi(x) + H(x^* - x)\| + 2\epsilon \|x^* - x\| + \tilde{\eta} \|\nabla \varphi(x)\|\right)$$

$$\leq \frac{2}{\alpha} \left(\|\nabla \varphi(x) + H(x^* - x)\| + 2\epsilon \|x^* - x\| + 2\epsilon \|\nabla \varphi(x)\|\right)$$

$$\overset{(D.1)}{\leq} \frac{2}{\alpha} \left(L_H \|x^* - x\|^2 + 2\epsilon \|x^* - x\| + 2\epsilon \|\nabla \varphi(x)\|\right). \qquad (F.2)$$

From Theorem F.2, we know $\frac{\alpha}{2} \|x - x^*\| \leq \|\nabla \varphi(x)\| \leq L_g \|x - x^*\|$, yielding (E.1).

Furthermore, we have

$$\|\nabla\varphi(x+d)\| \leq L_g\|x^* - (x+d)\| \overset{(F.2)}{\leq} \frac{2L_g}{\alpha}\left(L_H\|x^* - x\|^2 + 2\epsilon\|x^* - x\| + 2\epsilon\|\nabla\varphi(x)\|\right)$$
$$\leq \frac{2L_g}{\alpha}\left(\frac{4L_H}{\alpha^2}\|\nabla\varphi(x)\|^2 + \frac{4+2\alpha}{\alpha}\epsilon\|\nabla\varphi(x)\|\right).$$

$\square$

*Proof of Theorem E.3.* Let $r_k = \|x_k - x^*\|$, the proof is divided to three steps.

**Step 1** We show that $\mathrm{d\_type}_k = \mathrm{SOL}$ for $x_k \in B_{\delta_1}(x^*)$ regardless of whether the trial step or the fallback step is taken. By Theorem F.2, we have $\nabla^2\varphi(x) \succeq \frac{\alpha}{2}I_n$ for $x \in B_{\delta_0}(x^*)$. From Theorem B.2, when the fallback step is taken, then $\mathrm{d\_type}_k = \mathrm{SOL}$. On the other hand, if the trial step is taken, we will also invoke Theorem B.2 as follows. Let $a = (1+2\theta)^{-1} \in (0,1]$, we have

1. When $\omega_k^{\mathrm{t}} = g_k^{\frac{1}{2}}\min(1, g_k^\theta g_{k-1}^{-\theta})$, we know $(\omega_k^{\mathrm{t}})^a \geq g_k^{\frac{1}{2}}U_\varphi^{-a\theta} = \omega_k^{\mathrm{f}}U_\varphi^{-a\theta}$;

2. When $\omega_k^{\mathrm{t}} = \epsilon_k^{\frac{1}{2}+\theta}\epsilon_{k-1}^{-\theta}$, it still holds that $(\omega_k^{\mathrm{t}})^a \geq \omega_k^{\mathrm{f}}U_\varphi^{-a\theta}$.

Therefore, let $\bar{\rho} = \tau\sqrt{M_k}\omega_k^{\mathrm{f}}$ and $\rho = \sqrt{M_k}\omega_k^{\mathrm{t}}$, and note that from Theorem C.1 we have $M_k \leq U_M$, then let $b = \tau U_\varphi^{a\theta}U_M^{\frac{1-a}{2}}$, we know

$$\rho^a = M_k^{\frac{a}{2}}(\omega_k^{\mathrm{t}})^a \geq M_k^{\frac{a}{2}}\omega_k^{\mathrm{f}}U_\varphi^{-a\theta} = \tau^{-1}U_\varphi^{-a\theta}M_k^{\frac{a-1}{2}}\bar{\rho} \overset{(a\leq 1)}{\geq} \tau^{-1}U_\varphi^{-a\theta}U_M^{\frac{a-1}{2}}\bar{\rho} = b^{-1}\bar{\rho}.$$

Since the map $U \mapsto C(\alpha, a, b, U)$ defined in Theorem B.2 is non-increasing, we know

$$\inf_{x\in B_{\delta_0}(x^*)} C(\alpha/2, a, b, \|\nabla^2\varphi(x)\|) \geq C(\alpha/2, a, b, \|\nabla^2\varphi(x^*)\| + L_H\delta_0) =: \tilde{c} > 0.$$

From Theorem F.2, we know for $x_k \in B_{\delta_1}(x^*)$,

$$\rho = \sqrt{M_k}\omega_k^{\mathrm{t}} \leq U_M^{\frac{1}{2}}g_k^{\frac{1}{2}} \leq U_M^{\frac{1}{2}}(L_g\delta_1)^{\frac{1}{2}} \leq \min(\eta, \tilde{c}).$$

Thus, $\mathrm{CappedCG}$ is invoked with $\xi = \rho$ and the premises of the fourth item in Theorem B.2 are satisfied, which leads to $\mathrm{d\_type}_k = \mathrm{SOL}$.

**Step 2** This is a standard step showing that the Newton direction will be taken (see, e.g., Facchinei [17], Facchinei and Pang [18]).

We show that $m_k = 0$ for $x_k \in B_{\delta_2}(x^*)$ regardless of whether the trial step or the fallback step is taken. Define $\omega_k = \omega_k^{\mathrm{t}}$ if the $k$-th step is accepted and $\omega_k = \omega_k^{\mathrm{f}}$ otherwise, and denote $d_k$ as the direction generated in $\mathrm{NewtonStep}$ with such $\omega_k$. By the assumption and Theorem E.2, we have for $x_k \in B_{\delta_1}(x^*)$, it holds that $\omega_k \leq g_k^{\frac{1}{2}} \leq L_g^{\frac{1}{2}}r_k^{\frac{1}{2}}$, and $\sup_{x\in B_{\delta_1}(x^*)}\|\nabla^2\varphi(x)\| \leq L_g$, and

$$\|x_k + d_k - x^*\| \overset{(E.1)}{\leq} \frac{2}{\alpha}\left(L_Hr_k^2 + 2M_k^{\frac{1}{2}}(1+L_g)r_k\omega_k\right) \leq c_1r_k^{\frac{3}{2}}, \tag{F.3}$$

where we have used Theorem C.1 to obtain $M_k \leq U_M$. Using the mean-value theorem and noticing that $\nabla\varphi(x^*) = 0$, there exist $\zeta, \xi \in (0,1)$ and $H_\zeta = \nabla^2\varphi(x^* + \zeta(x_k - x^*))$, $H_\xi = \nabla^2\varphi(x^* + \xi(x_k + d_k - x^*))$ such that for $x_k \in B_{\delta_1}(x^*)$,

$$\varphi(x_k) - \varphi(x^*) = \frac{1}{2}(x_k - x^*)^\top H_\zeta(x_k - x^*),$$

$$\varphi(x_k + d_k) - \varphi(x^*) = \frac{1}{2}(x_k + d_k - x^*)^\top H_\xi(x_k + d_k - x^*) \overset{(F.3)}{\leq} \frac{L_gc_1^2}{2}r_k^3.$$

Combining them, we have for $x_k \in B_{\delta_1}(x^*)$,

$$\varphi(x_k + d_k) - \varphi(x_k) - \frac{1}{2}\nabla\varphi(x_k)^\top d_k$$

$$\leq \frac{L_g c_1^2}{2}r_k^3 - \frac{1}{2}(x_k - x^*)^\top H_\zeta(x_k - x^*) - \frac{1}{2}\nabla\varphi(x_k)^\top d_k$$

$$= \frac{L_g c_1^2}{2}r_k^3 - \frac{1}{2}(x_k + d_k - x^*)^\top H_\zeta(x_k - x^*) - \frac{1}{2}(\nabla\varphi(x_k) - H_\zeta(x_k - x^*))^\top d_k. \tag{F.4}$$

Let $\bar{x} = x^* + \zeta(x_k - x^*)$ and note $\nabla\varphi(x^*) = 0$, then

$$\|\nabla\varphi(x_k) - \zeta^{-1}\nabla\varphi(\bar{x})\| = \|(\nabla\varphi(x_k) - \nabla\varphi(x^*)) - \zeta^{-1}(\nabla\varphi(\bar{x}) - \nabla\varphi(x^*))\|$$

$$= \left\|\int_0^1 \nabla^2\varphi(x^* + t(x_k - x^*))(x_k - x^*)\mathrm{d}t - \zeta^{-1}\int_0^1 \nabla^2\varphi(x^* + t(\bar{x} - x^*))(\bar{x} - x^*)\mathrm{d}t\right\|$$

$$= \left\|\int_0^1 (\nabla^2\varphi(x^* + t(x_k - x^*)) - \nabla^2\varphi(x^* + t(\bar{x} - x^*)))(x_k - x^*)\mathrm{d}t\right\|$$

$$\leq L_H \int_0^1 t\|x_k - \bar{x}\|r_k\mathrm{d}t = L_H \int_0^1 t(1-\zeta)\|x_k - x^*\|r_k\mathrm{d}t \leq L_H r_k^2.$$

Therefore, we have for $x_k \in B_{\delta_1}(x^*)$,

$$\|\nabla\varphi(x_k) - H_\zeta(x_k - x^*)\|$$

$$\leq \left\|\zeta^{-1}\nabla\varphi(\bar{x}) - H_\zeta(x_k - x^*)\right\| + \left\|\zeta^{-1}\nabla\varphi(\bar{x}) - \nabla\varphi(x_k)\right\|$$

$$= \zeta^{-1}\left\|\nabla\varphi(\bar{x}) - \nabla\varphi(x^*) - H_\zeta(\bar{x} - x^*)\right\| + \left\|\zeta^{-1}\nabla\varphi(\bar{x}) - \nabla\varphi(x_k)\right\|$$

$$\leq \zeta^{-1}L_H\|\bar{x} - x^*\|^2 + L_H r_k^2 = (\zeta + 1)L_H r_k^2 \leq 2L_H r_k^2 \leq 2L_H \delta_1^{\frac{1}{2}} r_k^{\frac{3}{2}}. \tag{F.5}$$

We also note that by the definition $\delta_2^{\frac{1}{2}} \leq 1/(2c_1)$. Hence, $1 - c_1\delta_2^{\frac{1}{2}} \geq 1/2$ and for $x_k \in B_{\delta_2}(x^*)$,

$$\|d_k\| \leq \|x_k + d_k - x_*\| + \|x_k - x_*\| \overset{(\mathrm{F.3})}{\leq} c_1 r_k^{\frac{3}{2}} + r_k \leq (c_1\delta_2^{\frac{1}{2}} + 1)r_k \leq 2r_k, \tag{F.6}$$

$$\|d_k\| \geq \|x_k - x_*\| - \|x_k + d_k - x_*\| \overset{(\mathrm{F.3})}{\geq} r_k - c_1 r_k^{\frac{3}{2}} \geq (1 - c_1\delta_2^{\frac{1}{2}})r_k \geq \frac{r_k}{2}. \tag{F.7}$$

Combining the above two inequalities, we find for $x_k \in B_{\delta_2}(x^*)$,

$$|(\nabla\varphi(x_k) - H_\zeta(x_k - x_*))^\top d_k| \overset{(\mathrm{F.5})}{\leq} 4L_H \delta_1^{\frac{1}{2}} r_k^{\frac{5}{2}}, \tag{F.8}$$

$$|(x_k + d_k - x_*)^\top H_\zeta(x_k - x_*)| \overset{(\mathrm{F.3})}{\leq} L_g c_1 r_k^{\frac{5}{2}}. \tag{F.9}$$

Since d_type$_k$ = SOL, then using Theorem B.2 and note that $\nabla^2\varphi(x_k) \succeq \frac{\alpha}{2}\mathrm{I}_n$, we know

$$\nabla\varphi(x_k)^\top d_k \overset{(\mathrm{B.7})}{=} -d_k^\top(\nabla^2\varphi(x_k) + 2M_k^{\frac{1}{2}}\omega_k I)d_k \leq -\frac{\alpha}{2}\|d_k\|^2 \overset{(\mathrm{F.7})}{\leq} -\frac{\alpha}{8}r_k^2.$$

Substituting them back to (F.4), and note that $\mu \in (0, 1/2)$, we have for $x_k \in B_{\delta_2}(x^*)$,

$$\varphi(x_k + d_k) - \varphi(x_k) - \mu\nabla\varphi(x_k)^\top d_k$$

$$\leq \left(\frac{1}{2} - \mu\right)\nabla\varphi(x_k)^\top d_k + \left(\varphi(x_k + d_k) - \varphi(x_k) - \frac{1}{2}\nabla\varphi(x_k)^\top d_k\right)$$

$$\leq -\left(\frac{1}{2} - \mu\right)\frac{\alpha}{8}r_k^2 + \frac{1}{2}\left(L_g c_1^2\delta_1^{\frac{1}{2}} + L_g c_1 + 4L_H\delta_1^{\frac{1}{2}}\right)r_k^{\frac{5}{2}}.$$

We can see that the above term is negative as long as $r_k \leq \delta_2$, and therefore, the linesearch (2.4) holds with $m_k = 0$.

**Step 3** We show that the trial step (i.e., the step with using $\omega_k^\mathrm{t}$) is accepted. Since d_type$_k$ = SOL, then NewtonStep will not return a FAIL state, so it suffices to show $g_{k+\frac{1}{2}} = \|\nabla\varphi(x_k + d_k)\| \leq g_k$, where $d_k$ is the direction generated by NewtonStep with $\omega = \omega_k^\mathrm{t} \leq \sqrt{g_k}$. Then, by Theorem E.2 and (E.4) we have $g_{x+\frac{1}{2}} \leq g_k$ for $x_k \in B_{\delta_3}(x^*)$. □

## F.2 Local rate boosting lemma

In this section, we establish a generalized version of Theorem 3.7 in Theorem F.3 and Theorem F.4, which extends to the case of a $\nu$-Hölder continuous Hessian and reduces the Lipschitz Hessian in Assumption 2.1 when $\nu = 1$. The results in Theorem F.3 primarily characterize the behavior for $\theta \in [0, \nu]$, while the case of $\theta > \nu$ is analyzed separately in Theorem F.4. This division into two cases is mainly a technical necessity, as merging them could result in the preleading coefficient $c_k$ in (F.12) becoming unbounded.

**Lemma F.3.** *Let $\{g_k\}_{k \geq 0} \subseteq (0, \infty)$, $c_0 \geq 1$, $c \geq 1$, $1 \geq \nu > 0$, $\nu_0 = \bar{\nu} := \frac{\nu}{1+\nu}$, and $\theta \geq 0$. If $\log g_1 \leq \log c_0 + (1 + \nu_0) \log g_0$ and the following inequality holds for $k \geq 1$,*

$$g_{k+1} \leq c g_k^{1+\nu} + c g_k^{1+\bar{\nu}} \frac{g_k^\theta}{g_{k-1}^\theta}, \tag{F.10}$$

*and $g_0 \leq \min\left(1, (2c)^{-\frac{1}{\bar{\nu}}}, c_0^{-\frac{1}{\bar{\nu}}}\right)$, then we have $g_{k+1} \leq g_k$ and the following inequality holds for every $k \geq 0$:*

$$\log g_{k+1} \leq \log c_k + (1 + \nu_k) \log g_k, \tag{F.11}$$

*where we define $\bar{\theta} = \min(\theta, \nu)$ and $\nu_\infty = -\frac{1}{2}(1 - \bar{\nu} - \bar{\theta}) + \frac{1}{2}\sqrt{(1 - \bar{\nu} - \bar{\theta})^2 + 4\bar{\nu}} \in [\bar{\nu}, \nu]$ is the positive root of the equation $\bar{\nu} + \frac{\bar{\theta}\nu_\infty}{1+\nu_\infty} = \nu_\infty$, and[4]*

$$\log c_k := \log(2c) + \frac{\bar{\theta}}{1 + \nu_{k-1}} \log c_{k-1} \leq \left(1 + \frac{1}{\bar{\nu}}\right) \log(2c) + \log c_0, \tag{F.12}$$

$$\nu_k := \min\left(\nu, \bar{\nu} + \frac{\bar{\theta}\nu_{k-1}}{1 + \nu_{k-1}}\right) \geq \nu_\infty - \frac{\bar{\theta}^k(\nu_\infty - \bar{\nu})}{(1 + \bar{\nu})^{2k}} \geq \nu_\infty - \frac{\bar{\theta}^k}{(1 + \bar{\nu})^{2k}}. \tag{F.13}$$

*In particular, when $\theta \geq \nu$, we have $\nu_\infty = \nu$ and $v_k \geq \nu - \frac{\nu^k(\nu - \bar{\nu})}{(1+\bar{\nu})^{2k}}$.*

*Proof.* We first show that $\nu_\infty \in [\bar{\nu}, \nu]$. Define the map $T(\alpha) = \bar{\nu} + \frac{\bar{\theta}\alpha}{1+\alpha} - \alpha$ for $\alpha \in [\bar{\nu}, \nu]$. By reformulating it as $T(\alpha) = \bar{\nu} + \bar{\theta} + 1 - \left(\frac{\bar{\theta}}{1+\alpha} + (1 + \alpha)\right)$, we see that $T$ is strictly decreasing whenever $1 + \alpha \geq \sqrt{\bar{\theta}}$, which holds since $1 + \alpha \geq 1 + \bar{\nu} > 1 \geq \nu \geq \bar{\theta}$. Then, there exists a unique $\nu_\infty \in [\bar{\nu}, \nu]$ such that $T(\nu_\infty) = 0$ because $T(\bar{\nu}) = \frac{\bar{\theta}\bar{\nu}}{1+\bar{\nu}} \geq 0$ and $T(\nu) = \frac{\nu(\bar{\theta}-\nu)}{1+\nu} \leq 0$.

Let $\mathcal{I} \subseteq \mathbb{N}$ be the set such that $k \in \mathcal{I}$ if and only if

$$g_{k+1} \leq g_k, c_k \geq 1, \nu_k \leq \nu_\infty, \text{ and (F.11), (F.13) hold,}$$

$$\text{and } \log c_k \leq \frac{1 - (1 + \bar{\nu})^{-k}}{1 - (1 + \bar{\nu})^{-1}} \log(2c) + \log c_0.$$

First, we show that $0 \in \mathcal{I}$. Since $\nu_0 = \bar{\nu}$ and $g_0^{\bar{\nu}} \leq c_0^{-1}$, we have $g_1 \leq c_0 g_0^{1+\bar{\nu}} \leq g_0$. The other parts hold by assumption, and we have used $\nu_\infty \geq \bar{\nu}$ and the definition that $\nu_{-1} = 0$ in (F.13) for $k = 0$.

Next, we prove $\mathcal{I} = \mathbb{N}$ by induction. Suppose $0, \ldots, j - 1 \in \mathcal{I}$ for some $j \geq 1$, we will show that $j \in \mathcal{I}$. Since $j - 1 \in \mathcal{I}$, from (F.11) we have $g_j \leq c_{j-1} g_{j-1}^{1+\nu_{j-1}}$, and equivalently, $g_{j-1}^{-1} \leq \left(c_{j-1}^{-1} g_j\right)^{-\frac{1}{1+\nu_{j-1}}}$. Note that $c_{j-1} \geq 1$ and $g_j \leq g_{j-1}$, and $\frac{g_j^\theta}{g_{j-1}^\theta} \leq \frac{g_j^{\bar{\theta}}}{g_{j-1}^{\bar{\theta}}}$ for $\theta \geq \bar{\theta}$, we have

$$g_{j+1} \overset{(F.10)}{\leq} c g_j^{1+\nu} + c g_j^{1+\bar{\nu}} \frac{g_j^{\bar{\theta}}}{g_{j-1}^{\bar{\theta}}} \leq c g_j^{1+\nu} + c c_{j-1}^{\frac{\bar{\theta}}{1+\nu_{j-1}}} g_j^{1+\bar{\nu}+\frac{\bar{\theta}\nu_{j-1}}{1+\nu_{j-1}}}$$

$$\overset{(c, c_{j-1} \geq 1)}{\leq} 2 c c_{j-1}^{\frac{\bar{\theta}}{1+\nu_{j-1}}} \max\left(g_j^{1+\nu}, g_j^{1+\bar{\nu}+\frac{\bar{\theta}\nu_{j-1}}{1+\nu_{j-1}}}\right).$$

---

[4]We define $\nu_{-1} = 0$.

Therefore, we find that

$$\log g_{j+1} \le \underbrace{\log(2c) + \frac{\bar\theta}{1+\nu_{j-1}}\log c_{j-1}}_{\log c_j} + \underbrace{\min\left(1+\nu, 1+\bar\nu + \frac{\bar\theta\nu_{j-1}}{1+\nu_{j-1}}\right)\log g_j}_{1+\nu_j}. \qquad (\text{F.14})$$

Thus, (F.11) holds for $k = j$, and $\log c_j \ge \log(2c) \ge \log 2 \ge 0$, i.e., $c_j \ge 1$.

Since $[j-1] \subseteq \mathcal{I}$, we know $\{g_i\}_{0\le i\le j}$ is non-increasing, $g_j^{\bar\nu} \le g_0^{\bar\nu} \le (2c)^{-1}$, and $g_j \le g_{j-1}$. Note that $\bar\nu \le \nu$ and $g_j \le g_0 \le 1$, then $g_{j+1} \le cg_j^{1+\nu} + cg_j^{1+\bar\nu}(g_j g_{j-1}^{-1})^\theta \le 2cg_j^{1+\bar\nu} \le g_j$.

By (F.13), $\nu_{j-1} \ge \min(\bar\nu, \nu) = \bar\nu$ and we have

$$\log c_j \le \log(2c) + \frac{\bar\theta}{1+\bar\nu}\log c_{j-1}$$
$$\overset{(\bar\theta \le 1)}{\le} \log(2c) + \frac{1}{1+\bar\nu}\left(\frac{1-(1+\bar\nu)^{-(j-1)}}{1-(1+\bar\nu)^{-1}}\log(2c) + \log c_0\right)$$
$$\le \frac{1-(1+\bar\nu)^{-j}}{1-(1+\bar\nu)^{-1}}\log(2c) + \log c_0.$$

Finally, we show $\nu_j \le \nu_\infty$ and (F.13) holds for $k = j$. Define the map $F(\alpha) = \bar\nu + \frac{\bar\theta\alpha}{1+\alpha}$. We know $F(\alpha)$ is non-decreasing for $\alpha > 0$, and $F(\nu_\infty) = \nu_\infty$ by its definition. Since $\nu_{j-1} \le \nu_\infty$ and $F(\nu_{j-1}) \le F(\nu_\infty) = \nu_\infty \le \nu$, then $\nu_j = \min(\nu, F(\nu_{j-1})) = F(\nu_{j-1}) \le \nu_\infty$. Moreover, we have

$$0 \le \nu_\infty - \nu_j = F(\nu_\infty) - F(\nu_{j-1}) = \frac{\bar\theta(\nu_\infty - \nu_{j-1})}{(1+\nu_\infty)(1+\nu_{j-1})}$$
$$\le \frac{\bar\theta(\nu_\infty - \nu_{j-1})}{(1+\bar\nu)^2} \le \frac{\bar\theta^j(\nu_\infty - \bar\nu)}{(1+\bar\nu)^{2j}},$$

where the last inequality follows from the induction assumption.

Thus, we have $j \in \mathcal{I}$ and by induction $\mathcal{I} = \mathbb{N}$. $\qquad\square$

**Corollary F.4.** *Under the assumptions of Theorem F.3, if $\theta > \nu$ and $k \ge k_0 := \frac{\log\frac{\theta-\nu\bar\nu}{\theta-\nu} - \log\nu}{2\log(1+\bar\nu) - \log\nu} + 1$, then $g_k$ converges superlinearly with order $1+\nu$:*

$$\log g_k \le \left(1 + \theta + \frac{1}{\bar\nu}\right)\log(2c) + \theta\log c_0 + (1+\nu)\log g_{k-1}. \qquad (\text{F.15})$$

*Proof.* Since the assumptions are the same as those in Theorem F.3, the results therein are all valid. Furthermore, we note that in the proof of Theorem F.3, the following stronger variant of (F.14) can be obtained from (F.10):

$$\log g_{j+1} \le \underbrace{\log(2c) + \frac{\theta}{1+\nu_{j-1}}\log c_{j-1}}_{\hat c_j} + \underbrace{\min\left(1+\nu, 1+\bar\nu + \frac{\theta\nu_{j-1}}{1+\nu_{j-1}}\right)\log g_j}_{1+\hat\nu_j}. \qquad (\text{F.16})$$

Let $\alpha = \left(\frac{\theta}{\nu-\bar\nu} - 1\right)^{-1} = \left(\frac{\theta}{\nu\bar\nu} - 1\right)^{-1}$. Since $\theta > \nu$, then $\alpha > 0$ and $\frac{1}{\alpha} = \frac{\theta}{\nu\bar\nu} - 1 > \frac{1}{\bar\nu} - 1 = \frac{1}{\nu}$, i.e., $\alpha \in (0, \nu)$. When $\nu_{k-1} \ge \alpha$, we have

$$\hat\nu_k = \min\left(\nu, \bar\nu + \frac{\theta\nu_{k-1}}{1+\nu_{k-1}}\right) = \min\left(\nu, \bar\nu + \frac{\theta}{\nu_{k-1}^{-1}+1}\right)$$
$$\ge \min\left(\nu, \bar\nu + \frac{\theta}{\alpha^{-1}+1}\right) = \nu.$$

From Theorem F.3, we know $\nu_\infty = \nu$, and when $k - 1 \geq k_0 - 1 \geq \log_{\frac{\nu}{(1+\bar{\nu})^2}}(\nu - \alpha) = \frac{-\log(\nu - \alpha)}{2\log(1+\bar{\nu}) - \log \nu}$, the following inequality holds since $\nu \in (0, 1]$ and $1 + \bar{\nu} > 1$.

$$\nu_{k-1} \overset{(F.13)}{\geq} \nu - \frac{\nu^{k-1}(\nu - \bar{\nu})}{(1+\bar{\nu})^{2(k-1)}} \geq \nu - \frac{\nu^{k-1}}{(1+\bar{\nu})^{2(k-1)}} \geq \alpha.$$

Thus, for any $k \geq k_0$, we have $\hat{\nu}_j = \nu$, and

$$\log g_k \overset{(F.16)}{\leq} \log(2c) + \theta \log c_{k-1} + (1 + \nu) \log g_{k-1}$$
$$\overset{(F.12)}{\leq} \left(1 + \theta + \frac{1}{\bar{\nu}}\right) \log(2c) + \theta \log c_0 + (1 + \nu) \log g_{k-1}.$$

Finally, the proof is completed by noticing that $\nu - \alpha = \nu - \frac{\nu\bar{\nu}}{\theta - \nu\bar{\nu}} = \frac{\nu(\theta - \nu)}{\theta - \nu\bar{\nu}}$. $\qquad\square$

# G   Additional numerical results on the CUTEst benchmark

This section provides a detailed description of the experimental setup and additional results on the CUTEst benchmark to supplement Section 4. We implement our algorithm in MATLAB R2023a and denote the variant using the first regularizer in Theorem 2.2 as $\mathbf{ARNCG}_g$, and the variant using the second regularizer as $\mathbf{ARNCG}_\epsilon$. We use the official Julia implementation provided by Hamad and Hinder [26] for their method $\mathbf{CAT}$[5] and Dussault et al. [16]'s official implementation for their method $\mathbf{ARC}_q\mathbf{K}$[6]. As the code for $\mathbf{AN2CER}$ is not publicly available, we investigate several ways to implement it in MATLAB and report the best results, as detailed in Section G.1.

Our experimental settings follow those described by Hamad and Hinder [26], we conduct all experiments in a single-threaded environment on a machine running Ubuntu Server 22.04, equipped with dual-socket Intel(R) Xeon(R) Silver 4210 CPUs and 192 GB of RAM. Each socket is installed with three 32 GB RAM modules, running at 2400 MHz. The algorithm is considered successful if it terminates when $\epsilon_k \leq \epsilon = 10^{-5}$ such that $k \leq 10^5$. If the algorithm fails to terminate within 5 hours, it is also recorded as a failure.

We evaluate these algorithms using the standard CUTEst benchmark for nonlinear optimization [22]. Specifically, we consider all unconstrained problems with more than 100 variables that are commonly available through the Julia and MATLAB interfaces[7] of this benchmark, comprising a total of 124 problems. The dimensions of these problems range from 100 to 123200.

## G.1   Implementation details

$\mathbf{ARNCG}$   The initial point for each problem is provided by the benchmark itself. Other parameters of Algorithm 1 are set as follows:

$$\mu = 0.3, \beta = 0.5, \tau_- = 0.3, \tau = \tau_+ = 1.0, \gamma = 5, M_0 = 1 \text{ and } \eta = 0.01.$$

We consider two choices for $m_{\max}$:

1. Setting $m_{\max} = 1$ so that at most 4 function evaluations per each iteration.
2. Setting $m_{\max} = \lfloor \log_\beta 10^{-8} \rfloor$ to be the smallest integer such that $\beta^{m_{\max}+1} > 10^{-8}$.

In our experiments, we find that $m_{\max} = 1$ works well, and the algorithm is not sensitive to the above parameters, so we do not perform further fine-tuning. In the implementation of `CappedCG`, we do not keep the historical iterations to save memory. Instead, we evaluate (B.1) by regenerating the iterations. In practice, we observe that step (B.1) is triggered very infrequently, resulting in minimal computational overhead. The `TERM` state is primarily designed to ensure theoretical guarantees

---

[5]See `https://github.com/fadihamad94/CAT-Journal`.
[6]See the `ARCqKOp` method in `https://github.com/JuliaSmoothOptimizers/AdaptiveRegularization.jl`.
[7]See `https://github.com/JuliaSmoothOptimizers/CUTEst.jl` for the Julia interface, and `https://github.com/matcutest/matcutest` for the MATLAB interface.

for Hessian-vector products in Section C.3, and we find it is not triggered in practice. Since the termination condition of `CappedCG` using the error $\|r_k\| \leq \hat{\xi}\|r_0\|$ may not be appropriate for a large $\|r_0\|$, we instead require it to satisfy $\|r_k\| \leq \min(\hat{\xi}\|r_0\|, 0.01)$.

The fallback step in the main loop of Algorithm 1 is mainly designed for theoretical considerations, as described in Theorem 3.1. It ensures that an abrupt increase in the gradient norm followed by a sudden drop does not compromise the validity of this lemma but results in a wasted iteration. However, we note that this condition can be relaxed to the following to enhance practical performance:

$$\lambda g_{k+\frac{1}{2}} > g_k \text{ and } g_k \leq \lambda g_{k-1}, \text{ for } \lambda \in (0, 1]. \tag{G.1}$$

When $\lambda = 1$, this condition reduces to the original one. In our experiments, we explore the choices of $\lambda = 1$, $\lambda = 0.01$, and the impact of removing the fallback step (i.e., $\lambda = 0$). Moreover, we note that when $\theta = 0$, the fallback step and the trial step are identical so the choices of $\lambda$ do not affect the results. In practice, we suggest setting a small $\lambda$ or removing the fallback step.

We also terminate the algorithm and *mark it as a failure* if both the function value and gradient norm remain unchanged for 20 iterations or if the current search direction satisfies $\|d_k\| \leq 2 \times 10^{-16}$, or if the Lipschitz constant estimation satisfies $M_k \geq 10^{40}$, as these scenarios may indicate numerical issues. Figure 2 in the main text is generated under the above settings with $\lambda = 0$ and $m_{\max} = 1$.

For the Hessian evaluations, we only access it through the Hessian-vector products, and count the evaluation number as the number of iterations minus the number of the linesearch failures. Since when a linesearch failure occurs, the next point is the same as the current point and does not increase the oracle complexity of Hessian evaluations.

**AN2CER**   Our implementation follows the algorithm described in Gratton et al. [24, Section 2], with parameters adopted from their suggested values. The algorithm first attempts to solve the regularized Newton equation using the regularizer $\sqrt{\kappa_a M_k g_k}$. If this attempt fails, the minimal eigenvalue $\lambda_{\min}(\nabla^2\varphi(x_k))$ is computed. The algorithm then switches to the regularizer $\sqrt{M_k g_k} + [-\lambda_{\min}(\nabla^2\varphi(x_k))]_+$ when $\lambda_{\min}(\nabla^2\varphi(x_k)) > \kappa_C\sqrt{M_k g_k}$, and directly uses the corresponding eigenvector otherwise.

In AN2CER, the authors suggest using Cholesky factorization to solve the Newton equation and invoking the full eigendecomposition (i.e., the `eig` function in MATLAB) to find the minimal eigenvalue when the factorization fails. We observe that, in the current benchmark, it is more efficient to use `CappedCG` as the equation solver and compute the minimal eigenvalue using MATLAB's `eigs` function when `NC` is returned. This modification preserves the success rate and oracle evaluations of the original implementation while significantly reducing computational cost. We also note that there are several variants of AN2CER in Gratton et al. [24], and we find that the current version yields the best results among them.

## G.2   Results on the CUTEst benchmark

Following Hamad and Hinder [26], we report the shifted geometric mean[8] of Hessian, gradient and function evaluations, as well as the elapsed time in Tables 3 and 5. In our algorithm, we define normalized Hessian-vector products as the original products divided by the problem dimension $n$, which can be interpreted as the fraction of information about the Hessian that is revealed to the algorithm; the linesearch failure rate is the fraction of iterations that exceed the maximum allowed steps $m_{\max}$; and the second linesearch rate measures the fraction of times the linesearch rule (D.12) is invoked. The medians of these metrics are provided in Tables 4 and 6. The success rate as a function of oracle evaluations is plotted in Figures 5 and 6. When an algorithm fails, the elapsed time is recorded as twice the time limit (i.e., 10 hours), and the oracle evaluations are recorded as twice the iteration limit (i.e., $2 \times 10^5$). We note that the choices for handling failure cases in the reported metrics of these tables may affect the relative comparison of results with different success rates, although they follow the convention from previous works. Therefore, we suggest that readers also focus on the figures for a detailed analysis of each algorithm's behavior.

---

[8]For a dataset $\{a_i\}_{i \in [k]}$, the shifted geometric mean is defined as $\exp\left(\frac{1}{k}\sum_{i=1}^{k}\log(a_i + 1)\right)$, which accounts for cases where $a_i = 0$.

**The fallback parameter**   From Tables 3 and 4 and Figure 5, we observe that the choice of the fallback parameter $\lambda$ in (G.1) does not significantly affect the success rate, and the overall performance remains similar across different values of $\lambda$. For larger $\lambda$, the fallback step is generally triggered more frequently (as indicated by the "fallback rate"), leading to increased computational time and oracle evaluations. Interestingly, ARNCG$_\epsilon$ with $m_{\max} = 1$ seems an exception that $\lambda = 1$ is beneficial for specific problems and gives a slightly higher success rate.

**The regularization coefficients**   Tables 5 and 6 and Figure 6 present comparisons for different values of $\theta$. As $\theta$ increases, the performance initially improves but then declines. Larger $\theta$ imposes stricter tolerance requirements on `CappedCG` (as indicated by the number of Hessian-vector products in these tables), and increases computational costs, while smaller $\theta$ may lead to a slower local convergence. Thus, we recommend choosing $\theta \in [0.5, 1]$ to balance computational efficiency and local behavior.

We also note that this tolerance requirement is designed for local convergence and is not necessary for global complexity, so there may be room for improvement. For example, we can use a fixed tolerance $\eta$ when the current gradient norm is larger than a threshold, and switch to the current choice $\min(\eta, \sqrt{M_k}\omega_k)$ otherwise. We leave this for future exploration.

Although ARNCG$_g$ has a slightly higher worst-case complexity (by a double-logarithmic factor) than ARNCG$_\epsilon$, they exhibit similar empirical performance, and in some cases, ARNCG$_g$ even performs better.

A potential failure case *in practice* for ARNCG$_\epsilon$ occurs when the iteration enters a neighborhood with a small gradient norm and then escapes via a negative curvature direction. Consequently, $\epsilon_k$ stays small while $g_k$ may grow large, making the method resemble the fixed $\epsilon$ scenario. Interestingly, this same condition is also what introduces the logarithmic factor in ARNCG$_g$ *theoretically*.

**The linesearch parameter**   Since our algorithm relies on a linesearch step, it requires more function evaluations than CAT for large $m_{\max}$. If evaluating the target function is expensive, we may need to set a small $m_{\max}$, or even $m_{\max} = 0$. Under the latter case, at most two tests of the line search criteria are performed, and the parameter $M_k$ is increased when these tests fail. Our theory guarantees that $M_k = O(L_H)$, so this choice remains valid. In practice, we observe that using a relatively small $m_{\max}$ gives better results.

**Case studies for local behavior**   We present two benchmark problems that exhibit superlinear local convergence behavior. As illustrated in Figure 4, a larger $\theta$ gives faster local convergence. We only show the algorithm using the second regularizer in this figure, and note that the two regularizers have a similar behavior since in the local regime they reduce to $g_k^{\frac{1}{2}+\theta} g_{k-1}^{-\theta}$, as shown in the last paragraph of the proof of Theorem E.4. Generally, it is hard to identify when the algorithm enters the neighborhood for superlinear convergence. For `HIMMELBG`, the algorithm appears to be initialized near the local regime. For `ROSENBR`, the algorithm enters the local regime after approximately 20 iterations.

# H   Additional numerical results on physics-informed neural networks

This section provides a detailed description of the experimental setup and additional results on PINNs to supplement Section 4. Our experimental settings follow those described by Rathore et al. [47], and the code is adopted from their codebase, developed with Python 3.10.12. All experiments are conducted on NVIDIA P100 GPUs with 16 GB of VRAM.

## H.1   Problem setup

For a given domain $\Omega \subset \mathbb{R}^n$, we can define the following partial differential equation (PDE):

$$\begin{cases} \mathcal{D}u = 0, & x \in \Omega, \\ \mathcal{B}u = 0, & x \in \partial\Omega, \end{cases} \tag{H.1}$$

where $u$ denotes the solution to the equation, $\mathcal{D}$ is a differential operator, and $\mathcal{B}$ represents the boundary or initial condition operator. PINNs approximate the solution of the above PDE using a

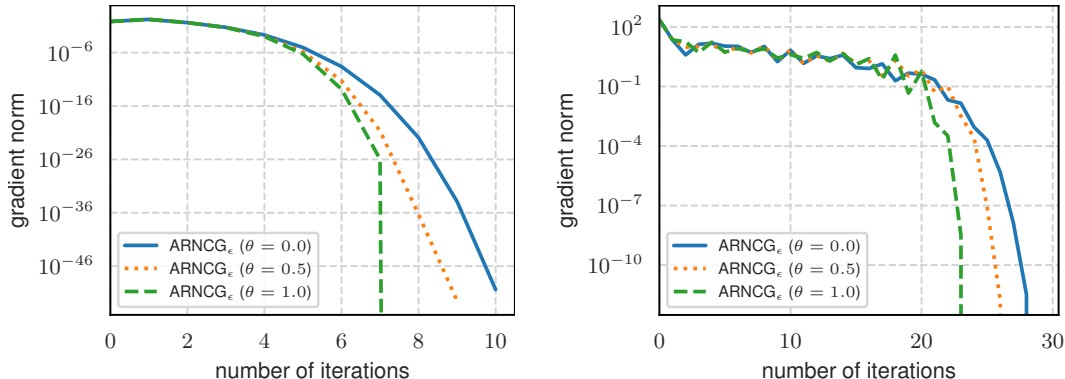

Figure 4: Illustration of the local behavior of our method on the `HIMMELBG` (left plot) and `ROSENBR` (right plot) problems from the CUTEst benchmark for $\lambda = 0$ and $m_{\max} = 1$. All methods converge to the same point.

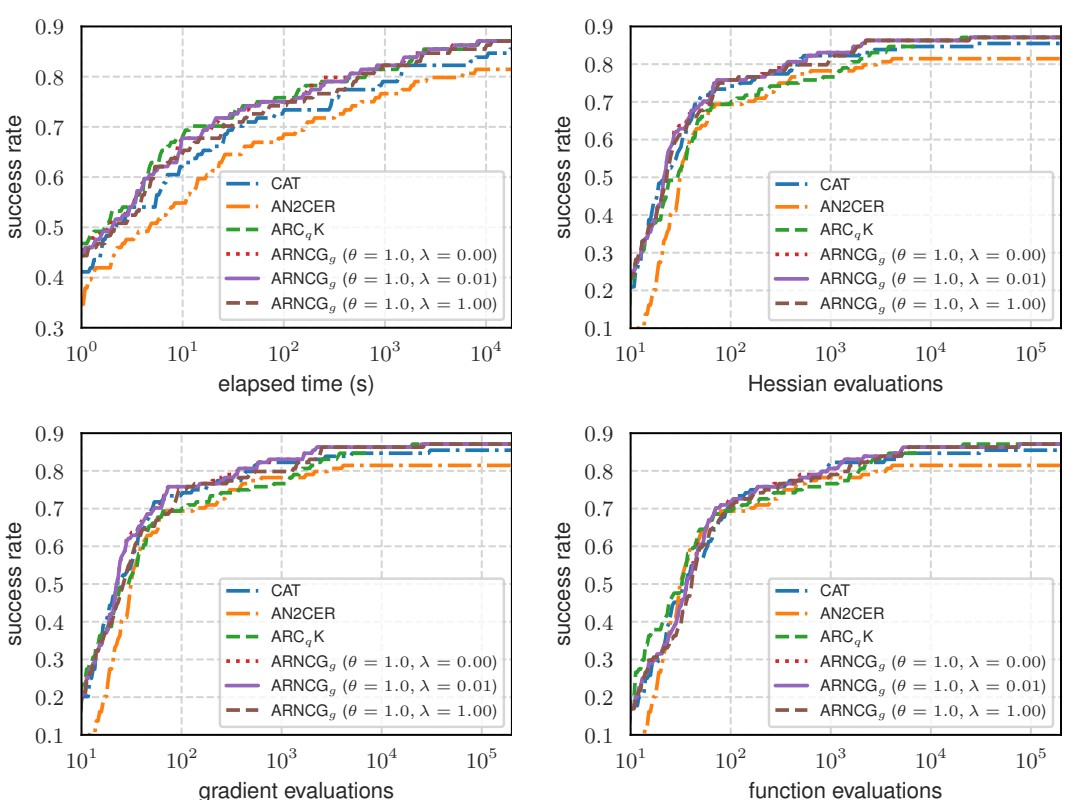

Figure 5: Comparison of success rates as functions of elapsed time, Hessian evaluations, gradient evaluations and function evaluations for solving problems in the CUTEst benchmark. The fallback parameter $\lambda$ in (G.1) varies, and $m_{\max} = 1$.

Table 3: Shifted geometric mean of the relevant metrics for different methods in the CUTEst benchmark. The fallback, second linesearch and linesearch failure rates are reported as mean values. The fallback parameter $\lambda$ in (G.1) varies.

| | Elapsed Time (s) | Hessian Evaluations | Gradient Evaluations | Function Evaluations | Hessian-vector Products (normalzied) | Success Rate (%) | Linesearch Failure Rate (%) | Second Linesearch Rate (%) | Fallback Rate (%) |
|---|---|---|---|---|---|---|---|---|---|
| AN2CER | 36.70 | 170.10 | 172.02 | 176.80 | 31.38 | 81.45 | N/A | N/A | N/A |
| CAT | 23.34 | 88.47 | 96.61 | 125.56 | N/A | 85.48 | N/A | N/A | N/A |
| ARC$_q$K | 16.16 | 113.21 | 113.84 | 119.51 | 11.97 | 87.10 | N/A | N/A | N/A |
| Results for $m_{\max} = 1$ and $\theta = 1.0$ | | | | | | | | | |
| ARNCG$_g$ ($\lambda = 0.00$) | 16.71 | 80.86 | 86.41 | 119.51 | 13.77 | 87.10 | 16.08 | 1.38 | 0.00 |
| ARNCG$_g$ ($\lambda = 0.01$) | 17.01 | 81.46 | 87.31 | 120.48 | 13.90 | 87.10 | 15.98 | 1.31 | 0.33 |
| ARNCG$_g$ ($\lambda = 1.00$) | 19.02 | 85.61 | 99.01 | 130.91 | 14.84 | 87.10 | 14.52 | 0.17 | 7.43 |
| ARNCG$_\epsilon$ ($\lambda = 0.00$) | 18.28 | 85.03 | 90.78 | 125.29 | 14.91 | 86.29 | 16.89 | 0.43 | 0.00 |
| ARNCG$_\epsilon$ ($\lambda = 0.01$) | 18.39 | 85.03 | 90.78 | 125.29 | 14.91 | 86.29 | 16.89 | 0.43 | 0.00 |
| ARNCG$_\epsilon$ ($\lambda = 1.00$) | 18.04 | 78.40 | 89.41 | 122.41 | 14.22 | 87.10 | 16.03 | 0.46 | 6.10 |
| Results for $m_{\max} = \lfloor \log_\beta 10^{-8} \rfloor$ and $\theta = 1.0$ | | | | | | | | | |
| ARNCG$_g$ ($\lambda = 0.00$) | 22.89 | 113.82 | 121.08 | 184.09 | 19.14 | 83.87 | 0.08 | 0.00 | 0.00 |
| ARNCG$_g$ ($\lambda = 0.01$) | 23.81 | 117.02 | 125.50 | 189.01 | 19.77 | 83.87 | 0.08 | 0.00 | 0.90 |
| ARNCG$_g$ ($\lambda = 1.00$) | 26.68 | 125.53 | 147.89 | 218.05 | 22.53 | 83.87 | 0.08 | 0.00 | 11.43 |
| ARNCG$_\epsilon$ ($\lambda = 0.00$) | 22.58 | 105.95 | 112.68 | 176.50 | 17.81 | 84.68 | 0.10 | 0.00 | 0.00 |
| ARNCG$_\epsilon$ ($\lambda = 0.01$) | 22.47 | 105.95 | 112.68 | 176.50 | 17.81 | 84.68 | 0.10 | 0.00 | 0.00 |
| ARNCG$_\epsilon$ ($\lambda = 1.00$) | 25.80 | 118.41 | 137.31 | 214.58 | 20.79 | 83.06 | 0.29 | 0.00 | 9.94 |

Table 4: Median of the relevant metrics for different methods in the CUTEst benchmark. The fallback parameter $\lambda$ in (G.1) varies.

| | Elapsed Time (s) | Hessian Evaluations | Gradient Evaluations | Function Evaluations | Hessian-vector Products (normalzied) | Success Rate (%) | Linesearch Failure Rate (%) | Second Linesearch Rate (%) | Fallback Rate (%) |
|---|---|---|---|---|---|---|---|---|---|
| AN2CER | 4.75 | 30.00 | 30.00 | 30.00 | 4.24 | 81.45 | N/A | N/A | N/A |
| CAT | 2.13 | 21.00 | 22.00 | 34.50 | N/A | 85.48 | N/A | N/A | N/A |
| ARC$_q$K | 1.71 | 28.50 | 28.50 | 32.00 | 0.62 | 87.10 | N/A | N/A | N/A |
| Results for $m_{\max} = 1$ and $\theta = 1.0$ | | | | | | | | | |
| ARNCG$_g$ ($\lambda = 0.00$) | 1.89 | 20.50 | 21.50 | 35.50 | 1.52 | 87.10 | 10.82 | 0.00 | 0.00 |
| ARNCG$_g$ ($\lambda = 0.01$) | 2.00 | 20.50 | 21.50 | 35.50 | 1.52 | 87.10 | 10.70 | 0.00 | 0.00 |
| ARNCG$_g$ ($\lambda = 1.00$) | 2.12 | 22.00 | 25.50 | 40.00 | 1.92 | 87.10 | 6.75 | 0.00 | 0.00 |
| ARNCG$_\epsilon$ ($\lambda = 0.00$) | 1.72 | 21.50 | 22.50 | 38.00 | 1.62 | 86.29 | 10.26 | 0.00 | 0.00 |
| ARNCG$_\epsilon$ ($\lambda = 0.01$) | 1.86 | 21.50 | 22.50 | 38.00 | 1.62 | 86.29 | 10.26 | 0.00 | 0.00 |
| ARNCG$_\epsilon$ ($\lambda = 1.00$) | 1.99 | 21.00 | 24.50 | 38.00 | 2.01 | 87.10 | 9.92 | 0.00 | 0.00 |
| Results for $m_{\max} = \lfloor \log_\beta 10^{-8} \rfloor$ and $\theta = 1.0$ | | | | | | | | | |
| ARNCG$_g$ ($\lambda = 0.00$) | 2.84 | 25.00 | 26.00 | 53.00 | 2.13 | 83.87 | 0.00 | 0.00 | 0.00 |
| ARNCG$_g$ ($\lambda = 0.01$) | 2.89 | 25.00 | 26.00 | 53.00 | 2.34 | 83.87 | 0.00 | 0.00 | 0.00 |
| ARNCG$_g$ ($\lambda = 1.00$) | 3.28 | 24.00 | 30.50 | 61.50 | 2.34 | 83.87 | 0.00 | 0.00 | 9.09 |
| ARNCG$_\epsilon$ ($\lambda = 0.00$) | 2.49 | 26.00 | 27.00 | 55.50 | 1.40 | 84.68 | 0.00 | 0.00 | 0.00 |
| ARNCG$_\epsilon$ ($\lambda = 0.01$) | 2.44 | 26.00 | 27.00 | 55.50 | 1.40 | 84.68 | 0.00 | 0.00 | 0.00 |
| ARNCG$_\epsilon$ ($\lambda = 1.00$) | 2.90 | 25.00 | 30.50 | 69.00 | 1.68 | 83.06 | 0.00 | 0.00 | 8.33 |

neural network $f_\theta$ parameterized by $\theta$, which is trained on the following residual-based loss function:

$$\varphi(\theta) = \frac{1}{n_{\text{res}}} \sum_{i=1}^{n_{\text{res}}} (\mathcal{D} f_\theta(x_r^i))^2 + \frac{1}{n_{\text{bc}}} \sum_{i=1}^{n_{\text{bc}}} (\mathcal{B} f_\theta(x_b^i))^2, \tag{H.2}$$

where $\{x_r^i\}_{i=1}^{n_{\text{res}}} \subseteq \Omega$ and $\{x_b^i\}_{i=1}^{n_{\text{bc}}} \subseteq \partial\Omega$ denote points sampled from the interior and boundary of the domain, respectively.

Following Rathore et al. [47], the solution to the PDE is approximated using a fully connected neural network $f_\theta$ with width $200$ and $3$ hidden layers, comprising a total of $81201$ parameters in double precision. The activation function is set to $\tanh$, and Xavier initialization is applied [21]. The training data consist of $n_{\text{res}} = 10^4$ points uniformly sampled from a mesh over the domain, where the mesh contains $101$ and $257$ uniformly spaced points along the $t$-axis and $x$-axis, respectively. The test data contains all points in this mesh. Since Rathore et al. [47] adopted the randomized Nyström method [20] to construct preconditioners and accelerate the CG computation, we also incorporate it to ensure a fair comparison.

We consider the three types of problems for training PINNs as in Rathore et al. [47]. Their specific forms are given below:

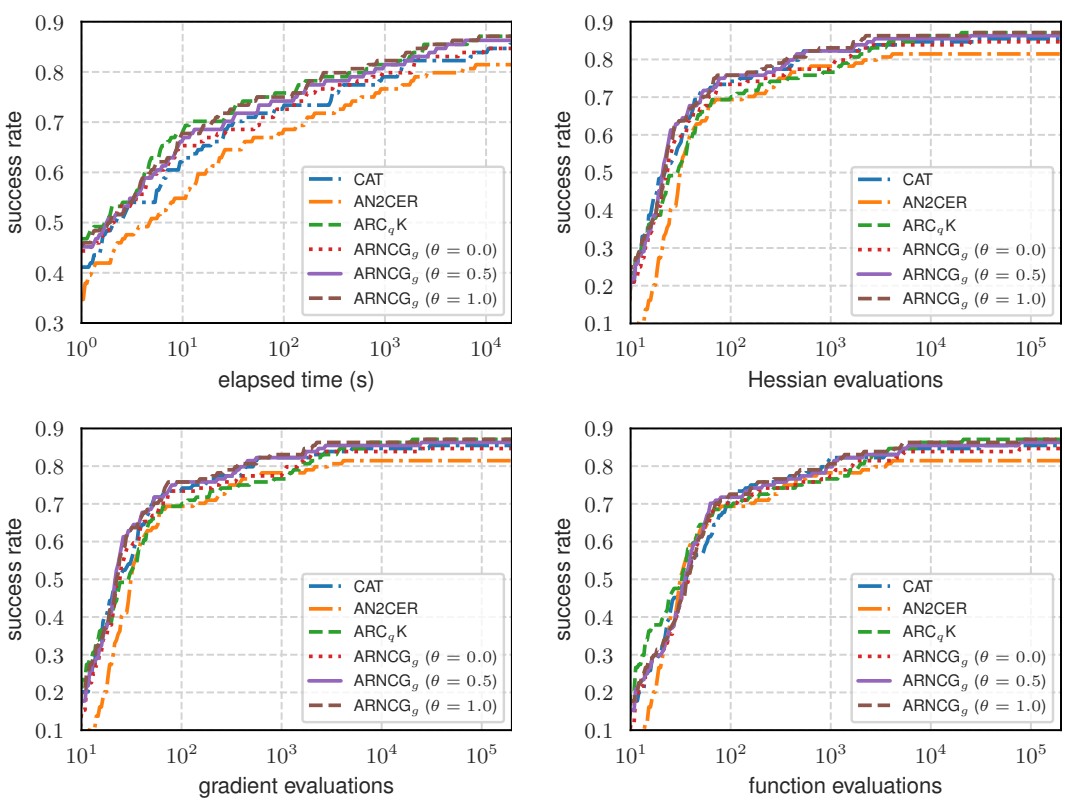

Figure 6: Comparison of success rates as functions of elapsed time, Hessian evaluations, gradient evaluations and function evaluations for solving problems in the CUTEst benchmark. The parameter $\theta$ in Theorem 2.2 varies, and the fallback step is removed, i.e., $\lambda = 0$ in (G.1), and $m_{\max} = 1$.

Table 5: Shifted geometric mean of the relevant metrics for different methods in the CUTEst benchmark. The linesearch failure rate is reported as mean values. The parameter $\theta$ in Theorem 2.2 and the linesearch parameter $m_{\max}$ vary, and $\lambda = 0$.

| | Elapsed Time (s) | Hessian Evaluations | Gradient Evaluations | Function Evaluations | Hessian-vector Products (normalzied) | Success Rate (%) | Linesearch Failure Rate (%) | Second Linesearch Rate (%) |
|---|---|---|---|---|---|---|---|---|
| AN2CER | 36.70 | 170.10 | 172.02 | 176.80 | 31.38 | 81.45 | N/A | N/A |
| CAT | 23.34 | 88.47 | 96.61 | 125.56 | N/A | 85.48 | N/A | N/A |
| ARC$_q$K | 16.16 | 113.21 | 113.84 | 119.51 | 11.97 | 87.10 | N/A | N/A |
| **Results for $m_{\max} = 1$ and $\lambda = 0$** | | | | | | | | |
| Fixed ($\omega_k = \sqrt{\epsilon}$) | 48.10 | 215.60 | 228.47 | 386.84 | 43.97 | 80.65 | 26.12 | 4.73 |
| ARNCG$_g$ ($\theta = 0.0$) | 21.58 | 111.12 | 117.85 | 151.15 | 17.73 | 84.68 | 13.78 | 0.00 |
| ARNCG$_g$ ($\theta = 0.5$) | 18.62 | 87.10 | 92.89 | 126.92 | 14.85 | 86.29 | 15.48 | 1.31 |
| ARNCG$_g$ ($\theta = 1.0$) | 16.71 | 80.86 | 86.41 | 119.51 | 13.77 | 87.10 | 16.08 | 1.38 |
| ARNCG$_g$ ($\theta = 1.5$) | 19.22 | 87.83 | 93.84 | 129.00 | 15.29 | 86.29 | 15.38 | 1.58 |
| ARNCG$_\epsilon$ ($\theta = 0.0$) | 18.39 | 90.95 | 96.67 | 129.71 | 15.28 | 85.48 | 15.49 | 0.50 |
| ARNCG$_\epsilon$ ($\theta = 0.5$) | 18.84 | 90.44 | 96.42 | 129.85 | 15.73 | 85.48 | 15.69 | 0.31 |
| ARNCG$_\epsilon$ ($\theta = 1.0$) | 18.28 | 85.03 | 90.78 | 125.29 | 14.91 | 86.29 | 16.89 | 0.43 |
| ARNCG$_\epsilon$ ($\theta = 1.5$) | 22.65 | 104.83 | 111.81 | 151.03 | 18.83 | 83.87 | 16.05 | 0.42 |
| **Results for $m_{\max} = \lfloor \log_\beta 10^{-8} \rfloor$ and $\lambda = 0$** | | | | | | | | |
| Fixed ($\omega_k = \sqrt{\epsilon}$) | 47.74 | 227.08 | 240.79 | 842.35 | 46.47 | 80.65 | 13.29 | 0.00 |
| ARNCG$_g$ ($\theta = 0.0$) | 27.64 | 143.93 | 152.15 | 213.62 | 23.10 | 83.06 | 0.13 | 0.00 |
| ARNCG$_g$ ($\theta = 0.5$) | 21.20 | 101.86 | 108.25 | 167.06 | 15.96 | 85.48 | 0.15 | 0.00 |
| ARNCG$_g$ ($\theta = 1.0$) | 22.89 | 113.82 | 121.08 | 184.09 | 19.14 | 83.87 | 0.08 | 0.00 |
| ARNCG$_g$ ($\theta = 1.5$) | 22.36 | 109.75 | 116.82 | 185.25 | 18.60 | 84.68 | 0.09 | 0.00 |
| ARNCG$_\epsilon$ ($\theta = 0.0$) | 22.09 | 113.33 | 120.03 | 179.29 | 18.35 | 83.87 | 0.09 | 0.00 |
| ARNCG$_\epsilon$ ($\theta = 0.5$) | 23.12 | 115.58 | 122.82 | 184.87 | 19.58 | 83.06 | 0.12 | 0.00 |
| ARNCG$_\epsilon$ ($\theta = 1.0$) | 22.58 | 105.95 | 112.68 | 176.50 | 17.81 | 84.68 | 0.10 | 0.00 |
| ARNCG$_\epsilon$ ($\theta = 1.5$) | 23.11 | 113.74 | 121.11 | 187.25 | 20.20 | 83.06 | 0.10 | 0.00 |

Table 6: Median of the relevant metrics for different methods in the CUTEst benchmark. The parameter $\theta$ in Theorem 2.2 and the linesearch parameter $m_{\max}$ vary, and $\lambda = 0$.

| | Elapsed Time (s) | Hessian Evaluations | Gradient Evaluations | Function Evaluations | Hessian-vector Products (normalzied) | Success Rate (%) | Linesearch Failure Rate (%) | Second Linesearch Rate (%) |
|---|---|---|---|---|---|---|---|---|
| AN2CER | 4.75 | 30.00 | 30.00 | 30.00 | 4.24 | 81.45 | N/A | N/A |
| CAT | 2.13 | 21.00 | 22.00 | 34.50 | N/A | 85.48 | N/A | N/A |
| $\text{ARC}_q\text{K}$ | 1.71 | 28.50 | 28.50 | 32.00 | 0.62 | 87.10 | N/A | N/A |
| **Results for $m_{\max} = 1$ and $\lambda = 0$** | | | | | | | | |
| Fixed ($\omega_k = \sqrt{\epsilon}$) | 10.75 | 36.50 | 37.50 | 90.00 | 7.29 | 80.65 | 33.16 | 0.00 |
| $\text{ARNCG}_g$ ($\theta = 0.0$) | 2.04 | 22.50 | 23.50 | 37.00 | 1.52 | 84.68 | 1.72 | 0.00 |
| $\text{ARNCG}_g$ ($\theta = 0.5$) | 1.77 | 20.00 | 21.00 | 34.00 | 1.52 | 86.29 | 9.52 | 0.00 |
| $\text{ARNCG}_g$ ($\theta = 1.0$) | 1.89 | 20.50 | 21.50 | 35.50 | 1.52 | 87.10 | 10.82 | 0.00 |
| $\text{ARNCG}_g$ ($\theta = 1.5$) | 2.46 | 22.00 | 23.00 | 38.00 | 1.72 | 86.29 | 10.00 | 0.00 |
| $\text{ARNCG}_\epsilon$ ($\theta = 0.0$) | 1.81 | 20.00 | 21.00 | 35.00 | 1.61 | 85.48 | 3.65 | 0.00 |
| $\text{ARNCG}_\epsilon$ ($\theta = 0.5$) | 1.91 | 20.00 | 21.00 | 35.00 | 1.74 | 85.48 | 7.12 | 0.00 |
| $\text{ARNCG}_\epsilon$ ($\theta = 1.0$) | 1.72 | 21.50 | 22.50 | 38.00 | 1.62 | 86.29 | 10.26 | 0.00 |
| $\text{ARNCG}_\epsilon$ ($\theta = 1.5$) | 1.95 | 22.00 | 23.00 | 40.50 | 1.93 | 83.87 | 10.00 | 0.00 |
| **Results for $m_{\max} = \lfloor \log_\beta 10^{-8} \rfloor$ and $\lambda = 0$** | | | | | | | | |
| Fixed ($\omega_k = \sqrt{\epsilon}$) | 12.27 | 39.50 | 40.50 | 323.50 | 7.59 | 80.65 | 0.00 | 0.00 |
| $\text{ARNCG}_g$ ($\theta = 0.0$) | 3.49 | 25.50 | 26.50 | 53.50 | 1.95 | 83.06 | 0.00 | 0.00 |
| $\text{ARNCG}_g$ ($\theta = 0.5$) | 2.37 | 24.00 | 25.00 | 52.50 | 1.35 | 85.48 | 0.00 | 0.00 |
| $\text{ARNCG}_g$ ($\theta = 1.0$) | 2.84 | 25.00 | 26.00 | 53.00 | 2.13 | 83.87 | 0.00 | 0.00 |
| $\text{ARNCG}_g$ ($\theta = 1.5$) | 2.73 | 26.00 | 27.00 | 54.00 | 2.10 | 84.68 | 0.00 | 0.00 |
| $\text{ARNCG}_\epsilon$ ($\theta = 0.0$) | 2.74 | 23.00 | 24.00 | 49.00 | 1.44 | 83.87 | 0.00 | 0.00 |
| $\text{ARNCG}_\epsilon$ ($\theta = 0.5$) | 2.31 | 24.00 | 25.00 | 53.50 | 1.43 | 83.06 | 0.00 | 0.00 |
| $\text{ARNCG}_\epsilon$ ($\theta = 1.0$) | 2.49 | 26.00 | 27.00 | 55.50 | 1.40 | 84.68 | 0.00 | 0.00 |
| $\text{ARNCG}_\epsilon$ ($\theta = 1.5$) | 2.86 | 25.50 | 26.50 | 55.50 | 2.10 | 83.06 | 0.00 | 0.00 |

**Convection problem**  This equation models physical phenomena such as heat conduction, and is defined as:

$$\begin{cases} \partial_t u + \beta \partial_x u = 0, & (x,t) \in (0, 2\pi) \times (0,1), \\ u(x,0) = \sin x, & x \in [0, 2\pi], \\ u(0,t) = u(2\pi, t), & t \in [0,1]. \end{cases}$$

In the experiments, the convection coefficient is set to $\beta = 40$.

**Reaction problem**  This equation models chemical reaction dynamics, and is given by:

$$\begin{cases} \partial_t u - \rho u(1-u) = 0, & (x,t) \in (0, 2\pi) \times (0,1), \\ u(x,0) = \exp\left(-\frac{8(x-\pi)^2}{\pi^2}\right), & x \in [0, 2\pi], \\ u(0,t) = u(2\pi, t), & t \in [0,1]. \end{cases}$$

The parameter is set to $\rho = 5$ in the experiments.

**Wave problem**  This equation is commonly used to describe wave phenomena such as acoustic and electromagnetic wave propagation:

$$\begin{cases} \partial_{tt}^2 u - 4\partial_{xx}^2 u = 0, & (x,t) \in (0,1) \times (0,1), \\ u(x,0) = \sin(\pi x) + \frac{1}{2}\sin(\beta \pi x), & x \in [0,1], \\ \partial_t u(x,0) = 0, & x \in [0,1], \\ u(0,t) = u(1,t) = 0, & t \in [0,1]. \end{cases}$$

In the experiments, the parameter is set to $\beta = 5$.

## H.2 Results

As suggested by Rathore et al. [47], we adopt the following training strategy: the neural network is first trained using Adam for $I_1$ iterations, followed by L-BFGS for $I_2$ iterations, and finally switched to NNCG or $\text{ARNCG}_g$. Since the per-iteration cost of RNMs varies significantly, we terminate training based on a fixed time budget rather than a fixed iteration count. The time limit is chosen

Table 7: Average training loss and test L2RE on training PINNs over 8 runs.

|  |  | Convection | Reaction | Wave |
|---|---|---|---|---|
| Training Loss | $\text{NNCG}_{0.1}$ | $3.35^{\pm 0.46} \times 10^{-7}$ | $3.03^{\pm 0.76} \times 10^{-7}$ | $4.52^{\pm 1.32} \times 10^{-3}$ |
|  | $\text{NNCG}_{0.01}$ | $8.90^{\pm 1.50} \times 10^{-8}$ | $5.20^{\pm 1.11} \times 10^{-8}$ | $2.79^{\pm 1.10} \times 10^{-3}$ |
|  | $\text{ARNCG}_g$ | $\mathbf{7.57^{\pm 4.93} \times 10^{-10}}$ | $\mathbf{2.04^{\pm 0.59} \times 10^{-9}}$ | $\mathbf{1.75^{\pm 0.50} \times 10^{-5}}$ |
| Test L2RE | $\text{NNCG}_{0.1}$ | $2.85^{\pm 0.37} \times 10^{-3}$ | $1.09^{\pm 0.12} \times 10^{-2}$ | $1.26^{\pm 0.26} \times 10^{-1}$ |
|  | $\text{NNCG}_{0.01}$ | $1.42^{\pm 0.21} \times 10^{-3}$ | $4.81^{\pm 0.54} \times 10^{-3}$ | $8.82^{\pm 2.41} \times 10^{-2}$ |
|  | $\text{ARNCG}_g$ | $\mathbf{6.82^{\pm 2.31} \times 10^{-5}}$ | $\mathbf{8.54^{\pm 1.27} \times 10^{-4}}$ | $\mathbf{6.96^{\pm 0.61} \times 10^{-3}}$ |

such that $\text{ARNCG}_g$ performs approximately 2000 iterations. We set $I_1 = 1000$ and $I_2 = 2000$ for the wave and reaction problems, and $I_1 = 11000$ and $I_2 = 1500$ for the convection problem. The corresponding time budgets are reported in the captions of Figure 7. Each experiment is repeated 8 times with different random seeds.

For NNCG, we evaluate two regularization coefficients, $\rho \in \{0.1, 0.01\}$, and denote the corresponding variants as $\text{NNCG}_\rho$, which were shown to perform best in practice [47]. The parameters for $\text{ARNCG}_g$ follow the setup described in Section G.1, with $\theta = 1$, $\lfloor \log_\beta m_{\max} \rfloor = 10^{-4}$, and $\gamma = 2$. This adjustment to the linesearch parameter is motivated by the relatively high computational cost of each iteration; using a smaller $m_{\max}$ would result in several wasted effort during updates of the Lipschitz estimate $M_k$.

The training loss curves are shown in Figure 7, while the average and best performance across runs are summarized in Table 7.[9] $\text{ARNCG}_g$ consistently outperforms NNCG by a large margin across all problems. We also emphasize that these RNMs do not require storing the full Hessian matrix or any matrix of comparable size, resulting in significantly lower memory usage compared to quasi-Newton methods such as BFGS and the Broyden method [52]. For example, the peak GPU memory usage for the convection, reaction and wave problems is 4.7GB, 3.3GB and 10.2GB, respectively.

---

[9]The L2RE in these tables means the $\ell_2$ relative error. Given the prediction $y \in \mathbb{R}^n$ and the groundtruth $z \in \mathbb{R}^n$, this error is defined by $\sqrt{\frac{\|y-z\|^2}{\|z\|^2}}$.

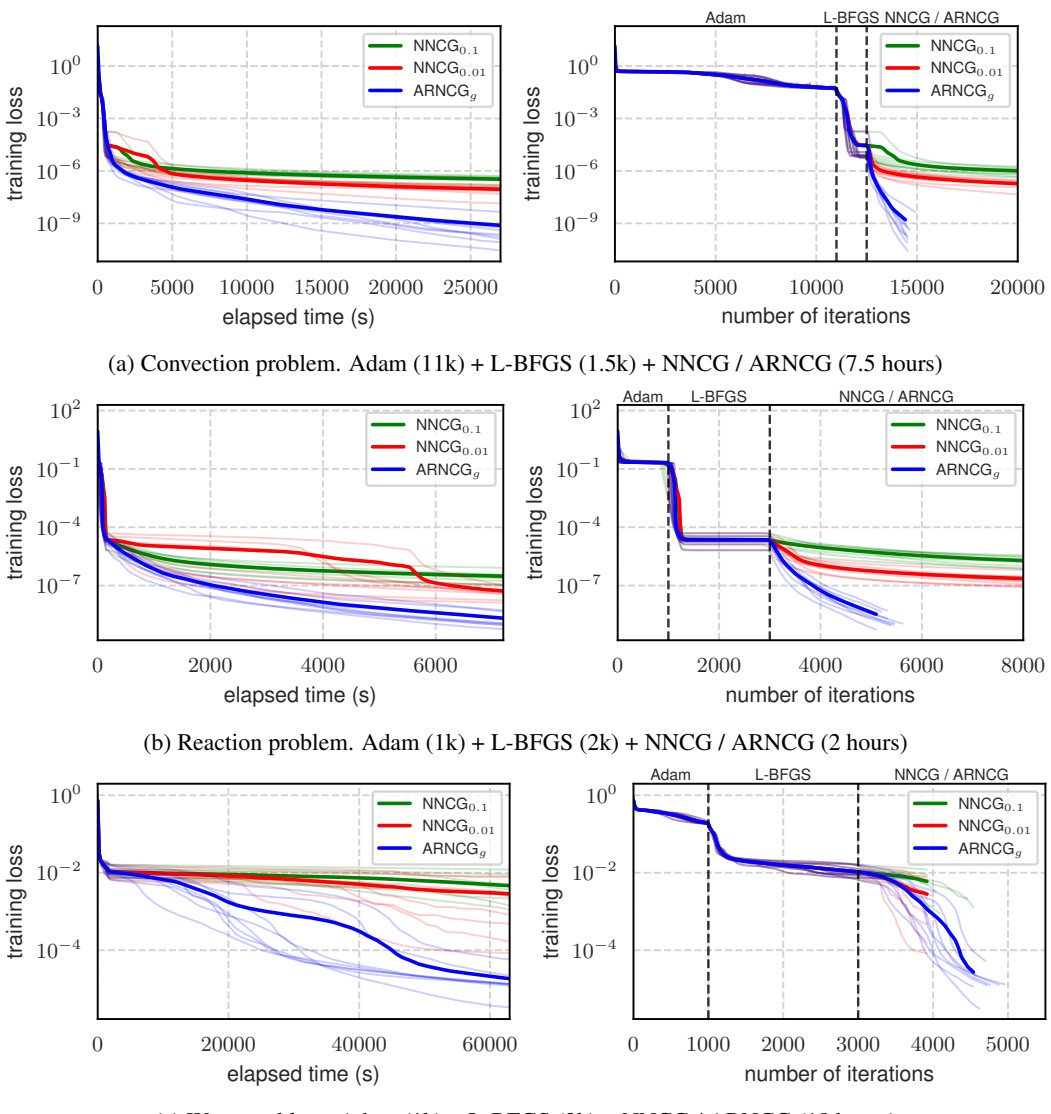

(a) Convection problem. Adam (11k) + L-BFGS (1.5k) + NNCG / ARNCG (7.5 hours)

(b) Reaction problem. Adam (1k) + L-BFGS (2k) + NNCG / ARNCG (2 hours)

(c) Wave problem. Adam (1k) + L-BFGS (2k) + NNCG / ARNCG (18 hours)

Figure 7: Loss curves for training PINNs. The numbers in parentheses for Adam and L-BFGS indicate the number of iterations, while those for NNCG / ARNCG denote total wall-clock time, which is selected such that ARNCG performs approximately 2k iterations. The subscript in NNCG denotes the regularization coefficient. Thin lines are 8 independent runs; the bold line shows the average.

