# OpenReview forum: "A Regularized Newton Method for Nonconvex Optimization with Global and Local Complexity Guarantees"
_NeurIPS.cc/2025/Conference — NeurIPS 2025 poster_

### Official Review · Reviewer_cCif · 2025-06-23

**Clarity:** 3
**Significance:** 3
**Originality:** 3
**Rating:** 5
**Confidence:** 3

**Summary:**

The authors focus on regularized Newton methods, which are widely studied yet still face a trade-off between global and local convergence. A key open question is whether a parameter-free algorithm can achieve both optimal global complexity and quadratic local convergence. To tackle this issue, the authors propose a regularizers using current and previous gradients, and apply a conjugate gradient approach with a negative curvature monitor to solve the regularized Newton equation. The algorithm is adaptive and does not require prior knowledge of the Hessian Lipschitz constant. The proposed method achieves a global complexity of $O(\epsilon^{-\frac{3}{2}})$ in terms of second-order oracle calls and $\tilde{O}(\epsilon^{-\frac{7}{4}})$ for Hessian-vector products. When the iterates converge to a point where the Hessian is positive definite, the method demonstrates quadratic local convergence. Preliminary numerical results highlight the competitiveness of the proposed algorithm.

**Questions:**

- In Table 1, why is the $\tilde{O}(1)$ term missing from the iteration complexity corresponding to He et al. [29, Theorem 1]? From the regularization coefficient, it seems they also assume a lack of prior knowledge about $L_H$ and make an estimate accordingly. Additionally, the phrase "$\epsilon$ is used in the regularization coefficient or minimal eigenvalue computation" is unclear. Could the authors briefly explain why $\epsilon$ is involved in these two places? A simple clarification here might enhance the readability.
- In line 13 of Algorithm 1, is there any theoretical guarantee for the value of $\alpha$? Could the authors provide more insight into this?
- Another point of interest is whether the value of $\theta$ in the algorithm could be adaptively selected rather than using a constant value throughout the entire algorithm? This could potentially improve performance.
- It appears that ARC$_q$K shows excellent time efficiency in most experiments. However, while the authors mention using slightly fewer Hessian evaluations in line 306, it does not seem to result in a better time efficiency. Could the authors further emphasize the practical advantages of the proposed algorithm compared to other methods, particularly in real-world applications? Highlighting these advantages will help readers understand the practical utility of the algorithm better.

**Ethical Concerns:**

["NO or VERY MINOR ethics concerns only"]

**Final Justification:**

The authors have addressed my concerns, and I will keep my initial score.

**Limitations:**

Yes.

**Paper Formatting Concerns:**

None.

**Quality:**

3

**Strengths And Weaknesses:**

### Strengths:
- The overall structure of the paper is clear and well-organized, with a thorough literature review.
- The explanation of the overall algorithmic process is comprehensive, allowing readers to clearly follow the algorithm's workflow. Additionally, although I have not thoroughly checked every step of the proof, it appears that the theoretical justifications are detailed and rigorous, providing solid theoretical guarantees.
- The paper introduces an adaptive Newton-CG algorithm, which cleverly detects the local region in practice. This enables the algorithm to adaptively balance the trade-off between global convergence and local convergence.
- The design for solving the Newton equation for subproblems is innovative, allowing for a cost-effective way to determine the presence of negative curvature directions.
- The experimental setup is detailed, with numerous experiments conducted to validate the effectiveness of the proposed algorithm.

### Weaknesses:
- Some of the symbols in the paper need clearer explanations. For example, the meaning of the symbol $\lesssim$ is not explained. Although line 87 mentions "The notations... are used in their standard sense to represent asymptotic behavior," a more detailed explanation would help readers who are not familiar with complexity theory.
- On line 260, "lemma" should be "proposition". Additionally, on line 272, the definition of $\alpha$ should be provided. On line 303, there is a missing space before the final "and" in the sentence. Lastly, on lines 951 and 956, the formulae should end with a comma instead of a period.
- The paper is somewhat lengthy. Although this is done for theoretical completeness, it makes the review process for a conference submission more difficult.

---

> ### Author Rebuttal · Authors · 2025-07-29
>
> We thank reviewer cCif for the valuable comments.
>
> > Weakness 1. Some of the symbols in the paper need clearer explanations. For example, the meaning of the symbol  is not explained. Although line 87 mentions "The notations... are used in their standard sense to represent asymptotic behavior," a more detailed explanation would help readers who are not familiar with complexity theory.
>
> Thanks for your suggestions, we will add these explanations in revision.
>
> > Weakness 2. On line 260, "lemma" should be "proposition". Additionally, on line 272, the definition of  should be provided. On line 303, there is a missing space before the final "and" in the sentence. Lastly, on lines 951 and 956, the formulae should end with a comma instead of a period.
>
> Thanks for your suggestions, we will fix them in revision.
>
> > Weakness 3. The paper is somewhat lengthy. Although this is done for theoretical completeness, it makes the review process for a conference submission more difficult.
>
> Thank you for the feedback. While we acknowledge that the paper is somewhat lengthy due to the theoretical completeness, we have made an effort to highlight the main ideas and intuitions in the main text. The technical details and extended discussions are deferred to the appendix to maintain the flow of the core narrative. We believe this structure strikes a balance between accessibility and rigor. We are also open to reorganizing or condensing the material further in the final version to improve readability. For example, moving the key ingradient `LipEstimation` of the algorithm to the main text.
>
> > Q1: Clarification about Tab 1.
>
> 1. In their method, the Lipschitz constant estimator $M_k$ is determined through an inner iteration. However, in Table 1, we report only the number of outer iterations of their algorithm, which is why the $\tilde{O}(1)$ term is omitted. Each inner iteration involves solving a regularized Newton equation, and as shown in their Theorem 4 (specifically Eq. (29)), this introduces an additive $O(\log L_H)$ factor due to the constant $T = O(\log L_H)$ (see their Eq. (25)). In contrast, each iteration of the other compared methods solves only one regularized Newton equation.
>
> 2. As an example, in Royer's method they choose the regularizer to be fixed $\sqrt{\epsilon}$, which directly gives a sufficient descent $O(\epsilon^{1.5})$ _at every iteration_, but such a choice is not adaptive and may degrade  empirical performance when the tolerance parameter $\epsilon$ is small (as shown in the "fixed \omega$ case of our experiemnt). The advantage of this choice is that it may simplify the theoretical analysis and give an iteration complexity without a multiplicative log factor. On the other hand, "ME" represents the algorithm needs to compute the minimal eigenvalue to determine some parameters, e.g., the regularizer. We will clarify them in the table.
>
>
> > Q2: The choice of $\alpha$ in Alg 1, L13.
>
> It is based on a observation that in Lem D.3(1), choosing the stepsize as the r.h.s. of (D.9) would lead to a linesearch accepted, and the choice of $\alpha$ is designed to track this stepsize, and leave the unknown part $C_M$ in the r.h.s. determined by the linesearch.
>
> > Q3: Whether $\theta$ can be adaptively chosen？
>
> This is an interesting question. However, adaptively choosing $\theta$ would require a principled selection rule. We suspect that relaxing the Lipschitz continuity assumption on the Hessian to a $\nu$-Hölder condition may offer insights into such a rule, as in this case, setting $\theta > \nu$ yields the best possible local convergence rate.
>
> > Q4: It appears that ARCqK shows excellent time efficiency in most experiments. However, while the authors mention using slightly fewer Hessian evaluations in line 306, it does not seem to result in a better time efficiency. Could the authors further emphasize the practical advantages of the proposed algorithm compared to other methods, particularly in real-world applications? Highlighting these advantages will help readers understand the practical utility of the algorithm better.
>
> Thanks for your suggestion. The differences of ARCqK and our methods in the CUTest benchmark are not quiet significant. ARCqK is a cubic regularization method, but their implementation also solves a regularized Newton equation but they need to search the corresponding regularization coefficient to make the solution satisifies the optimal condition of the cubic regularized subproblem. In contrast, our method can give the descent direction directly and may be helpful when solving the equation is expensive as in the PINN example.

---

> > ### Comment · Reviewer_cCif · 2025-08-04
> > **Thank you**
> >
> > Thank you for your reply. Your patient responses have addressed all of the questions and weaknesses I raised. I will keep my score.

---

### Official Review · Reviewer_9yAV · 2025-07-01

**Clarity:** 3
**Significance:** 3
**Originality:** 3
**Rating:** 4
**Confidence:** 4

**Summary:**

This paper introduces an adaptive Newton’s method that achieves both global and local convergence. In general, optimization algorithms require different hyperparameter settings to ensure global versus local optimal convergence, and these settings are often heuristic and not known in advance. The proposed method addresses this challenge by introducing a regularization term delta_k, which adaptively monitors the algorithm’s behavior to balance global and local convergence. This adaptive scheme enables the algorithm to achieve optimal performance in both regimes, in contrast to previous methods that rely on fixed hyperparameters.

**Questions:**

**Questions:**

(1) When training deep neural networks, how to handle the Hessian matrix? In general, the exact computation of Hessian on the full batch is difficult. Do you use Hessian-free (Hessian-vector products) methods or quasi-Newton methods that does not use the exact Hessian? Or do you adopt lazy Hessian?

(2) Can the algorithm be extended to be stochastic? Most of machine learning problems may require mini-batch sampling. What is the theoretical convergence guarantees?

(3) In Figure 2, ARC and ANRCG performs comparably. Can you please explain? I think ARC is an adaptive version of CR, which also enjoys the (optimal) global and local convergence. The idea of ANRCG should be similar to ARC, am I correct? I also know another parameter-free Newton’s method (https://arxiv.org/abs/2311.11489) that enjoys global and local convergence, can you discuss? I guess the core idea should be very similar.

(4) What are the theoretical convergence results if without Lipschitz smoothness? Is it possible?

**Ethical Concerns:**

["NO or VERY MINOR ethics concerns only"]

**Final Justification:**

I recommend borderline accepting the paper.

**Quality:**

4

**Strengths And Weaknesses:**

**Strength:**

(1) The algorithm is an adaptive Newton-CG, which is both locally and globally convergent. The technique is intuitively convincing.

(2) Theoretical analyses are thorough and rigorous, including the results in the main content and appendices.

(3) Experiments are comprehensive, including CUTEst and PINNs.

**Weaknesses:**

The method requires the computation of exact Hessian matrix.

---

> ### Author Rebuttal · Authors · 2025-07-29
>
> We thank reviewer 9yAV for the valuable comments.
>
> > Q1: When training deep neural networks, how to handle the Hessian matrix? In general, the exact computation of Hessian on the full batch is difficult. Do you use Hessian-free (Hessian-vector products) methods or quasi-Newton methods that does not use the exact Hessian? Or do you adopt lazy Hessian?
>
> Our algorithm only relies on the Hessian-vector products (HVPs), which makes the computational and memory costs relatively modest (see Tab 2). Besides, PyTorch supports HVP computation well, so it does not introduce many implementation overhead.
> We do not adopt the lazy Hessian, but it may be a promising direction for future work.
>
> > Q2: Can the algorithm be extended to be stochastic? Most of machine learning problems may require mini-batch sampling. What is the theoretical convergence guarantees?
>
> Thanks for your questions. We think it can be applied to stochastic settings empirically without theoretically guarantees.  The main challenge lies in the estimation of the Lipschitz constant and the line search procedure, both of which require evaluations of the full objective function. It is not immediately clear how the method behaves when these are replaced with their stochastic counterparts. As discussed in Sec 5, we leave it as a future work.
>
> > Q3: ARC and ARNCG, and Jiang's TR (2311.11489)
>
> 1. Relationship to ARC: ARC is an adaptive CR, but there are important differences to our method. The core step of ARC is to minimize a cubically regularized subproblem, which can also be viewed as  solving a regularized Newton equation $(\nabla^2\varphi(x) + \rho I)d = -\nabla \varphi(x)$, where $\rho \propto \sigma \| d \|$ depends on the solution $\|d\|$. A key step of ARC is to search an appropriate value of $\rho$, while our method uses a prescribed coefficient $\rho$ that does not depends on the solution $d$, which avoids the search step.
>
> 2. Relation to Jiang's trust-region method: Jiang's method is a trust-region type algorithm with a quadratic regularization parameter $\sigma_k$ and an additional free parameter: the trust-region radius $r_k$.
>
>    - For the local convergence rate, as shown in their Tab 2, they first check whether the scale of the minimum eigenvalue of the Hessian is sufficiently large. If so, they set $\sigma_k = 0$ and use the trust-region constraint purely as a globalization mechanism. Under a local convexity assumption, this effectively reduces to the classical Newton method near the solution, allowing for a quadratic convergence rate. However, this strategy requires an additional eigenvalue check and is not directly applicable to regularized Newton methods (RNMs), which rely solely on a regularization parameter. As a result, achieving a quadratic rate is more challenging for RNMs.
>    - For the global convergence rate, their global analysis relies on partitioning iterations into successful and unsuccessful sets, similar to the approach described in our Sec 2. This leads to an extra multiplicative $\log(1/\epsilon)$ factor in the complexity bound. In contrast, our new partitioning technique introduced in Sec 3, along with the second choice of regularization coefficients in Thm 2.2, eliminates this additional factor. We believe our new partitioning rule can also improve their log factor to $\log \log (1/\epsilon)$.
>
> > Q4: What are the theoretical convergence results if without Lipschitz smoothness? Is it possible?
>
> It may be possible to relax the assumption to $\nu$-Holder continuous one. Under this assumption our Lem 3.7 and its refined version in App F.2 guarantees that $\theta > \nu$ gives a best possible local rate $1 + \nu$. However, to obtain an optimal global rate we may need to modify the Lipschitz estimation rule and dynamically estimate the Holder exponent $\nu$, which needs further efforts and we leave it as the future work.

---

> > ### Comment · Reviewer_9yAV · 2025-08-06
> >
> > Thank you so much for the detailed point-to-point response. I will maintain my score, which is positive. For cubic subproblem, we can still solve it approximately (very well) by Hessian-vector products (Krylov subspace). This kind of gradient dependent regularization seems to be more adaptive and meaningful. Therefore, I still think that Adaptive cubic regularization is a good choice and I didnt see many advantages of the proposed method.

---

### Official Review · Reviewer_hnj5 · 2025-07-03

**Clarity:** 1
**Significance:** 2
**Originality:** 2
**Rating:** 3
**Confidence:** 4

**Summary:**

This paper seeks to create a parameter-free algorithm that achieves optimal convergence rates, both locally and globally.  They propose a new class of regularizers based on gradient information and leverage a CG approach with potential early exit if negative curvature information is detected.  Their algorithm leverages the “Capped CG” approach that is essentially CG that additionally exits with negative curvature conditions either if detected explicitly or convergence rates are detected to be too slow, in which case the CG iterates are regenerated to find negative curvature (B1 page 17).  It may also terminate if the convergence of CG exceeds the theoretical threshold denoted by $J(M, \bar\rho, \xi) + 1$.  An addition LipEstimation function is used to improve the estimate of the Lipschitz factor M that is either increase by a factor of $\gamma$ or decreased by a factor $1/\gamma$.
Their algorithm is described in “Algorithm 1” where the function NewtonStep is called potentially twice, first with a trial step and if the step returns FAIL, a fallback step is used.  The heart of NewtonStep is CappedCG based on Royer [49].  Additionally, it provides the choice regularized based on the estimate of $ L_H$,$ M$, as well as $\omega_k$ based on past gradient norms.  They seek to balance the need for the regularization to be proportional to $\sqrt(g_k)$ far from the solution but smaller than $g_k$ when close.  That is, unmodified Newton’s method without a regularizer would ideal if near a positive-definite solution point.  So in a sense the regularize must converge at a much slower rate until sufficiently close to a solution.  So the magic is to detect that the algorithm is behaving like it is sufficiently close or far while balancing that fact that the estimate of $L_H$ itself maybe part of the problem.
After CappedCG, post processing handles the CG converged cases, versus max iteration and/or negative curvature found. They provide the option for a back-tracking like line search to find $\beta^m$ where $m$ is an integer as close to $0$ as possible to get the desired decrease.  In practice, it seems that $m_{max} = 1$ works, indicating the machinery of the algorithm is correctly balancing the step.  (If larger values of $m$ were needed, this would hint that the direction was too large and the system was not sufficiently regularized for the current location.)
They provide comprehensive theoretical convergence proofs as well as numerical results with some CUTEr problems and a physics-informed neural network.

**Questions:**

Is it possible to clarify the algorithm and make it easier to follow, as done by some of the earlier links?  I needed to jump around a lot to follow.

For the CUTEr problems, I think it is important to compare to a simple shifted Steihaug-Toint variant and explicitly report the number of iterations, evaluations, and matrix-vector products in a table, or reference such tables from similar papers.

I think the paper has a lot of strong results, but it is just too long for an article of this length.  Is it possible to lay out the information for better clarity, making it less necessary to flip back and forth from the appendix to the starting page?

**Ethical Concerns:**

["NO or VERY MINOR ethics concerns only"]

**Final Justification:**

The ST method is simple to implement and I think important to compare to. Also detailed tables s hard to beat and I would still need to see detail results as suggested: explicitly report the number of iterations, evaluations, and matrix-vector products in a table, or reference such tables from simil"ar papers."

**Limitations:**

Yes

**Paper Formatting Concerns:**

The body of the paper is in the appendix.  I believe this paper is better for an optimization journal that does not place such strict bounds on the number of pages.  It is clear that a lot of effort was made to reduce the number of lines by making equations inline, etc.  It is just a bad fit for this journal

**Quality:**

3

**Strengths And Weaknesses:**

Weakness and Strength
The paper has an impressive amount of theory backing and seems to achieve its goals of proving desire convergence theoretical bounds for the proposed algorithm.  The numerical results hint that the method is competitive against similar algorithms in this genres.  It provides sound/comprehensive algorithms for generating steps, determining the regularization sequence sizing, accounting for negative curvature, and improving the estimate for $L_H$.  It is sufficiently balanced in that, according to page 40, 1123, the choice of $m_{max}=1$ seems to work well, implying that the algorithm often can take full steps along the generated direction.  The algorithm has a good number of hyperparameters; it appears that very little tuning was needed, and good default values are known.

Ultimately, I think this is a great paper for an optimization journal where significantly longer lengths are permitted.  The algorithm and ingredients is sufficiently complex and interdependent that paper is not comprehensible without going deep into the appendix. Algorithm 1 is itself based on CappedCG and LipEstimation, defined in the appendix, which are critical players to understand the algorithm. The Appendix itself ends on page 48 and the material is not supplemental but critical to the entire paper.  Thus this is not a 9 page paper but a much larger optimization paper that has essentially moved its body to the index.
Lastly, the algorithm seems overly complex for what it is attempting.  It is essentially using CG algorithm with early exits if negative curvature is detected.  It seems a cubic regularized variant of shifted Steihaug-Toint algorithms that behave very similarly: https://optimization-online.org/wp-content/uploads/2004/09/960.pdf and work well for the CUTEr problems, as does Steihaug-Toint itself: https://epubs.siam.org/doi/10.1137/0720042.  The method seems similar in nature to papers like this: https://arxiv.org/pdf/2311.11489 .

---

> ### Author Rebuttal · Authors · 2025-07-29
>
> We thank reviewer hnj5 for the valuable comments.
>
> > Q1: the clarity of this article.
>
> Thank you for the valuable suggestions. If the paper is accepted, an additional page becomes available. We plan to include the `LipEstimation` procedure in the main text. The `CappedCG` component is a slight modification of the method by Royer et al., so we can present a brief (informal) version to convey its main idea. We believe these changes, along with the intuitive explanations and high-level overview in Sec 2.1, would improve clarity and make the algorithm easier to follow.
>
> > Q2: the comparsion with the shifted Steihaug-Toint variant in the CUTest benchmark
>
> Thank you for the suggestion. In our experiments, we selected recent methods from the categories of trust-region methods (CAT), cubic regularization methods (ARCqK), and quadratic regularization methods (AN2CER). According to the results reported in  ARCqK, it is “more efficient than a classic Steihaug–Toint trust-region method.” We believe this comparison is representative, and we will clarify it in the revised version.
>
> > Q3: The method seems similar in nature to papers like this: [https://arxiv.org/pdf/2311.11489](https://arxiv.org/pdf/2311.11489)
>
> There are some notable differences. Jiang's method is a trust-region type algorithm with a quadratic regularization parameter $\sigma_k$ and an additional free parameter: the trust-region radius $r_k$.
>
> - For the local convergence rate, as shown in their Tab 2, they first check whether the scale of the minimum eigenvalue of the Hessian is sufficiently large. If so, they set $\sigma_k = 0$ and use the trust-region constraint purely as a globalization mechanism. Under a local convexity assumption, this effectively reduces to the classical Newton method near the solution, allowing for a quadratic convergence rate. However, this strategy requires an additional eigenvalue check and is not directly applicable to regularized Newton methods (RNMs), which rely solely on a regularization parameter. As a result, achieving a quadratic rate is more challenging for RNMs.
> - For the global convergence rate, their global analysis relies on partitioning iterations into successful and unsuccessful sets, similar to the approach described in our Sec 2. This leads to an extra multiplicative $\log(1/\epsilon)$ factor in the complexity bound. In contrast, our new partitioning technique introduced in Sec 3, along with the second choice of regularization coefficients in Thm 2.2, eliminates this additional factor. We believe our new partitioning rule can also improve their log factor to $\log \log (1/\epsilon)$.

---

### Official Review · Reviewer_MUHH · 2025-07-11

**Clarity:** 3
**Significance:** 2
**Originality:** 2
**Rating:** 4
**Confidence:** 4

**Summary:**

In this submission, the authors propose a novel regularized Newton’s method and demonstrate that two classes of regularizers can achieve both optimal global convergence rates and quadratic local convergence. These new regularizers are constructed using current and historical gradient information, and the authors employ the conjugate gradient method with negative curvature to solve the regularized Newton equation. Notably, the proposed algorithm requires no prior knowledge of the Hessian Lipschitz constant while simultaneously attaining optimal global complexity and quadratic local convergence. Empirical results from numerical experiments align well with the theoretical analysis.

**Questions:**

Could the authors conduct numerical experiments on simpler or toy objective functions, in addition to the more complex non-convex problems? Currently, the results focus on challenging benchmarks such as the CUTEst suite and physics-informed neural networks. While these are valuable examples of non-convex problems, the reviewer is interested in seeing how the proposed algorithm performs on much simpler non-convex functions—for example, a quadratic function with an indefinite Hessian matrix. Such experiments would provide clearer intuition about the behavior and effectiveness of the proposed regularized Newton’s method, complementing the results on more complicated deep neural network models.

**Ethical Concerns:**

["NO or VERY MINOR ethics concerns only"]

**Final Justification:**

Based on the rebuttal, I retain my current score.

**Limitations:**

The authors has presented some limitations in last discussion section of this submission. Please check the other weakness section for all limitations of this submission.

**Paper Formatting Concerns:**

No Paper Formatting Concerns

**Quality:**

3

**Strengths And Weaknesses:**

This is a high-quality theoretical optimization submission with a clear structure and well-written language. It offers two notable theoretical contributions to second-order Newton’s methods:

1. The proposed algorithm is adaptive and parameter-free, meaning it does not require any prior knowledge of the Hessian Lipschitz constant or other parameters. All assumptions are clearly stated in Assumption 2.1, which is relatively simple. This provides a practical advantage over other algorithms that require parameters of the objective function, which can be difficult to estimate in real-world applications.

2. The proposed regularized Newton’s method simultaneously achieves optimal global complexity and quadratic local convergence rates, making it a standout contribution. Combined with its parameter-free nature, this algorithm is both theoretically significant and practically applicable to general non-convex optimization problems.

Furthermore, the authors validate their theoretical claims through numerical experiments on problems such as physics-informed neural networks, with empirical results consistent with the theoretical analysis.

Weaknesses:

1. Although the iteration complexity is clearly stated, the paper should provide more detailed analysis of the computational complexity per iteration, particularly regarding the computational cost of the capped conjugate gradient method used in the algorithm. While the oracle complexity is presented in Theorem 2.3, some of the oracle complexity discussion currently in the appendix should be moved to the main text to improve accessibility.

2. The authors should clarify and be more specific in Theorem 2.2 regarding the local quadratic convergence rates. Instead of the Big-O notation $g_{k + 1} = \mathcal{O}(g_k^2)$ it would be helpful to present the exact form of the local quadratic convergence rate. Additionally, the precise size of the neighborhood where quadratic convergence occurs and the number of iterations required to reach this neighborhood should be clearly stated.

3. In the numerical experiments, it is recommended that the authors compare the proposed Adaptive Regularized Newton-CG method against established quasi-Newton methods such as BFGS, DFP, SR1, and L-BFGS. Moreover, conducting experiments on both convex and non-convex problems would provide a more comprehensive assessment of the algorithm’s performance across different settings.

In summary, this submission is a high-quality optimization paper and deserves a publication.

---

> ### Author Rebuttal · Authors · 2025-07-29
>
> We thank reviewer MUHH for the valuable comments.
>
> > Weakness 1: the computational complexity and oracle complexity
>
> Thank you for the helpful suggestions. The overall computational complexity can be viewed as the product of the number of HVP evaluations (Thm 2.3) and the cost of each HVP evaluation, which is typically fixed and does not vary across iterations. We will clarify this point and move the discussion of oracle complexity accordingly in revision.
>
> > Weakness 2: the exact version of local convergence and the size of the neighborhood
>
> The non-asymptotic statement of the local convergence is presented in Prop E.4 in appendix. We will move it to the main text. For the size of the local convergence neighborhood, some discussions are presented before Lem 3.7. Its size depends on two factors: (1). the assumption of the solution, i.e., $L_H I \succeq \nabla^2 \varphi(x^*) \succeq \alpha I$. Since our regularizer is at most $g_k^{0.5}$, the local convergence rate is at least 1.5 and its neighborhood does not depend on $\theta$; (2). the parameter $\theta$. When entering the neighborhood with 1.5 local rate, it can be boosted to 2 within $O(\log\log poly(L^\theta_H, U_\varphi^\theta))+ 2\log \frac{2\theta-1}{2\theta-2}$ steps.
>
> > Weakness 3 and Question: the comparsion with simper problems and quasi-Newton type methods.
>
> Thanks for your suggestions. The CUTest benchmark also includes a subset of simple functions, e.g., the Rosenbrock's banana function $f(x, y) = (a - x)^2 + b(y - x^2)^2$. There is an illustrative figure in Fig. 4 in p.42. We will clarify this point in the text surrounding the figure.
>
> We note that in prior work on Newton-type methods evaluated on the CUTEst benchmark, quasi-Newton methods such as BFGS, DFP, and SR1 are typically not included for comparison, as they represent a different class of algorithms with distinct convergence properties and design goals. In our case, the focus is on methods that explicitly leverage second-order information.
>
> Nevertheless, we do compare against L-BFGS in the PINN experiment, where it is commonly used in practice. Other quasi-Newton methods such as BFGS and SR1 are less applicable in this setting due to their high memory requirements, as they need to store dense matrices of the same size as the Hessian, which becomes impractical for large-scale problems like PINNs.

---

> > ### Comment · Reviewer_MUHH · 2025-08-05
> >
> > Thank you for your response. I will retain my current score

---

### Decision · Program_Chairs · 2025-09-17

**Decision:**

Accept (poster)

**Comment:**

The paper studies quadratic regularized Newton methods, focusing on trade-off between global and local convergence. The authors propose a parameter-free variant that uses new gradient-based regularizers and a modofied conjugate gradient solver with negative curvature detection. The method avoids dependence on the Hessian Lipschitz constant while achieving optimal global complexity and quadratic local convergence. Empirical results on CUTEst benchmark problems and several physics-informed neural network models suggest the approach is competitive with related second-order methods.

While the contributions are novel and interesting, the paper could be significantly improved by following the reviewers’ suggestions, and the authors are strongly encouraged to implement these in the next revision.